# PARAMETER-BASED VALUE FUNCTIONS

**Francesco Faccio, Louis Kirsch & Jürgen Schmidhuber**
The Swiss AI Lab IDSIA, USI, SUPSI
{francesco,louis,juergen}@idsia.ch

## ABSTRACT

Traditional off-policy actor-critic Reinforcement Learning (RL) algorithms learn value functions of a single target policy. However, when value functions are updated to track the learned policy, they forget potentially useful information about old policies. We introduce a class of value functions called Parameter-Based Value Functions (PBVFs) whose inputs include the policy parameters. They can generalize across different policies. PBVFs can evaluate the performance of any policy given a state, a state-action pair, or a distribution over the RL agent's initial states. First we show how PBVFs yield novel off-policy policy gradient theorems. Then we derive off-policy actor-critic algorithms based on PBVFs trained by Monte Carlo or Temporal Difference methods. We show how learned PBVFs can zero-shot learn new policies that outperform any policy seen during training. Finally our algorithms are evaluated on a selection of discrete and continuous control tasks using shallow policies and deep neural networks. Their performance is comparable to state-of-the-art methods.

## 1 INTRODUCTION

Value functions are central to Reinforcement Learning (RL). For a given policy, they estimate the value of being in a specific state (or of choosing a particular action in a given state). Many RL breakthroughs were achieved through improved estimates of such values, which can be used to find optimal policies (Tesauro, 1995; Mnih et al., 2015). However, learning value functions of arbitrary policies without observing their behavior in the environment is not trivial. Such off-policy learning requires to correct the mismatch between the distribution of updates induced by the behavioral policy and the one we want to learn. Common techniques include Importance Sampling (IS) (Hesterberg, 1988) and deterministic policy gradient methods (DPG) (Silver et al., 2014), which adopt the actor-critic architecture (Sutton, 1984; Konda & Tsitsiklis, 2001; Peters & Schaal, 2008).

Unfortunately, these approaches have limitations. IS suffers from large variance (Cortes et al., 2010; Metelli et al., 2018; Wang et al., 2016) while traditional off-policy actor-critic methods introduce off-policy objectives whose gradients are difficult to follow since they involve the gradient of the action-value function with respect to the policy parameters $\nabla_\theta Q^{\pi_\theta}(s, a)$ (Degris et al., 2012; Silver et al., 2014). This term is usually ignored, resulting in biased gradients for the off-policy objective. Furthermore, off-policy actor-critic algorithms learn value functions of a single target policy. When value functions are updated to track the learned policy, the information about old policies is lost.

We address the problem of generalization across many value functions in the off-policy setting by introducing a class of *parameter-based value functions* (PBVFs) defined for any policy. PBVFs are value functions whose inputs include the policy parameters, the PSSVF $V(\theta)$, PSVF $V(s, \theta)$, and PAVF $Q(s, a, \theta)$. PBVFs can be learned using Monte Carlo (MC) (Metropolis & Ulam, 1949) or Temporal Difference (TD) (Sutton, 1988) methods. The PAVF $Q(s, a, \theta)$ leads to a novel stochastic and deterministic off-policy policy gradient theorem and, unlike previous approaches, can directly compute $\nabla_\theta Q^{\pi_\theta}(s, a)$. Based on these results, we develop off-policy actor-critic methods and compare our algorithms to two strong baselines, ARS and DDPG (Mania et al., 2018; Lillicrap et al., 2015), outperforming them in some environments.

We make theoretical, algorithmic, and experimental contributions: Section 2 introduces the standard MDP setting; Section 3 formally presents PBVFs and derive algorithms for $V(\theta)$, $V(s, \theta)$ and $Q(s, a, \theta)$; Section 4 describes the experimental evaluation using shallow and deep policies; Sections 5 and 6 discuss related and future work. Proofs and derivations can be found in Appendix A.2.

## 2 BACKGROUND

We consider a Markov Decision Process (MDP) (Stratonovich, 1960; Puterman, 2014) $\mathcal{M} = (\mathcal{S}, \mathcal{A}, P, R, \gamma, \mu_0)$ where at each step an agent observes a state $s \in \mathcal{S}$, chooses action $a \in \mathcal{A}$, transitions into state $s'$ with probability $P(s'|s, a)$ and receives a reward $R(s, a)$. The agent starts from an initial state, chosen with probability $\mu_0(s)$. It is represented by a parametrized stochastic policy $\pi_\theta : \mathcal{S} \to \Delta(\mathcal{A})$, which provides the probability of performing action $a$ in state $s$. $\Theta$ is the space of policy parameters. The policy is deterministic if for each state $s$ there exists an action $a$ such that $\pi_\theta(a|s) = 1$. The return $R_t$ is defined as the cumulative discounted reward from time step t: $R_t = \sum_{k=0}^{T-t-1} \gamma^k R(s_{t+k+1}, a_{t+k+1})$, where T denotes the time horizon and $\gamma$ a real-valued discount factor. The performance of the agent is measured by the cumulative discounted expected reward (expected return), defined as $J(\pi_\theta) = \mathbb{E}_{\pi_\theta}[R_0]$. Given a policy $\pi_\theta$, the state-value function $V^{\pi_\theta}(s) = \mathbb{E}_{\pi_\theta}[R_t|s_t = s]$ is defined as the expected return for being in a state $s$ and following policy $\pi_\theta$. By integrating over the state space $\mathcal{S}$, we can express the maximization of the expected cumulative reward in terms of the state-value function $J(\pi_\theta) = \int_\mathcal{S} \mu_0(s) V^{\pi_\theta}(s) \, \mathrm{d}s$. The action-value function $Q^{\pi_\theta}(s, a)$, which is defined as the expected return for performing action $a$ in state $s$, and following the policy $\pi_\theta$, is $Q^{\pi_\theta}(s, a) = \mathbb{E}_{\pi_\theta}[R_t|s_t = s, a_t = a]$, and it is related to the state-value function by $V^{\pi_\theta}(s) = \int_\mathcal{A} \pi_\theta(a|s) Q^{\pi_\theta}(s, a) \, \mathrm{d}a$. We define as $d^{\pi_\theta}(s')$ the discounted weighting of states encountered starting at $s_0 \sim \mu_0(s)$ and following the policy $\pi_\theta$: $d^{\pi_\theta}(s') = \int_\mathcal{S} \sum_{t=1}^\infty \gamma^{t-1} \mu_0(s) P(s \to s', t, \pi_\theta) \, \mathrm{d}s$, where $P(s \to s', t, \pi_\theta)$ is the probability of transitioning to $s'$ after t time steps, starting from s and following policy $\pi_\theta$. Sutton et al. (1999) showed that, for stochastic policies, the gradient of $J(\pi_\theta)$ does not involve the derivative of $d^{\pi_\theta}(s)$ and can be expressed in a simple form:

$$\nabla_\theta J(\pi_\theta) = \int_\mathcal{S} d^{\pi_\theta}(s) \int_\mathcal{A} \nabla_\theta \pi_\theta(a|s) Q^{\pi_\theta}(s, a) \, \mathrm{d}a \, \mathrm{d}s. \tag{1}$$

Similarly, for deterministic policies Silver et al. (2014) obtained the following:

$$\nabla_\theta J(\pi_\theta) = \int_\mathcal{S} d^{\pi_\theta}(s) \nabla_\theta \pi_\theta(s) \nabla_a Q^{\pi_\theta}(s, a)|_{a=\pi_\theta(s)} \, \mathrm{d}s. \tag{2}$$

**Off-policy RL** In off-policy policy optimization, we seek to find the parameters of the policy maximizing a performance index $J_b(\pi_\theta)$ using data collected from a behavioral policy $\pi_b$. Here the objective function $J_b(\pi_\theta)$ is typically modified to be the value function of the target policy, integrated over $d_\infty^{\pi_b}(s) = \lim_{t \to \infty} P(s_t = s|s_0, \pi_b)$, the limiting distribution of states under $\pi_b$ (assuming it exists) (Degris et al., 2012; Imani et al., 2018; Wang et al., 2016). Throughout the paper we assume that the support of $d_\infty^{\pi_b}$ includes the support of $\mu_0$ so that the optimal solution for $J_b$ is also optimal for $J$. Formally, we want to find:

$$J_b(\pi_{\theta*}) = \max_\theta \int_\mathcal{S} d_\infty^{\pi_b}(s) V^{\pi_\theta}(s) \, \mathrm{d}s = \max_\theta \int_\mathcal{S} d_\infty^{\pi_b}(s) \int_\mathcal{A} \pi_\theta(a|s) Q^{\pi_\theta}(s, a) \, \mathrm{d}a \, \mathrm{d}s. \tag{3}$$

Unfortunately, in the off-policy setting, the states are obtained from $d_\infty^{\pi_b}$ and not from $d_\infty^{\pi_\theta}$, hence the gradients suffer from a distribution shift (Liu et al., 2019; Nachum et al., 2019). Moreover, since we have no access to $d_\infty^{\pi_\theta}$, a term in the policy gradient theorem corresponding to the gradient of the action value function with respect to the policy parameters needs to be estimated. This term is usually ignored in traditional off-policy policy gradient theorems[1]. In particular, when the policy is stochastic, Degris et al. (2012) showed that:

$$\nabla_\theta J_b(\pi_\theta) = \int_\mathcal{S} d_\infty^{\pi_b}(s) \int_\mathcal{A} \pi_b(a|s) \frac{\pi_\theta(a|s)}{\pi_b(a|s)} \left( Q^{\pi_\theta}(s, a) \nabla_\theta \log \pi_\theta(a|s) + \nabla_\theta Q^{\pi_\theta}(s, a) \right) \, \mathrm{d}a \, \mathrm{d}s \tag{4}$$

$$\approx \int_\mathcal{S} d_\infty^{\pi_b}(s) \int_\mathcal{A} \pi_b(a|s) \frac{\pi_\theta(a|s)}{\pi_b(a|s)} \left( Q^{\pi_\theta}(s, a) \nabla_\theta \log \pi_\theta(a|s) \right) \, \mathrm{d}a \, \mathrm{d}s. \tag{5}$$

Analogously, Silver et al. (2014) provided the following approximation for deterministic policies[2]:

$$\nabla_\theta J_b(\pi_\theta) = \int_\mathcal{S} d_\infty^{\pi_b}(s) \left( \nabla_\theta \pi_\theta(s) \nabla_a Q^{\pi_\theta}(s, a)|_{a=\pi_\theta(s)} + \nabla_\theta Q^{\pi_\theta}(s, a)|_{a=\pi_\theta(s)} \right) \, \mathrm{d}s \tag{6}$$

$$\approx \int_\mathcal{S} d_\infty^{\pi_b}(s) \left( \nabla_\theta \pi_\theta(s) \nabla_a Q^{\pi_\theta}(s, a)|_{a=\pi_\theta(s)} \right) \, \mathrm{d}s. \tag{7}$$

---

[1]With tabular policies, dropping this term still results in a convergent algorithm (Degris et al., 2012).
[2]In the original formulation of Silver et al. (2014) $d_\infty^{\pi_b}(s)$ is replaced by $d^{\pi_b}(s)$.

Although the term $\nabla_\theta Q^{\pi_\theta}(s, a)$ is dropped, there might be advantages in using the approximate gradient of $J_b$ in order to find the maximum of the original RL objective $J$. Indeed, if we were on-policy, the approximated off-policy policy gradients by Degris et al. (2012); Silver et al. (2014) would revert to the on-policy policy gradients, while an exact gradient for $J_b$ would necessarily introduce a bias. However, when we are off-policy, it is not clear whether this would be better than using the exact gradient of $J_b$ in order to maximize $J$. In this work, we assume that $J_b$ can be considered a good objective for off-policy RL and we derive an exact gradient for it.

## 3 PARAMETER-BASED VALUE FUNCTIONS

In this section, we introduce our parameter-based value functions, the PSSVF $V(\theta)$, PSVF $V(s, \theta)$, and PAVF $Q(s, a, \theta)$ and their corresponding learning algorithms. First, we augment the state and action-value functions, allowing them to receive as an input also the weights of a parametric policy. The parameter-based state-value function (PSVF) $V(s, \theta) = \mathbb{E}[R_t | s_t = s, \theta]$ is defined as the expected return for being in state $s$ and following policy parameterized by $\theta$. Similarly, the parameter-based action-value function (PAVF) $Q(s, a, \theta) = \mathbb{E}[R_t | s_t = s, a_t = a, \theta]$ is defined as the expected return for being in state s, taking action $a$ and following policy parameterized by $\theta$. Using PBVFs, the RL objective becomes: $J(\pi_\theta) = \int_{\mathcal{S}} \mu_0(s) V^\pi(s, \theta) \, \mathrm{d}s$. Maximizing this objective leads to on-policy policy gradient theorems that are analogous to the traditional ones (Sutton et al., 1999; Silver et al., 2014):

**Theorem 3.1.** *Let $\pi_\theta$ be stochastic. For any Markov Decision Process, the following holds:*

$$\nabla_\theta J(\pi_\theta) = \mathbb{E}_{s \sim d^{\pi_\theta}(s), a \sim \pi_\theta(.|s)} \left[ (Q(s, a, \theta) \nabla_\theta \log \pi_\theta(a|s)) \right]. \tag{8}$$

**Theorem 3.2.** *Let $\pi_\theta$ be deterministic. Under standard regularity assumptions (Silver et al., 2014), for any Markov Decision Process, the following holds:*

$$\nabla_\theta J(\pi_\theta) = \mathbb{E}_{s \sim d^{\pi_\theta}(s)} \left[ \nabla_a Q(s, a, \theta)|_{a = \pi_\theta(s)} \nabla_\theta \pi_\theta(s) \right]. \tag{9}$$

Parameter-based value functions allow us also to learn a function of the policy parameters that directly approximates $J(\pi_\theta)$. In particular, the parameter-based start-state-value function (PSSVF) is defined as:

$$V(\theta) := \mathbb{E}_{s \sim \mu_0(s)}[V(s, \theta)] = \int_{\mathcal{S}} \mu_0(s) V(s, \theta) \, \mathrm{d}s = J(\pi_\theta). \tag{10}$$

**Off-policy RL** In the off-policy setting, the objective to be maximized becomes:

$$J_b(\pi_{\theta^*}) = \max_\theta \int_{\mathcal{S}} d_\infty^{\pi_b}(s) V(s, \theta) \, \mathrm{d}s = \max_\theta \int_{\mathcal{S}} \int_{\mathcal{A}} d_\infty^{\pi_b}(s) \pi_\theta(a|s) Q(s, a, \theta) \, \mathrm{d}a \, \mathrm{d}s. \tag{11}$$

By taking the gradient of the performance $J_b$ with respect to the policy parameters $\theta$ we obtain novel policy gradient theorems. Since $\theta$ is continuous, we need to use function approximators $V_\mathbf{w}(\theta) \approx V(\theta)$, $V_\mathbf{w}(s, \theta) \approx V(s, \theta)$ and $Q_\mathbf{w}(s, a, \theta) \approx Q(s, a, \theta)$. Compatible function approximations can be derived to ensure that the approximated value function is following the true gradient. Like in previous approaches, this would result in linearity conditions. However, here we consider nonlinear function approximation and we leave the convergence analysis of linear PBVFs as future work. In episodic settings, we do not have access to $d_\infty^{\pi_b}$, so in the algorithm derivations and in the experiments we approximate it by sampling trajectories generated by the behavioral policy. In all cases, the policy improvement step can be very expensive, due to the computation of the $\arg\max$ over a continuous space $\Theta$. Actor-critic methods can be derived to solve this optimization problem, where the critic (PBVFs) can be learned using TD or MC methods, while the actor is updated following the gradient with respect to the critic. Although our algorithms on PSSVF and PSVF can be used with both stochastic and deterministic policies, removing the stochasticity of the action-selection process might facilitate learning the value function. All our algorithms make use of a replay buffer.

### 3.1 PARAMETER-BASED START-STATE-VALUE FUNCTION $V(\theta)$

We first derive the PSSVF $V(\theta)$. Given the original performance index $J$, and taking the gradient with respect to $\theta$, we obtain:

$$\nabla_\theta J(\pi_\theta) = \int_{\mathcal{S}} \mu_0(s) \nabla_\theta V(s, \theta) \, \mathrm{d}s = \mathbb{E}_{s \sim \mu_0(s)}[\nabla_\theta V(s, \theta)] = \nabla_\theta V(\theta). \tag{12}$$

In Algorithm 1, the critic $V_{\mathbf{w}}(\theta)$ is learned using MC to estimate the value of any policy $\theta$. The actor is then updated following the direction of improvement suggested by the critic. Since the main application of PSSVF is in episodic tasks[3], we optimize for the undiscounted objective.

---

**Algorithm 1** Actor-critic with Monte Carlo prediction for $V(\theta)$

---

    **Input**: Differentiable critic $V_{\mathbf{w}} : \Theta \to \mathcal{R}$ with parameters $\mathbf{w}$; deterministic or stochastic actor $\pi_\theta$ with parameters $\theta$; empty replay buffer $D$
    **Output** : Learned $V_{\mathbf{w}} \approx V(\theta) \forall \theta$, learned $\pi_\theta \approx \pi_{\theta^*}$
  Initialize critic and actor weights $\mathbf{w}, \theta$
  **repeat**:
      Generate an episode $s_0, a_0, r_1, s_1, a_1, r_2, \ldots, s_{T-1}, a_{T-1}, r_T$ with policy $\pi_\theta$
      Compute return $r = \sum_{k=1}^{T} r_k$
      Store $(\theta, r)$ in the replay buffer $D$
      **for** many steps **do**:
        Sample a batch $B = \{(r, \theta)\}$ from $D$
        Update critic by stochastic gradient descent: $\nabla_{\mathbf{w}} \mathbb{E}_{(r,\theta) \in B}[r - V_{\mathbf{w}}(\theta)]^2$
      **end for**
      **for** many steps **do**:
        Update actor by gradient ascent: $\nabla_\theta V_{\mathbf{w}}(\theta)$
      **end for**
  **until** convergence

---

## 3.2 PARAMETER-BASED STATE-VALUE FUNCTION $V(s, \theta)$

Learning the value function using MC approaches can be difficult due to the high variance of the estimate. Furthermore, episode-based algorithms like Algorithm 1 are unable to credit good actions in bad episodes. Gradient methods based on TD updates provide a biased estimate of $V(s, \theta)$ with much lower variance and can credit actions at each time step. Taking the gradient of $J_b(\pi_\theta)$ in the PSVF formulation[4], we obtain:

$$\nabla_\theta J_b(\pi_\theta) = \int_{\mathcal{S}} d_\infty^{\pi_b}(s) \nabla_\theta V(s, \theta) \, \mathrm{d}s = \mathbb{E}_{s \sim d_\infty^{\pi_b}(s)}[\nabla_\theta V(s, \theta)]. \tag{13}$$

Algorithm 2 (Appendix) uses the actor-critic architecture, where the critic is learned via TD[5].

## 3.3 PARAMETER-BASED ACTION-VALUE FUNCTION $Q(s, a, \theta)$

The introduction of the PAVF $Q(s, a, \theta)$ allows us to derive new policy gradients theorems when using a stochastic or deterministic policy.

**Stochastic policy gradients** We want to use data collected from some stochastic behavioral policy $\pi_b$ in order to learn the action-value of a target policy $\pi_\theta$. Traditional off-policy actor-critic algorithms only approximate the gradient of $J_b$, since they do not estimate the gradient of the action-value function with respect to the policy parameters $\nabla_\theta Q^{\pi_\theta}(s, a)$ (Degris et al., 2012; Silver et al., 2014). With PBVFs, we can directly compute this contribution to the gradient. This yields an exact policy gradient theorem for $J_b$:

**Theorem 3.3.** *For any Markov Decision Process, the following holds:*

$$\nabla_\theta J_b(\pi_\theta) = \mathbb{E}_{s \sim d_\infty^{\pi_b}(s), a \sim \pi_b(\cdot|s)} \left[ \frac{\pi_\theta(a|s)}{\pi_b(a|s)} \left( Q(s, a, \theta) \nabla_\theta \log \pi_\theta(a|s) + \nabla_\theta Q(s, a, \theta) \right) \right]. \tag{14}$$

Algorithm 3 (Appendix) uses an actor-critic architecture and can be seen as an extension of Off-PAC (Degris et al., 2012) to PAVF.

---

[3]Alternatives include regenerative method for MC estimation (Rubinstein & Kroese, 2016).

[4]Compared to standard methods based on the state-value function, we can directly optimize the policy following the performance gradient of the PSVF, obtaining a policy improvement step in a model-free way.

[5]Note that the differentiability of the policy $\pi_\theta$ is never required in PSSVF and PSVF.

**Deterministic policy gradients** Estimating $Q(s, a, \theta)$ is in general a difficult problem due to the stochasticity of the policy. Deterministic policies of the form $\pi : \mathcal{S} \to \mathcal{A}$ can help improving the efficiency in learning value functions, since the expectation over the action space is no longer required. Using PBVFs, we can write the performance of a policy $\pi_\theta$ as:

$$J_b(\pi_\theta) = \int_\mathcal{S} d_\infty^{\pi_b}(s)V(s, \theta)\,\mathrm{d}s = \int_\mathcal{S} d_\infty^{\pi_b}(s)Q(s, \pi_\theta(s), \theta)\,\mathrm{d}s. \tag{15}$$

Taking the gradient with respect to $\theta$ we obtain a deterministic policy gradient theorem:

**Theorem 3.4.** *Under standard regularity assumptions (Silver et al., 2014), for any Markov Decision Process, the following holds:*

$$\nabla_\theta J_b(\pi_\theta) = \mathbb{E}_{s \sim d_\infty^{\pi_b}(s)} \left[ \nabla_a Q(s, a, \theta)|_{a=\pi_\theta(s)} \nabla_\theta \pi_\theta(s) + \nabla_\theta Q(s, a, \theta)|_{a=\pi_\theta(s)} \right]. \tag{16}$$

Algorithm 4 (Appendix) uses an actor-critic architecture and can be seen as an extension of DPG (Silver et al., 2014) to PAVF. Despite the novel formulation of algorithm 3, we decided to avoid the stochasticity of the policy and to implement and analyze only the deterministic PAVF.

# 4    EXPERIMENTS[6]

Applying algorithms 1, 2 and 4 directly can lead to convergence to local optima, due to the lack of exploration. In practice, like in standard deterministic actor-critic algorithms, we use a noisy version of the current learned policy in order to act in the environment and collect data to encourage exploration. More precisely, at each episode we use $\pi_{\tilde{\theta}}$ with $\tilde{\theta} = \theta + \epsilon, \epsilon \sim \mathcal{N}(0, \sigma^2 I)$ instead of $\pi_\theta$ and then store $\tilde{\theta}$ in the replay buffer. In our experiments, we report both for our methods as well as the baselines the performance of the policy without parameter noise.

## 4.1    VISUALIZING PBVFS USING LQRS

We start with an illustrative example that allows us to visualize how PBVFs are learning to estimate the expected return over the parameter space. For this purpose, we use an instance of the 1D Linear Quadratic Regulator (LQR) problem and a linear deterministic policy with bias. In figure 1, we plot the episodic $J(\theta)$, the cumulative return that an agent would obtain by acting in the environment using policy $\pi_\theta$ for a single episode, and the cumulative return predicted by the PSSVF $V(\theta)$ for two different times during learning. At the beginning of the learning process, the PSSVF is able to provide just a local estimation of the performance of the agent, since only few data have been observed. However, after 1000 episodes, it is able to provide a more accurate global estimate over the parameter space. Appendix A.4.1 contains a similar visualization for PSVF and PAVF, environment details and hyperparameters used.

## 4.2    MAIN RESULTS

Given the similarities between our PAVF and DPG, Deep Deterministic Policy Gradients (DDPG) is a natural choice for the baseline. Additionally, the PSSVF $V(\theta)$ resembles evolutionary methods as the critic can be interpreted as a global fitness function. Therefore, we decided to include in the comparison Augmented Random Search (ARS) which is known for its state-of-the-art performance using only linear policies in continuous control tasks. For the policy, we use a 2-layer MLP (64,64) with tanh activations and a linear policy followed by a tanh nonlinearity. Figure 2 shows results for deterministic policies with both architectures. In all the tasks the PSSVF is able to achieve at least the same performance compared to ARS, often outperforming it. In the Inverted Pendulum environment, PSVF and PAVF with deep policy are very slow to converge, but they excel in the Swimmer task and MountainCarContinuous. In Reacher, all PBVFs fail to learn the task, while DDPG converges quickly to the optimal policy. We conjecture that for this task it is difficult to perform a search in parameter space. On the other hand, in MountainCarContinuous, the reward is more sparse and DDPG only rarely observes positive reward when exploring in action space. In Appendix A.4 we include additional results for PSSVF and PSVF with stochastic policies and hyperparameters. We analyze the sensitivity of the algorithms on the choice of hyperparameters in Appendix A.4.4.

---

[6]Code is available at: `https://github.com/FF93/Parameter-based-Value-Functions`

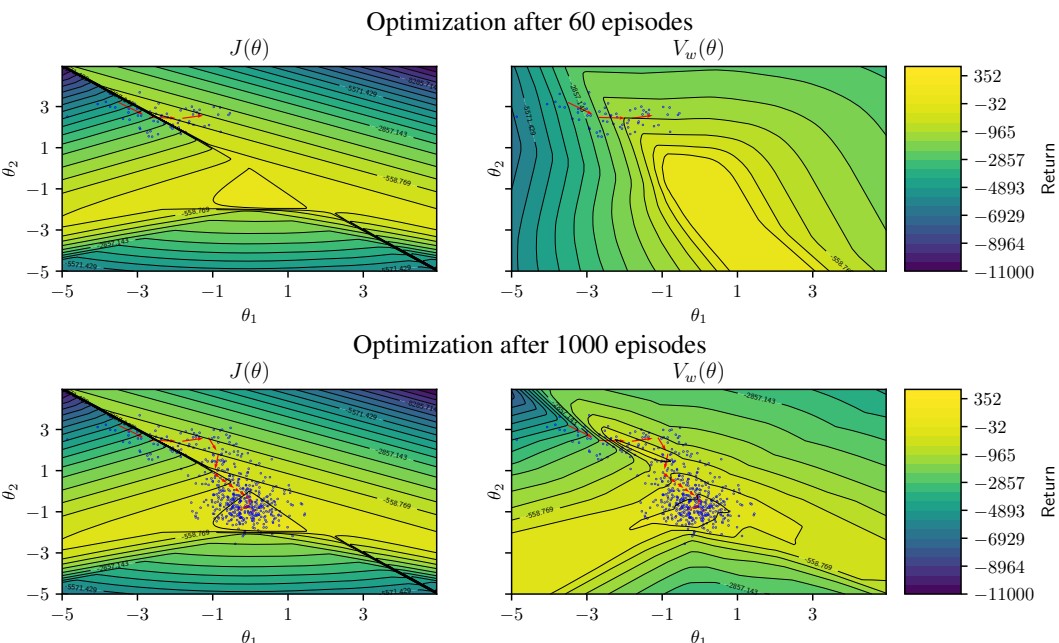

Figure 1: True episodic return $J(\theta)$ and PSSVF estimation $V(\theta)$ as a function of the policy parameters at two different stages in training. The red arrows represent an optimization trajectory in parameter space. The blue dots represent the perturbed policies used to train $V(\theta)$.

### 4.3 ZERO-SHOT LEARNING

In order to test whether PBVFs are generalizing across the policy space, we perform the following experiment with shallow deterministic policies: while learning using algorithm 1, we stop training and randomly initialize 5 policies. Then, without interacting with the environment, we train these policies offline, in a zero-shot manner, following only the direction of improvement suggested by $\nabla_\theta V_w(\theta)$, whose weights $w$ remain frozen. We observe that shallow policies can be effectively trained from scratch. Results for PSSVFs in Swimmer-v3 are displayed in figure 3. In particular, we compare the performance of the policy learned, the best perturbed policy for exploration seen during training and five policies learned from scratch at three different stages in training. We note that after the PSSVF has been trained for 100,000 time steps interactions with the environment (first snapshot), these policies are already able to outperform both the current policy and any policy seen while training the PSSVF. They achieve an average return of 297, while the best observed return was 225. We include additional results for PSVF and PAVF in different environments, using shallow and deep policies in Appendix A.4.2. When using deep policies, we obtain similar results only for the simplest environments. For this task, we use the same hyperparameters as in figure 2.

### 4.4 OFFLINE LEARNING WITH FRAGMENTED BEHAVIORS

In our last experiment, we investigate how PSVFs are able to learn in a completely offline setting. The goal is to learn a good policy in Swimmer-v3 given a fixed dataset containing 100,000 transitions, without additional environment interactions. Furthermore, the policy generating the data is perturbed every 200 time steps, for a total of 5 policies per episode. Observing only incomplete trajectories for each policy parameter makes TD bootstrapping harder: In order to learn, the PSVF needs to generalize across both the state and the parameter space. Given the fixed dataset, we first train the PSVF, minimizing the TD error. Then, at different stages during learning, we train 5 new shallow deterministic policies. Figure 4 describes this process. We note that at the beginning of training, when the PSVF $V(s, \theta)$ has a larger TD error, these policies have poor performance. However, after 7000 gradient updates, they are able to achieve a reward of 237, before eventually degrading to 167. They outperform the best policy in the dataset used to train the PSVF, whose return is only of 58.

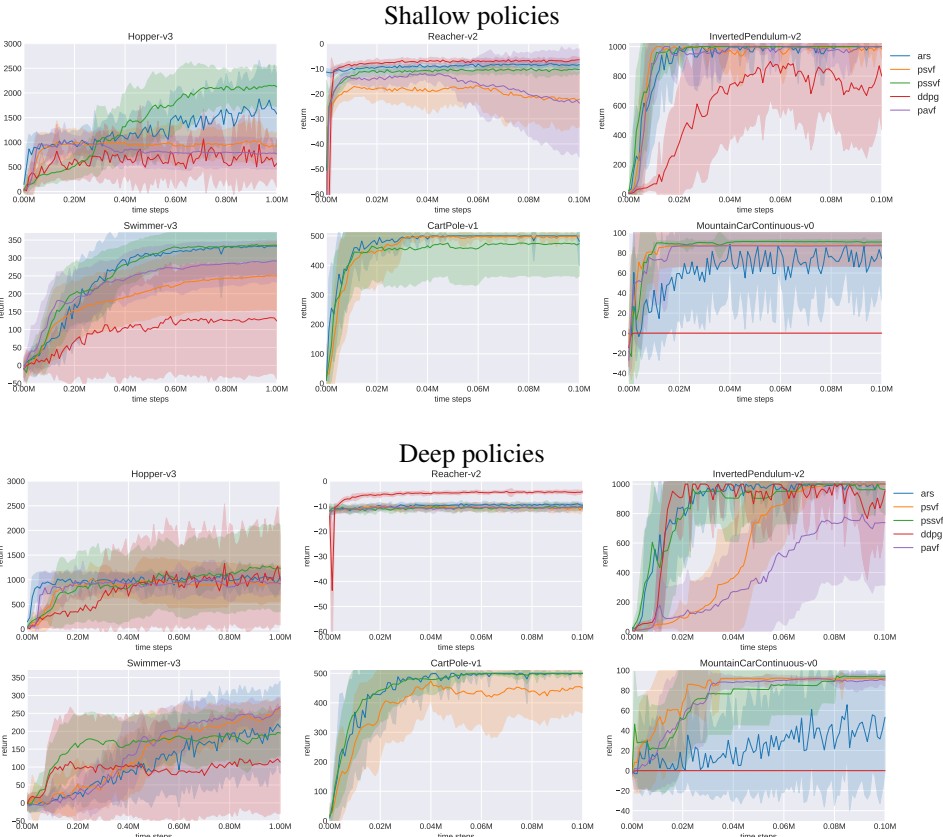

Figure 2: Average return of shallow and deep deterministic policies as a function of the number of time steps used for learning (across 20 runs, one standard deviation), for different environments and algorithms. We use the best hyperparameters found when maximizing the average return.

# 5 RELATED WORK

There are two main classes of similar algorithms performing search in policy parameter space. Evolutionary algorithms (Wierstra et al., 2014; Salimans et al., 2017; Mania et al., 2018) iteratively estimate a fitness function evaluating the performance of a population of policies and then perform gradient ascent in parameter space, often estimating the gradient using finite difference approximation. By replacing the performance of a population through a likelihood estimation, evolutionary algorithms become a form of Parameter Exploring Policy Gradients (Sehnke et al., 2008; 2010). Our methods are similar to evolution since our value function can be seen as a fitness. Unlike evolution, however, our approach allows for obtaining the fitness gradient directly and is more suitable for reusing past data. While direct $V(\theta)$ optimization is strongly related to evolution, our more informed algorithms optimize $V(s, \theta)$ and $Q(s, a, \theta)$. That is, ours both perform a search in policy parameter space AND train the value function and the policy online, without having to wait for the ends of trials or episodes.

The second related class of methods involves surrogate functions (Box & Wilson, 1951; Booker et al., 1998; Moore & Schneider, 1996). They often use local optimizers for generalizing across fitness functions. In particular, Bayesian Optimization (BO) (Snoek et al., 2012; 2015) uses a surrogate function to evaluate the performance of a model over a set of hyperparameters and follows the uncertainty on the surrogate to query the new data to sample. Unlike BO, we do not build a probabilistic model and we use the gradient of the value function instead of a sample from the posterior to decide which policy parameters to use next in the policy improvement step.

The possibility of augmenting the value functions with auxiliary parameters was already

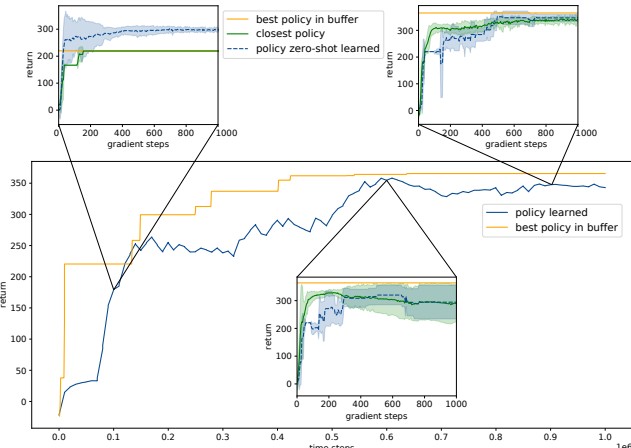

Figure 3: Policies learned from scratch during training. The plot in the center represents the return of the agent learning while interacting with the environment using Algorithm 1. We compare the best noisy policy $\pi_{\tilde{\theta}}$ used for exploration to the policy $\pi_\theta$ learned through the critic. The learning curves in the small plots represent the return obtained by policies trained from scratch following the fixed critic $V_\mathbf{w}(\theta)$ after different time steps of training. The return of the closest policy (L2 distance) in the replay buffer with respect to the policy learned from scratch is depicted in green.

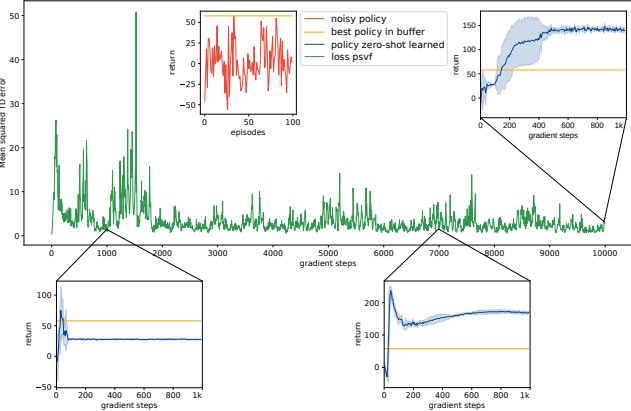

Figure 4: Offline learning of PSVF. We plot the mean squared TD error of a PSVF trained using data coming from a set of noisy policies. In the small plots, we compare the return obtained by policies trained from scratch following the fixed critic $V_\mathbf{w}(s, \theta)$ after different time steps of value function training and the return of the best noisy policy used to train V.

considered in work on General Value Functions (Sutton et al., 2011), where the return is defined with respect to an arbitrary reward function. Universal Value Function Approximators (Schaul et al., 2015) extended this approach to learn a single value function $V^{\pi_\theta}(s, g)$, representing the value, given possible agent goals $g$. In particular, they learn different embeddings for states and goals, exploiting their common structure, and they show generalization to new unseen goals. Similarly, our PSVF $V(s, \theta)$ is able to generalize to unseen policies, observing data for only a few $(s, \theta)$ pairs. General and Universal Value Functions have not been applied to learn a single value function for every possible policy.

Policy Evaluation Networks (PENs) (Harb et al., 2020) are closely related to our work and share the same motivation. PENs focus on the simplest PSSVF $V(\theta)$ trained without an actor-critic architecture. Like in some of our experiments, the authors show how following the direction of improvement suggested by $V(\theta)$ leads to an increase in policy performance. They also suggest to explore in future work a more complex setting where a PSVF $V(s, \theta)$ is learned using an actor-critic

architecture. Our work directly introduces the PSVF $V(s, \theta)$ and PAVF $Q(s, a, \theta)$ and presents novel policy gradient theorems for PAVFs when stochastic or deterministic policies are used. There are many differences between our approach to learning $V(\theta)$ and theirs. For example, we do not use a fingerprint mechanism (Harb et al., 2020) for embedding the weights of complex policies. Instead, we simply parse all the policy weights as inputs to the value function, even in the nonlinear case. Fingerprinting may be important for representing nonlinear policies without losing information about their structure and for saving memory required to store the weights. Harb et al. (2020) focus on the offline setting. They first use randomly initialized policies to perform rollouts and collect reward from the environment. Then, once $V(\theta)$ is trained using the data collected, many gradient ascent steps through V yield new, unseen, randomly initialized policies in a zero-shot manner, exhibiting improved performance. They train their value function using small nonlinear policies of one hidden layer and 30 neurons on Swimmer-v3. They evaluate 2000 deterministic policies on 500 episodes each (1 million policy evaluations), achieving a final expected return of $\approx 180$ on new policies trained from scratch through V. On the other hand, in our zero-shot learning experiment using a linear PSSVF, after only 100 policy evaluations, we obtain a return of 297. In our main experiments, we showed that a fingerprint mechanism is not necessary for the tasks we analyzed: even when using a much bigger 2-layers MLP policy, we are able to outperform the results in PEN. Although Harb et al. (2020) use Swimmer-v3 "to scale up their experiments", our results suggest that Swimmer-v3 does not conclusively demonstrate possible benefits of their policy embedding.

Gradient Temporal Difference (Sutton et al., 2009a;b; Maei et al., 2009; 2010; Maei, 2011) and Emphatic Temporal Difference methods (Sutton et al., 2016) were developed to address convergence under on-policy and off-policy (Precup et al., 2001) learning with function approximation. The first attempt to obtain a stable off-policy actor-critic algorithm under linear function approximation was called Off-PAC (Degris et al., 2012), where the critic is updated using GTD($\lambda$) (Maei, 2011) to estimate the state-value function. This algorithm converges when using tabular policies. However, in general, the actor does not follow the true gradient direction for $J_b$. A paper on DPG (Silver et al., 2014) extended the Off-PAC policy gradient theorem (Degris et al., 2012) to deterministic policies. This was coupled with a deep neural network to solve continuous control tasks through Deep Deterministic Policy Gradients (Lillicrap et al., 2015). Imani et al. (2018) used emphatic weights to derive an exact off-policy policy gradient theorem for $J_b$. Differently from Off-PAC, they do not ignore the gradient of the action-value function with respect to the policy, which is incorporated in the emphatic weighting: a vector that needs to be estimated. Our off-policy policy gradients provide an alternative approach that does not need emphatic weights.

The widely used off-policy objective function $J_b$ suffers the distribution shift problem. Liu et al. (2019) provided an off-policy policy gradient theorem which is unbiased for the true RL objective $J(\pi_\theta)$, introducing a term $d_\infty^{\pi_\theta}/d_\infty^{\pi_b}$ that corrects the mismatch between the states distributions. Despite their sound off-policy formulation, estimating the state weighting ratio remains challenging. All our algorithms are based on the off-policy actor-critic architecture. The two algorithms based on $Q(s, a, \theta)$ can be viewed as analogous to Off-PAC and DPG where the critic is defined for all policies and the actor is updated following the true gradient with respect to the critic.

## 6 LIMITATIONS AND FUTURE WORK

We introduced PBVFs, a novel class of value functions which receive as input the parameters of a policy and can be used for off-policy learning. We showed that PBVFs are competitive to ARS and DDPG (Mania et al., 2018; Lillicrap et al., 2015) while generalizing across policies and allowing for zero-shot training in an offline setting. Despite their positive results on shallow and deep policies, PBVFs suffer the curse of dimensionality when the number of policy parameters is high. Embeddings similar to those used in PENs (Harb et al., 2020) may be useful not only for saving memory and computational time, but also for facilitating search in parameter space. We intend to evaluate the benefits of such embeddings and other dimensionality reduction techniques. We derived off-policy policy gradient theorems, showing how PBVFs follow the true gradient of the performance $J_b$. With these results, we plan to analyze the convergence of our algorithms using stochastic approximation techniques (Borkar, 2009) and test them on environments where traditional methods are known to diverge (Baird, 1995). Finally, we want to investigate how PBVFs applied to supervised learning tasks or POMDPs, can avoid BPTT by mapping the weights of an RNN to its loss.

ACKNOWLEDGMENTS

We thank Paulo Rauber, Imanol Schlag, Miroslav Strupl, Róbert Csordás, Aleksandar Stanić, Anand Gopalakrishnan, Sjoerd Van Steenkiste and Julius Kunze for their feedback. This work was supported by the ERC Advanced Grant (no: 742870). We also thank NVIDIA Corporation for donating a DGX-1 as part of the Pioneers of AI Research Award and to IBM for donating a Minsky machine.

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

# A   APPENDIX

INDEX OF THE APPENDIX

In the following, we briefly recap the contents of the appendix.

- Appendix A.1 contains additional related works

- Appendix A.2 reports all proofs and derivations.

- Appendix A.3 illustrates implementation details and pseudocode.

- Appendix A.4 provides the hyperparameters used in the experiments and further results.

## A.1   ADDITIONAL RELATED WORKS

Recent work (Unterthiner et al., 2020) shows how to map the weights of a trained Convolutional Neural Network to its accuracy. Experiments show how these predictions allow for performance rankings of neural networks on new unseen tasks. These maps are either learned by taking the flattened weights as input or using simple statistics. However, these predictions do not guide the training process of CNNs.

In 1990, adaptive critics trained by TD were used to predict the gradients of an RNN from its activations (Schmidhuber, 1990), avoiding backpropagation through time (BPTT) (Werbos, 1990). This idea was later used to update the weights of a neural network asynchronously (Jaderberg et al., 2017). In our work, the critic is predicting errors instead of gradients. If applied to POMDPs, or supervised learning tasks involving long time lags between relevant events, the PSSVF could avoid BPTT by viewing the parameters of an RNN as a static object and mapping them to their loss (negative reward).

Additional differences between our work and Policy Evaluation Networks (PENs) (Harb et al., 2020) concern the optimization problem: we do not predict a bucket index for discretized reward, but perform a regression task. Therefore our loss is simply the mean squared error between the prediction of $V(\theta)$ and the reward obtained by $\pi_\theta$, while their loss (Harb et al., 2020) is the KL divergence between the predicted and target distributions. Both approaches optimize the undiscounted objective when learning $V(\theta)$.

## A.2   PROOFS AND DERIVATIONS

**Theorem 3.1.** *Let $\pi_\theta$ be stochastic. For any Markov Decision Process, the following holds:*

$$\nabla_\theta J(\pi_\theta) = \mathbb{E}_{s \sim d^{\pi_\theta}(s), a \sim \pi_\theta(.|s)} \left[ (Q(s, a, \theta) \nabla_\theta \log \pi_\theta(a|s)) \right]. \tag{8}$$

*Proof.* The proof follows the standard approach by  Sutton et al. (1999) and we report it for completeness. We start by deriving an expression for $\nabla_\theta V(s, \theta)$:

$$\nabla_\theta V(s,\theta) = \nabla_\theta \int_{\mathcal{A}} \pi_\theta(a|s) Q(s,a,\theta)\, \mathrm{d}a = \int_{\mathcal{A}} \nabla_\theta \pi_\theta(a|s) Q(s,a,\theta) + \pi_\theta(a|s) \nabla_\theta Q(s,a,\theta)\, \mathrm{d}a$$

$$= \int_{\mathcal{A}} \nabla_\theta \pi_\theta(a|s) Q(s,a,\theta) + \pi_\theta(a|s) \nabla_\theta \left( R(s,a) + \gamma \int_{\mathcal{S}} P(s'|s,a) V(s',\theta)\, \mathrm{d}s' \right)\, \mathrm{d}a$$

$$= \int_{\mathcal{A}} \nabla_\theta \pi_\theta(a|s) Q(s,a,\theta) + \pi_\theta(a|s) \gamma \int_{\mathcal{S}} P(s'|s,a) \nabla_\theta V(s',\theta)\, \mathrm{d}s'\, \mathrm{d}a$$

$$= \int_{\mathcal{A}} \nabla_\theta \pi_\theta(a|s) Q(s,a,\theta) + \pi_\theta(a|s) \gamma \int_{\mathcal{S}} P(s'|s,a) \times$$

$$\times \int_{\mathcal{A}} \nabla_\theta \pi_\theta(a'|s') Q(s',a',\theta) + \pi_\theta(a'|s') \gamma \int_{\mathcal{S}} P(s''|s',a') \nabla_\theta V(s'',\theta)\, \mathrm{d}s''\, \mathrm{d}a'\, \mathrm{d}s'\, \mathrm{d}a$$

$$= \int_{\mathcal{S}} \sum_{t=0}^{\infty} \gamma^t P(s \to s', t, \pi_\theta) \int_{\mathcal{A}} \nabla_\theta \pi_\theta(a|s') Q(s',a,\theta)\, \mathrm{d}a\, \mathrm{d}s'.$$

Taking the expectation with respect to $s_0 \sim \mu_0(s)$ we have:

$$\nabla_\theta J(\theta) = \nabla_\theta \int_{\mathcal{S}} \mu_0(s) V(s,\theta)\, \mathrm{d}s = \int_{\mathcal{S}} \mu_0(s) \nabla_\theta V(s,\theta)\, \mathrm{d}s$$

$$= \int_{\mathcal{S}} \mu_0(s) \int_{\mathcal{S}} \sum_{t=0}^{\infty} \gamma^t P(s \to s', t, \pi_\theta) \int_{\mathcal{A}} \nabla_\theta \pi_\theta(a|s) Q(s,a,\theta)\, \mathrm{d}s'\, \mathrm{d}a\, \mathrm{d}s$$

$$= \int_{\mathcal{S}} d^{\pi_\theta}(s) \int_{\mathcal{A}} \nabla_\theta \pi_\theta(a|s) Q(s,a,\theta)\, \mathrm{d}a\, \mathrm{d}s$$

$$= \mathbb{E}_{s \sim d^{\pi_\theta}(s), a \sim \pi_\theta(.|s)} \left[ (Q(s,a,\theta) \nabla_\theta \log \pi_\theta(a|s)) \right].$$

$\square$

**Theorem 3.2.** *Let $\pi_\theta$ be deterministic. Under standard regularity assumptions (Silver et al., 2014), for any Markov Decision Process, the following holds:*

$$\nabla_\theta J(\pi_\theta) = \mathbb{E}_{s \sim d^{\pi_\theta}(s)} \left[ \nabla_a Q(s,a,\theta)|_{a=\pi_\theta(s)} \nabla_\theta \pi_\theta(s) \right]. \tag{9}$$

*Proof.* The proof follows the standard approach by Silver et al. (2014) and we report it for completeness. We start by deriving an expression for $\nabla_\theta V(s,\theta)$:

$$\nabla_\theta V(s,\theta) = \nabla_\theta Q(s, \pi_\theta(s), \theta) = \nabla_\theta \left( R(s, \pi_\theta(s)) + \gamma \int_{\mathcal{S}} P(s'|s, \pi_\theta(s)) V(s',\theta)\, \mathrm{d}s' \right)$$

$$= \nabla_\theta \pi_\theta(s) \nabla_a R(s,a)|_{a=\pi_\theta(s)} +$$

$$+ \gamma \int_{\mathcal{S}} P(s'|s, \pi_\theta(s)) \nabla_\theta V(s',\theta) + \nabla_\theta \pi_\theta(s) \nabla_a P(s'|s,a)|_{a=\pi_\theta(s)}\, \mathrm{d}s'$$

$$= \nabla_\theta \pi_\theta(s) \nabla_a \left( R(s,a) + \gamma \int_{\mathcal{S}} P(s'|s,a) V(s',\theta)\, \mathrm{d}s' \right)|_{a=\pi_\theta(s)} +$$

$$+ \gamma \int_{\mathcal{S}} P(s'|s, \pi_\theta(s)) \nabla_\theta V(s',\theta)\, \mathrm{d}s'$$

$$= \nabla_\theta \pi_\theta(s) \nabla_a Q(s,a,\theta)|_{a=\pi_\theta(s)} + \gamma \int_{\mathcal{S}} P(s'|s, \pi_\theta(s)) \nabla_\theta V(s',\theta)\, \mathrm{d}s'$$

$$= \nabla_\theta \pi_\theta(s) \nabla_a Q(s,a,\theta)|_{a=\pi_\theta(s)} +$$

$$+ \gamma \int_{\mathcal{S}} P(s'|s, \pi_\theta(s)) \nabla_\theta \pi_\theta(s') \nabla_a Q(s',a,\theta)|_{a=\pi_\theta(s')}\, \mathrm{d}s' +$$

$$+ \gamma \int_{\mathcal{S}} P(s'|s, \pi_\theta(s)) \gamma \int_{\mathcal{S}} P(s''|s', \pi_\theta(s')) \nabla_\theta V(s'',\theta)\, \mathrm{d}s''\, \mathrm{d}s'$$

$$= \int_{\mathcal{S}} \sum_{t=0}^{\infty} \gamma^t P(s \to s', t, \pi_\theta) \nabla_\theta \pi_\theta(s') \nabla_a Q(s',a,\theta)|_{a=\pi_\theta(s')}\, \mathrm{d}s'$$

Taking the expectation with respect to $s_0 \sim \mu_0(s)$ we have:

$$
\nabla_\theta J(\theta) = \nabla_\theta \int_{\mathcal{S}} \mu_0(s) V(s,\theta) \, \mathrm{d}s = \int_{\mathcal{S}} \mu_0(s) \nabla_\theta V(s,\theta) \, \mathrm{d}s
$$

$$
= \int_{\mathcal{S}} \mu_0(s) \int_{\mathcal{S}} \sum_{t=0}^{\infty} \gamma^t P(s \to s', t, \pi_\theta) \nabla_\theta \pi_\theta(s') \nabla_a Q(s', a, \theta)|_{a=\pi_\theta(s')} \, \mathrm{d}s' \, \mathrm{d}s
$$

$$
= \int_{\mathcal{S}} d^{\pi_\theta}(s) \nabla_\theta \pi_\theta(s) \nabla_a Q(s, a, \theta)|_{a=\pi_\theta(s)} \, \mathrm{d}s
$$

$$
= \mathbb{E}_{s \sim d^{\pi_\theta}(s)} \left[ \nabla_\theta \pi_\theta(s) \nabla_a Q(s, a, \theta)|_{a=\pi_\theta(s)} \right]
$$

$\square$

**Theorem 3.3.** *For any Markov Decision Process, the following holds:*

$$
\nabla_\theta J_b(\pi_\theta) = \mathbb{E}_{s \sim d^{\pi_b}_\infty(s), a \sim \pi_b(.|s)} \left[ \frac{\pi_\theta(a|s)}{\pi_b(a|s)} \left( Q(s, a, \theta) \nabla_\theta \log \pi_\theta(a|s) + \nabla_\theta Q(s, a, \theta) \right) \right]. \quad (14)
$$

*Proof.*

$$
\nabla_\theta J_b(\pi_\theta) = \nabla_\theta \int_{\mathcal{S}} d^{\pi_b}_\infty(s) V(s,\theta) \, \mathrm{d}s \tag{17}
$$

$$
= \nabla_\theta \int_{\mathcal{S}} d^{\pi_b}_\infty(s) \int_{\mathcal{A}} \pi_\theta(a|s) Q(s, a, \theta) \, \mathrm{d}a \, \mathrm{d}s \tag{18}
$$

$$
= \int_{\mathcal{S}} d^{\pi_b}_\infty(s) \int_{\mathcal{A}} [Q(s, a, \theta) \nabla_\theta \pi_\theta(a|s) + \pi_\theta(a|s) \nabla_\theta Q(s, a, \theta)] \, \mathrm{d}a \, \mathrm{d}s \tag{19}
$$

$$
= \int_{\mathcal{S}} d^{\pi_b}_\infty(s) \int_{\mathcal{A}} \frac{\pi_b(a|s)}{\pi_b(a|s)} \pi_\theta(a|s) [Q(s, a, \theta) \nabla_\theta \log \pi_\theta(a|s) + \nabla_\theta Q(s, a, \theta)] \, \mathrm{d}a \, \mathrm{d}s \tag{20}
$$

$$
= \mathbb{E}_{s \sim d^{\pi_b}_\infty(s), a \sim \pi_b(.|s)} \left[ \frac{\pi_\theta(a|s)}{\pi_b(a|s)} \left( Q(s, a, \theta) \nabla_\theta \log \pi_\theta(a|s) + \nabla_\theta Q(s, a, \theta) \right) \right] \tag{21}
$$

$\square$

**Theorem 3.4.** *Under standard regularity assumptions (Silver et al., 2014), for any Markov Decision Process, the following holds:*

$$
\nabla_\theta J_b(\pi_\theta) = \mathbb{E}_{s \sim d^{\pi_b}_\infty(s)} \left[ \nabla_a Q(s, a, \theta)|_{a=\pi_\theta(s)} \nabla_\theta \pi_\theta(s) + \nabla_\theta Q(s, a, \theta)|_{a=\pi_\theta(s)} \right]. \quad (16)
$$

*Proof.*

$$
\nabla_\theta J_b(\pi_\theta) = \int_{\mathcal{S}} d^{\pi_b}_\infty(s) \nabla_\theta Q(s, \pi_\theta(s), \theta) \, \mathrm{d}s \tag{22}
$$

$$
= \int_{\mathcal{S}} d^{\pi_b}_\infty(s) \left[ \nabla_a Q(s, a, \theta)|_{a=\pi_\theta(s)} \nabla_\theta \pi_\theta(s) + \nabla_\theta Q(s, a, \theta)|_{a=\pi_\theta(s)} \right] \mathrm{d}s \tag{23}
$$

$$
= \mathbb{E}_{s \sim d^{\pi_b}_\infty(s)} \left[ \nabla_a Q(s, a, \theta)|_{a=\pi_\theta(s)} \nabla_\theta \pi_\theta(s) + \nabla_\theta Q(s, a, \theta)|_{a=\pi_\theta(s)} \right] \tag{24}
$$

$\square$

## A.3 IMPLEMENTATION DETAILS

### A.3.1

In this appendix, we report the implementation details for PSSVF, PSVF, PAVF and the baselines. We specify for each hyperparameter, which algorithms and tasks are sharing them.

Shared hyperparameters:

- Deterministic policy architecture (continuous control tasks): We use three different deterministic policies: a linear mapping between states and actions; a single-layer MLP with 32 neurons and tanh activation; a 2-layers MLP (64,64) with tanh activations. All policies contain a bias term and are followed by a tanh nonlinearity in order to bound the action.

- Deterministic policy architecture (discrete control tasks): We use three different deterministic policies: a linear mapping between states and a probability distribution over actions; a single-layer MLP with 32 neurons and tanh activation; a 2-layers MLP (64,64) with tanh activations. The deterministic action $a$ is obtained choosing $a = \arg\max \pi_\theta(a|s)$. All policies contain a bias term.

- Stochastic policy architecture (continuous control tasks): We use three different stochastic policies: a linear mapping; a single-layer MLP with 32 neurons and tanh activation; a 2-layers MLP (64,64) with tanh activations all mapping from states to the mean of a Normal distribution. The variance is state-independent and parametrized as $e^{2\Omega}$ with diagonal $\Omega$. All policies contain a bias term. Actions sampled are given as input to a tanh nonlinearity in order to bound them in the action space.

- Stochastic policy architecture (discrete control tasks): We use three different deterministic policies: a linear mapping between states and a probability distribution over actions; a single-layer MLP with 32 neurons and tanh activation; a 2-layers MLP (64,64) with tanh activations. All policies contain a bias term.

- Policy initialization: all weights and biases are initialized using the default Pytorch initialization for PBVFs and DDPG and are set to zero for ARS.

- Critic architecture: 2-layers MLP (512,512) with bias and ReLU activation functions for PSVF, PAVF; 2-layers MLP (256,256) with bias and ReLU activation functions for DDPG.

- Critic initialization: all weights and biases are initialized using the default Pytorch initialization for PBVFs and DDPG.

- Batch size: 128 for DDPG, PSVF, PAVF; 16 for PSSVF.

- Actor's frequency of updates: every episode for PSSVF; every batch of episodes for ARS; every 50 time steps for DDPG, PSVF, PAVF.

- Critic's frequency of updates: every episode for PSSVF; every 50 time steps for DDPG, PSVF, PAVF.

- Replay buffer: the size is 100k; data are sampled uniformly.

- Optimizer: Adam for PBVFs and DDPG.

Tuned hyperparameters:

- Number of directions and elite directions for ARS ([directions, elite directions]): tuned with values in $[[1,1],[4,1],[4,4],[16,1],[16,4],[16,16]]$.

- Policy's learning rate: tuned with values in $[1e-2, 1e-3, 1e-4]$.

- Critic's learning rate: tuned with values in $[1e-2, 1e-3, 1e-4]$.

- Noise for exploration: the perturbation for the action (DDPG) or the parameter is sampled from $\mathcal{N}(0, \sigma I)$ with $\sigma$ tuned with values in $[1, 1e-1]$ for PSSVF, PSVF, PAVF; $[1e-1, 1e-2]$ for DDPG; $[1, 1e-1, 1e-2, 1e-3]$ for ARS. For stochastic PSSVF and PSVF we include also the value $\sigma = 0$, although it almost never results optimal.

Environment hyperparameters:

- Environment interactions: 1M time steps for Swimmer-v3 and Hopper-v3; 100k time steps for all other environments.

- Discount factor for TD algorithms: 0.999 for Swimmer; 0.99 for all other environments.

- Survival reward in Hopper: True for DDPG, PSVF, PAVF; False for ARS, PSSVF.

Algorithm-specific hyperparameters:

- Critic's number of updates: 50 for DDPG, 5 for PSVF and PAVF; 10 for PSSVF.

- Actor's number of updates: 50 for DDPG, 1 for PSVF and PAVF; 10 for PSSVF.
- Observation normalization: False for DDPG; True for all other algorithms.
- Starting steps in DDPG (random actions and no training): first $1\%$.
- Polyak parameter in DDPG: 0.995.

**PAVF** $\nabla_\theta Q(s, a, \theta)$ **ablation**    We investigate the effect of the term $\nabla_\theta Q(s, a, \theta)$ in the off-policy policy gradient theorem for deterministic PAVF. We follow the same methodology as in our main experiments to find the optimal hyperparameters when updating using the now biased gradient:

$$\nabla_\theta J_b(\pi_\theta) \approx \mathbb{E}_{s \sim d_\infty^{\pi_b}(s)} \left[ \nabla_a Q(s, a, \theta)|_{a=\pi_\theta(s)} \nabla_\theta \pi_\theta(s) \right], \tag{25}$$

which corresponds to the gradient that DDPG is following. Figure 5 reports the results for Hopper and Swimmer using shallow and deep policies. We observe a significant drop in performance in Swimmer when removing part of the gradient. In Hopper the loss of performance is less significant, possibly because both algorithms tend to converge to the same sub-optimal behavior.

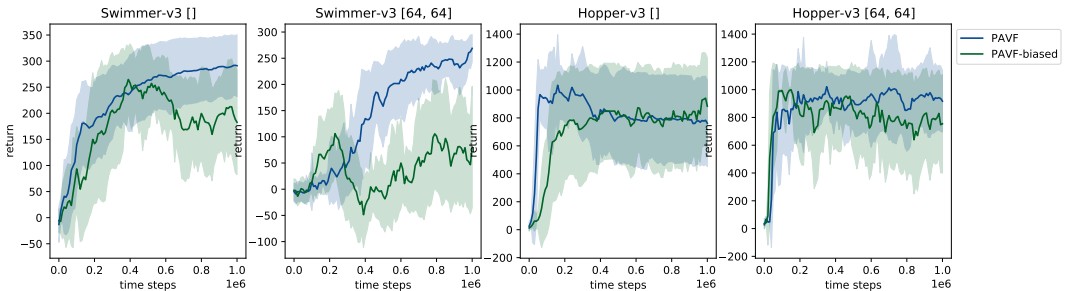

Figure 5: Performance of PAVF and biased PAVF (PAVF without the gradient of the action-value function with respect to the policy parameters) using deterministic policies. We use the hyperparameters maximizing average return and report the best hyperparameters found for the biased version in Table 1. Learning curves are averaged over 20 seeds.

Table 1: Table of best hyperparameters for biased PAVFs

| **Learning rate policy** | Policy: | [] | [64,64] |
|---|---|---|---|
| | Metric: | avg | avg |
| Swimmer-v3 | | 1e-3 | 1e-4 |
| Hopper-v3 | | 1e-4 | 1e-4 |
| **Learning rate critic** | | | |
| Swimmer-v3 | | 1e-4 | 1e-4 |
| Hopper-v3 | | 1e-3 | 1e-3 |
| **Noise for exploration** | | | |
| Swimmer-v3 | | 1.0 | 1.0 |
| Hopper-v3 | | 0.1 | 0.1 |

**ARS**    For ARS, we used the official implementation provided by the authors and we modified it in order to use nonlinear policies. More precisely, we used the implementation of ARSv2-t (Mania et al., 2018), which uses observation normalization, elite directions and an adaptive learning rate based on the standard deviation of the return collected. To avoid divisions by zero, which may happen if all data sampled have the same return, we perform the standardization only in case the standard deviation is not zero. In the original implementation of ARS (Mania et al., 2018), the survival bonus for the reward in the Hopper environment is removed to avoid local minima. Since we wanted our PSSVF to be close to their setting, we also applied this modification. We did not remove the survival bonus from all TD algorithms and we did not investigate how this could affect their performance. We provide a comparison of the performance of PSSVF with and without the bonus in figure 6 using deterministic policies.

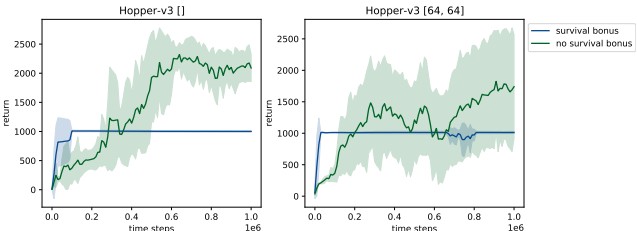

Figure 6: Performance of PSSVF with and without the survival bonus for the reward in Hopper-v3 when using the hyperparameters maximizing the average return. Learning curves are averaged over 5 seeds.

**DDPG**  For DDPG, we used the Spinning Up implementation provided by OpenAI (Achiam, 2018), which includes target networks for the actor and the critic and no learning for a fixed set of time steps, called starting steps. We did not include target networks and starting steps in our PBVFs, although they could potentially help stabilizing training. The implementation of DDPG that we used (Achiam, 2018) does not use observation normalization. In preliminary experiments we observed that it failed to significantly increase or decrease performance, hence we did not use it. Another difference between our TD algorithms and DDPG consists in the number of updates of the actor and the critic. Since DDPG's critic needs to keep track of the current policy, the critic and the actor are updated in a nested form, with the first's update depending on the latter and vice versa. Our PSVF and PAVF do not need to track the policy learned, hence, when it is time to update, we need only to train once the critic for many gradient steps and then train the actor for many gradient steps. This requires less compute. On the other hand, when using nonlinear policies, our PBVFs suffer the curse of dimensionality. For this reason, we profited from using a bigger critic. In preliminary experiments, we observed that DDPG's performance did not change significantly through a bigger critic. We show differences in performance for our methods when removing observation normalization and when using a smaller critic (MLP(256,256)) in figure 7. We observe that the performance is decreasing if observation normalization is removed. However, only for shallow policies in Swimmer and deep policies in Hopper there seems to be a significant benefit. Future work will assess when bigger critics help.

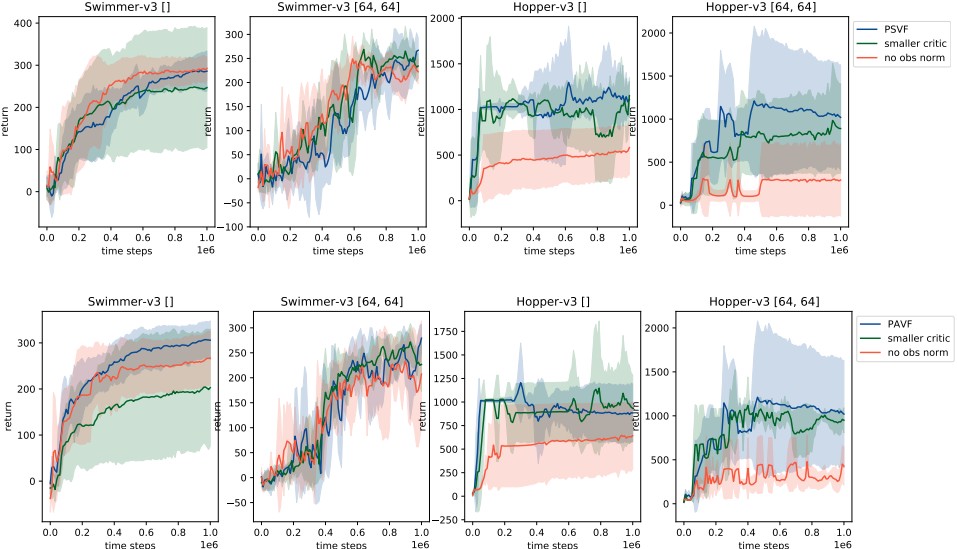

Figure 7: Learning curves for PSVF and PAVF for different environments and policies removing observation normalization and using a smaller critic. We use the hyperparameters maximizing the average return. Learning curves are averaged over 5 seeds. For this ablation we use deterministic policies.

**Discounting in Swimmer** For TD algorithms, we chose a fixed discount factor $\gamma = 0.99$ for all environments but Swimmer-v3. This environment is known to be challenging for TD based algorithms because discounting causes the agents to become too short-sighted. We observed that, with the standard discounting, DDPG, PSVF and PAVF were not able to learn the task. However, making the algorithms more far-sighted greatly improved their performance. In figure 8 we report the return obtained by DDPG, PSVF and PAVF for different values of the discount factor in Swimmer when using deterministic policies.

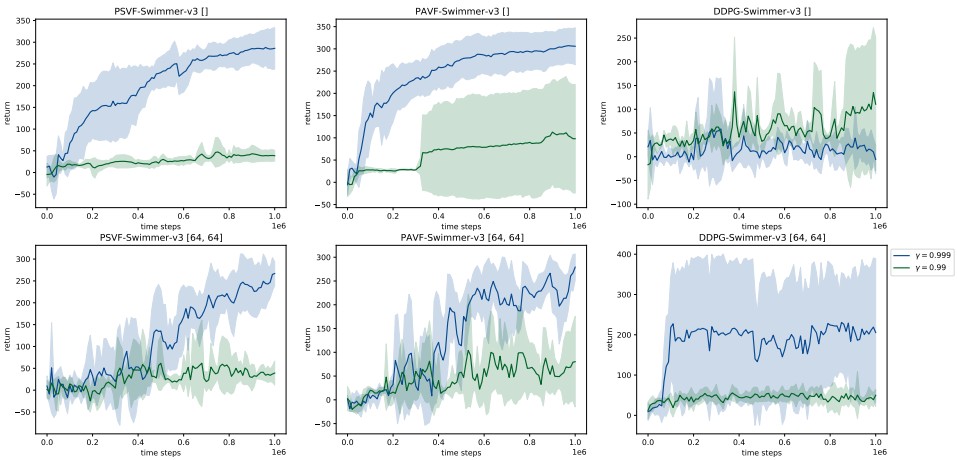

Figure 8: Effect of different choices of the discount factor in Swimmer-v3 for PSVF, PAVF and DDPG, with shallow and deep deterministic policies. We use the hyperparameters maximizing the average return. Learning curves are averaged over 5 seeds

### A.3.2 PSEUDOCODE

---

**Algorithm 2** Actor-critic with TD prediction for $V(s, \theta)$

---

**Input**: Differentiable critic $V_{\mathbf{w}} : \mathcal{S} \times \Theta \to \mathcal{R}$ with parameters $\mathbf{w}$; deterministic or stochastic actor $\pi_\theta$ with parameters $\theta$; empty replay buffer $D$
**Output** : Learned $V_{\mathbf{w}} \approx V(s, \theta)$, learned $\pi_\theta \approx \pi_{\theta^*}$
Initialize critic and actor weights $\mathbf{w}, \theta$
**repeat**:
    Observe state s, take action $a = \pi_\theta(s)$, observe reward $r$ and next state $s'$
    Store $(s, \theta, r, s')$ in the replay buffer $D$
    **if** it's time to update **then**:
        **for** many steps **do**:
            Sample a batch $B_1 = \{(s, \tilde{\theta}, r, s')\}$ from $D$
            Update critic by stochastic gradient descent:
            $\nabla_{\mathbf{w}} \frac{1}{|B_1|} \mathbb{E}_{(s, \tilde{\theta}, r, s') \in B_1} [V_{\mathbf{w}}(s, \tilde{\theta}) - (r + \gamma V_{\mathbf{w}}(s', \tilde{\theta}))]^2$
        **end for**
        **for** many steps **do**:
            Sample a batch $B_2 = \{(s)\}$ from $D$
            Update actor by stochastic gradient ascent: $\nabla_\theta \frac{1}{|B_2|} \mathbb{E}_{s \in B_2} [V_{\mathbf{w}}(s, \theta)]$
        **end for**
    **end if**
**until** convergence

---

---

**Algorithm 3** Stochastic actor-critic with TD prediction for $Q(s, a, \theta)$

---

**Input**: Differentiable critic $Q_{\mathbf{w}} : \mathcal{S} \times \mathcal{A} \times \Theta \rightarrow \mathcal{R}$ with parameters $\mathbf{w}$; stochastic differentiable actor $\pi_\theta$ with parameters $\theta$; empty replay buffer $D$

**Output** : Learned $Q_{\mathbf{w}} \approx Q(s, a, \theta)$, learned $\pi_\theta \approx \pi_{\theta^*}$

Initialize critic and actor weights $\mathbf{w}, \theta$

**repeat**:

    Observe state s, take action $a = \pi_\theta(s)$, observe reward $r$ and next state $s'$

    Store $(s, a, \theta, r, s')$ in the replay buffer $D$

    **if** it's time to update **then**:

        **for** many steps **do**:

            Sample a batch $B_1 = \{(s, a, \tilde{\theta}, r, s')\}$ from $D$

            Update critic by stochastic gradient descent:

            $\nabla_{\mathbf{w}} \frac{1}{|B_1|} \mathbb{E}_{(s,a,\tilde{\theta},r,s') \in B_1} [Q_{\mathbf{w}}(s, a, \tilde{\theta}) - (r + \gamma Q_{\mathbf{w}}(s', a' \sim \pi_{\tilde{\theta}}(s'), \tilde{\theta}))]^2$

        **end for**

        **for** many steps **do**:

            Sample a batch $B_2 = \{(s, a, \tilde{\theta})\}$ from $D$

            Update actor by stochastic gradient ascent:

            $\frac{1}{|B_2|} \mathbb{E}_{(s,a,\tilde{\theta}) \in B_2} \left[ \frac{\pi_\theta(a|s)}{\pi_{\tilde{\theta}}(a|s)} (Q(s, a, \theta) \nabla_\theta \log \pi_\theta(a|s) + \nabla_\theta Q(s, a, \theta)) \right]$

        **end for**

    **end if**

**until** convergence

---

**Algorithm 4** Deterministic actor-critic with TD prediction for $Q(s, a, \theta)$

---

**Input**: Differentiable critic $Q_{\mathbf{w}} : \mathcal{S} \times \mathcal{A} \times \Theta \rightarrow \mathcal{R}$ with parameters $\mathbf{w}$; differentiable deterministic actor $\pi_\theta$ with parameters $\theta$; empty replay buffer $D$

**Output** : Learned $Q_{\mathbf{w}} \approx Q(s, a, \theta)$, learned $\pi_\theta \approx \pi_{\theta^*}$

Initialize critic and actor weights $\mathbf{w}, \theta$

**repeat**:

    Observe state s, take action $a = \pi_\theta(s)$, observe reward $r$ and next state $s'$

    Store $(s, a, \theta, r, s')$ in the replay buffer $D$

    **if** it's time to update **then**:

        **for** many steps **do**:

            Sample a batch $B_1 = \{(s, a, \tilde{\theta}, r, s')\}$ from $D$

            Update critic by stochastic gradient descent:

            $\nabla_{\mathbf{w}} \frac{1}{|B_1|} \mathbb{E}_{(s,a,\tilde{\theta},r,s') \in B_1} [Q_{\mathbf{w}}(s, a, \tilde{\theta}) - (r + \gamma Q_{\mathbf{w}}(s', \pi_{\tilde{\theta}}(s'), \tilde{\theta}))]^2$

        **end for**

        **for** many steps **do**:

            Sample a batch $B_2 = \{(s)\}$ from $D$

            Update actor by stochastic gradient ascent:

            $\frac{1}{|B_2|} \mathbb{E}_{s \in B_2} [\nabla_\theta \pi_\theta(s) \nabla_a Q_{\mathbf{w}}(s, a, \theta)|_{a=\pi_\theta(s)} + \nabla_\theta Q_{\mathbf{w}}(s, a, \theta)|_{a=\pi_\theta(s)}]$

        **end for**

    **end if**

**until** convergence

---

## A.4 EXPERIMENTAL DETAILS

### A.4.1 LQR

For our visualization experiment, we employ an instance of the Linear Quadratic Regulator. Here, the agent observes a 1-D state, corresponding to its position and chooses a 1-D action. The transitions are $s' = s + a$ and there is a quadratic negative term for the reward: $R(s, a) = -s^2 - a^2$. The agent starts in state $s_0 = 1$ and acts in the environment for 50 time steps. The state space is bounded in [-2,2]. The goal of the agent is to reach and remain in the origin. The agent is expected to perform small steps towards the origin when it uses the optimal policy. For this task, we use a

deterministic policy without tanh nonlinearity and we do not use observation normalization. Below additional details and plots for different algorithms.

**PSSVF**    We use a learning rate of $1e-3$ for the policy and $1e-2$ for the PSSVF. Weights are perturbed every episode using $\sigma = 0.5$. The policy is initialized with weight $3.2$ and bias $-3.5$. All the other hyperparameters are set to their default. The true episodic $J(\theta)$ is computed by running 10,000 policies in the environment with parameters in $[-5, 5] \times [-5, 5]$. $V_w(\theta)$ is computed by measuring the output of the PSSVF on the same set of policies. Each red arrow in figure 1 represents 200 update steps of the policy.

**PSVF ad PAVF**    Using the exact same setting, we run PSVF and PAVF in LQR environment and we compare learned $V(s_0, \theta)$ and $Q(s_0, \pi_\theta(s_0), \theta)$ with the true PSVF and PAVF over the parameter space. Computing the value of the true PSVF and PAVF requires computing the infinite sum of discounted reward obtained by the policy. Here we approximate it by running 10,000 policies in the environment with parameters in $[-5, -5] \times [-5, 5]$ for 500 time steps. This, setting $\gamma = 0.99$, provides a good approximation of their true values, since further steps in the environment result in almost zero discounted reward from $s_0$. We use a learning rate of $1e-2$ for the policy and $1e-1$ for the PSVF and PAVF. Weights are perturbed every episode using $\sigma = 0.5$. The policy is updated every 10 time steps using 2 gradient steps; the PSVF and PAVF are updated every 10 time steps using 10 gradient updates. The critic is a 1-layer MLP with 64 neurons and tanh nonlinearity.

In Figures 9 and 10 we report $J(\theta)$, the cumulative discounted reward that an agent would obtain by acting in the environment for infinite time steps using policy $\pi_\theta$ and the cumulative return predicted by the PSVF and PAVF for two different times during learning. Like in the PSSVF experiment, the critic is able improve its predictions over the parameter space. Since in the plots $V(s, \theta)$ and $Q(s, \pi_\theta(s), \theta)$ are evaluated only in $s_0$, the results show that PBVFs are able to effectively bootstrap the values of future states. Each red arrow in Figures 9 and 10 represents 50 update steps of the policy.

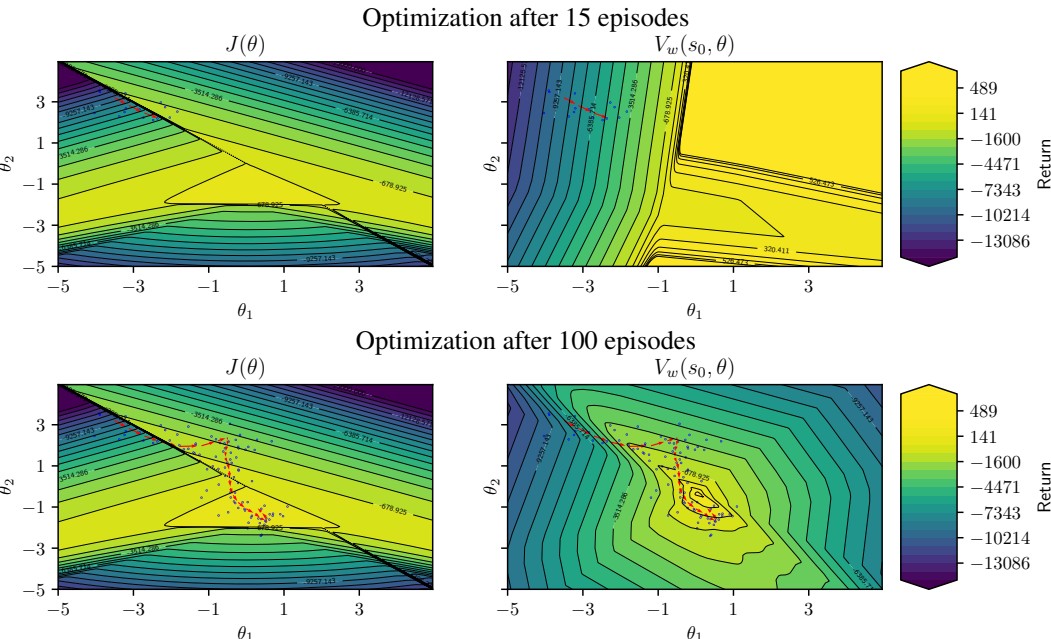

Figure 9: True cumulative discounted reward $J(\theta)$ and PSVF estimation $V_w(s_0, \theta)$ as a function of the policy parameters at two different stages in training. The red arrows represent an optimization trajectory in parameter space. The blue dots represent the perturbed policies used to train $V_w(s_0, \theta)$.

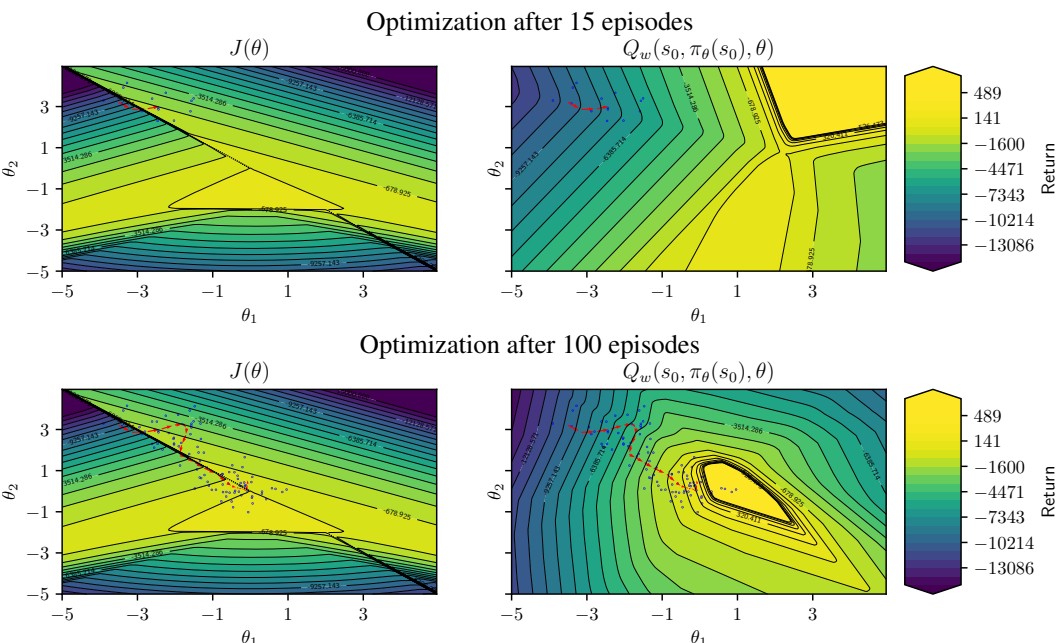

Figure 10: True cumulative discounted reward $J(\theta)$ and PAVF estimation $Q_w(s_0, \pi_\theta(s_0), \theta)$ as a function of the policy parameters at two different stages in training. The red arrows represent an optimization trajectory in parameter space. The blue dots represent the perturbed policies used to train $Q_w(s_0, \pi_\theta(s_0), \theta)$.

### A.4.2 OFFLINE EXPERIMENTS

**Zero-shot learning**  We evaluate the performance of the policies learned from scratch evaluating them with 5 test trajectories every 5 gradient steps. In addition to the results in the main paper, we report in Figures 11 and 12 a comparison of zero-shot performance between PSSVF, PSVF and PAVF in three different environments using deterministic shallow and deep policies (2-layers MLP(64,64)). In this task we use the same hyperparameters found in tables 4, 6 and 8. One additional hyperparameter needs to be considered: the learning rate of the policies trained from scratch. In Figure 3 of the main paper, we use a tuned learning rate of 0.02 that we found working particularly well for PSSVF in the Swimmer environment. In the additional experiments in Figures 11 and 12, we use a learning rate of 0.05 that we found working well across all policies, environments and algorithms when learning zero-shot.

We observe that, using shallow policies, PBVFs can effectively zero-shot learn policies with performance comparable to the policy learned in the environment without additional tuning for the learning rate. We note the regular presence of a spike in performance followed by a decline due to the policy going to regions of the parameter space never observed. This suggests that there is a trade-off between exploiting the generalization of the critic and remaining in the part of the parameter space where the critic is accurate. Measuring the width of these spikes can be useful for determining the number of offline gradient steps to perform in the general algorithm. When using deep policies the results become much worse and zero-shot learned policies can recover the performance of the main policy being learned only in simple environments and at beginning of training (eg. MountainCarContinuous). We observe that, when the critic is trained (last column), the replay buffer contains policies that are very distant to policies randomly initialized. This might explain why the zero-shot performance is better sometimes at the beginning of training (eg. second column). However, since PBVFs in practice perform mostly local off-policy evaluation around the learned policy, this problem is less prone to arise in our main experiments.

**Offline learning with fragmented behaviors**  In this task, data are generated by perturbing a randomly initialized deterministic policy every 200 time steps and using it to act in the environment.

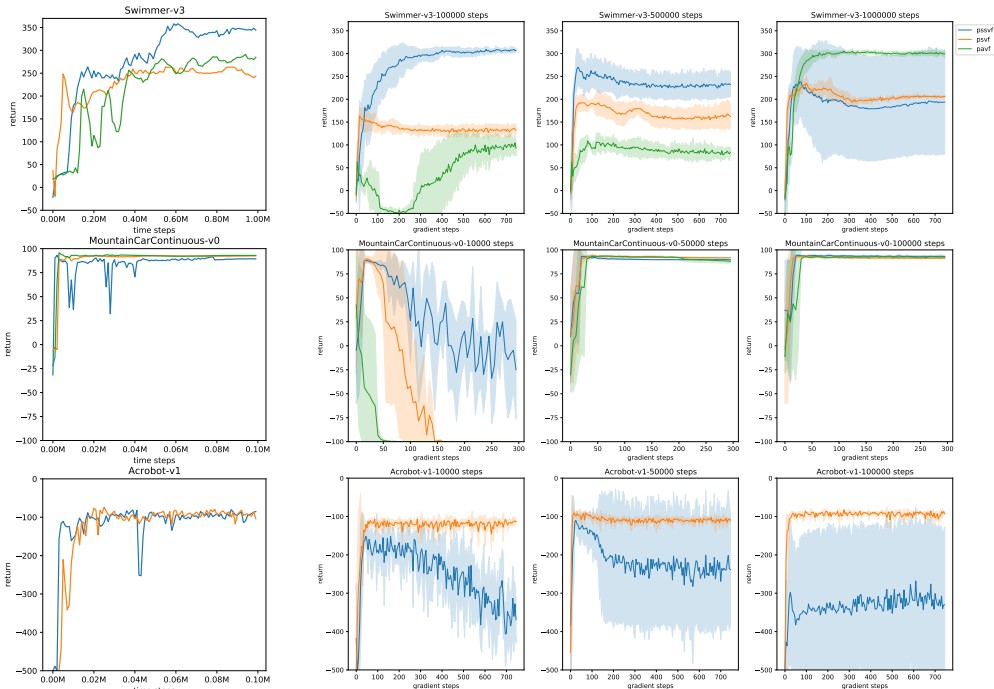

Figure 11: Shallow policies learned from scratch during training. The plots in the left column represent the return of agents learning while interacting with the environment using different algorithms. The learning curves in the other plots represent the return obtained by policies trained from scratch following the fixed critics after different time steps of training. Zero-shot learning curves are averaged over 5 seeds.

We use $\sigma = 0.5$ for the perturbations. After the dataset is collected, the PSVF is trained using a learning rate of $1e - 3$ with a batch size of $128$. When the policy is learned, we use a learning rate of $0.02$. All other hyperparameters are set to default values.

### A.4.3   FULL EXPERIMENTAL RESULTS

**Methodology**   In order to ensure a fair comparison of our methods and the baselines, we adopt the following procedure. For each hyperparameter configuration, for each environment and policy architecture, we run 5 instances of the learning algorithm using different seeds. We measure the learning progress by running 100 evaluations while learning the deterministic policy (without action or parameter noise) using 10 test trajectories. We use two metrics to determine the best hyperparameters: the average return over policy evaluations during the whole training process and the average return over policy evaluations during the last 20% time steps. For each algorithm, environment and policy architecture, we choose the two hyperparameter configurations maximizing the performance of the two metrics and test them on 20 new seeds, reporting average and final performance in table 2 and 3 respectively.

Figures 13 and 14 report all the learning curves from the main paper and for a small non linear policy with 32 hidden neurons.

**Stochastic policies**   We include some results for stochastic policies when using PSSVF and PSVF. Figures 15 and 16 show a comparison with the baselines when using shallow and deep policies respectively. We observe results sometimes comparable, but often inferior with respect to deterministic policies. In particular, when using shallow policies, PBVFs are able to outperform the baselines in the MountainCar environment, while obtaining comparable performance in CartPole and Inverted-Pendulum. Like in previous experiments, PBVFs fail to learn a good policy in Reacher. When using deep policies, the results are slightly different: PBVFs outperform ARS and DDPG in Swimmer, but fail to learn InvertedPendulum. Although the use of stochastic policies can help smoothing the

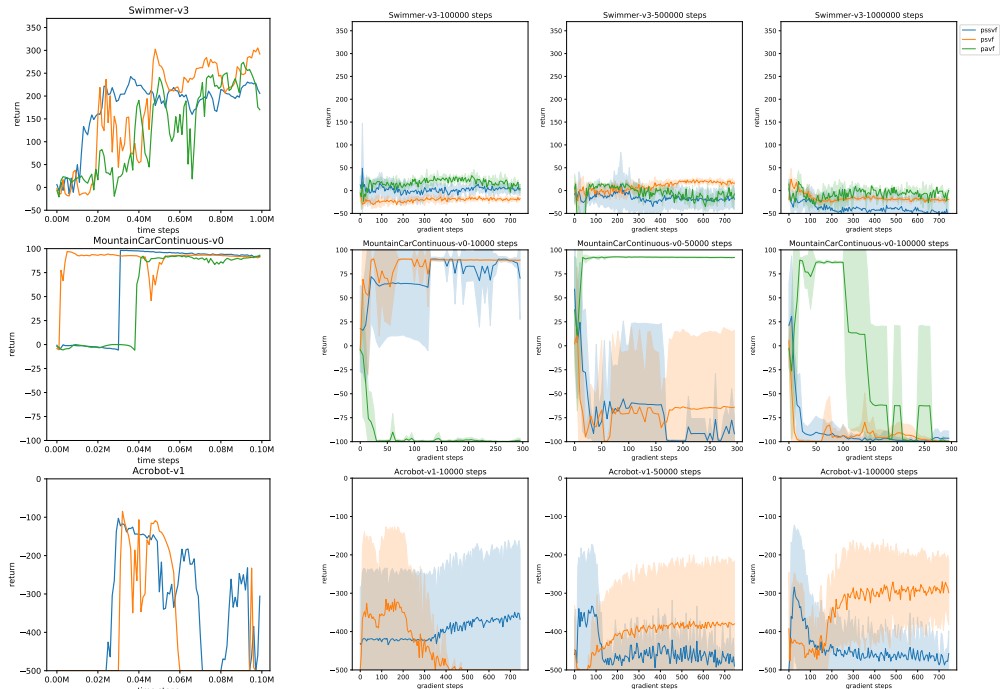

Figure 12: Deep policies learned from scratch during training. The plots in the left column represent the return of agents learning while interacting with the environment using different algorithms. The learning curves in the other plots represent the return obtained by policies trained from scratch following the fixed critics after different time steps of training. Zero-shot learning curves are averaged over 5 seeds.

objective function and allows the agent exploring in action space, we believe that the lower variance provided by deterministic policies can facilitate learning PBVFs.

### A.4.4 SENSITIVITY ANALYSIS

In the following, we report the sensitivity plots for all algorithms, for all deterministic policy architectures and environments. In particular, figure 17, 18, 19, 20 and 21 show the performance of each algorithm given different hyperparameters tried during training. We observe that in general deep policies are more sensitive and, apart for DDPG, achieve often a better performance than smaller policies. The higher sensitivity displayed by ARS is in part caused by the higher number of hyperparameters we tried when tuning the algorithm.

### A.4.5 TABLE OF BEST HYPERPARAMETERS

We report for each algorithm, environment, and policy architecture the best hyperparameters found when optimizing for average return or final return in tables 4, 5, 6, 7, 8 and 9.

Table 2: Average return with standard deviation (across 20 seeds) for hypermarameters optimizing the average return during training using deterministic policies. Square brackets represent the number of neurons per layer of the policy. [] represents a linear policy.

| Policy: [] | MountainCar Continuous-v0 | Inverted Pendulum-v2 | Reacher -v2 | Swimmer -v3 | Hopper -v3 |
|---|---|---|---|---|---|
| ARS | $63 \pm 6$ | $886 \pm 72$ | $-9.2 \pm 0.3$ | $228 \pm 89$ | $1184 \pm 345$ |
| PSSVF | $85 \pm 4$ | $944 \pm 33$ | $-11.7 \pm 0.9$ | $259 \pm 47$ | $1392 \pm 287$ |
| DDPG | $0 \pm 0$ | $612 \pm 169$ | $-8.6 \pm 0.9$ | $95 \pm 112$ | $629 \pm 145$ |
| PSVF | $84 \pm 20$ | $926 \pm 34$ | $-19.7 \pm 6.0$ | $188 \pm 71$ | $917 \pm 249$ |
| PAVF | $82 \pm 21$ | $913 \pm 40$ | $-17.0 \pm 7.7$ | $231 \pm 56$ | $814 \pm 223$ |
| **Policy:[32]** | | | | | |
| ARS | $37 \pm 11$ | $851 \pm 46$ | $-9.6 \pm 0.3$ | $139 \pm 78$ | $1003 \pm 66$ |
| PSSVF | $60 \pm 33$ | $701 \pm 138$ | $10.4 \pm 0.5$ | $189 \pm 35$ | $707 \pm 668$ |
| DDPG | $0 \pm 0$ | $816 \pm 36$ | $-5.7 \pm 0.3$ | $61 \pm 32$ | $1384 \pm 125$ |
| PSVF | $71 \pm 25$ | $529 \pm 281$ | $-11.9 \pm 1.2$ | $226 \pm 33$ | $864 \pm 272$ |
| PAVF | $71 \pm 27$ | $563 \pm 228$ | $-10.9 \pm 1.1$ | $222 \pm 28$ | $793 \pm 322$ |
| **Policy: [64,64]** | | | | | |
| ARS | $28 \pm 8$ | $812 \pm 239$ | $-9.8 \pm 0.3$ | $129 \pm 68$ | $964 \pm 47$ |
| PSSVF | $72 \pm 22$ | $850 \pm 93$ | $-10.7 \pm 0.2$ | $158 \pm 59$ | $922 \pm 568$ |
| DDPG | $0 \pm 0$ | $834 \pm 36$ | $-5.5 \pm 0.4$ | $92 \pm 117$ | $767 \pm 627$ |
| PSVF | $80 \pm 9$ | $580 \pm 107$ | $-10.7 \pm 0.6$ | $137 \pm 38$ | $843 \pm 282$ |
| PAVF | $73 \pm 10$ | $399 \pm 219$ | $-10.7 \pm 0.5$ | $142 \pm 26$ | $875 \pm 136$ |

| Policy: [] | Acrobot-v1 | CartPole-v1 |
|---|---|---|
| ARS | $-161 \pm 23$ | $476 \pm 13$ |
| PSSVF | $-137 \pm 14$ | $443 \pm 105$ |
| PSVF | $-148 \pm 25$ | $459 \pm 28$ |
| **Policy:[32]** | | |
| ARS | $-296 \pm 38$ | $395 \pm 141$ |
| PSSVF | $-251 \pm 80$ | $463 \pm 18$ |
| PSVF | $-270 \pm 113$ | $413 \pm 61$ |
| **Policy: [64,64]** | | |
| ARS | $-335 \pm 35$ | $416 \pm 105$ |
| PSSVF | $-281 \pm 117$ | $452 \pm 34$ |
| PSVF | $-397 \pm 71$ | $394 \pm 71$ |

Table 3: Final return with standard deviation (across 20 seeds) for hypermarameters optimizing the final return during training using deterministic policies.

| Policy: [] | MountainCar Continuous-v0 | Inverted Pendulum-v2 | Reacher -v2 | Swimmer -v3 | Hopper -v3 |
|---|---|---|---|---|---|
| ARS | $73 \pm 5$ | $657 \pm 477$ | $-8.6 \pm 0.5$ | $334 \pm 34$ | $1443 \pm 713$ |
| PSSVF | $84 \pm 28$ | $970 \pm 126$ | $-10.0 \pm 1.0$ | $350 \pm 8$ | $1560 \pm 911$ |
| DDPG | $0 \pm 1$ | $777 \pm 320$ | $-7.3 \pm 0.4$ | $146 \pm 152$ | $704 \pm 234$ |
| PSVF | $76 \pm 36$ | $906 \pm 289$ | $-16.5 \pm 1.6$ | $238 \pm 107$ | $1067 \pm 340$ |
| PAVF | $68 \pm 42$ | $950 \pm 223$ | $-17.2 \pm 15.4$ | $298 \pm 40$ | $720 \pm 281$ |
| **Policy:[32]** | | | | | |
| ARS | $54 \pm 20$ | $936 \pm 146$ | $-9.2 \pm 0.4$ | $239 \pm 117$ | $1048 \pm 68$ |
| PSSVF | $89 \pm 22$ | $816 \pm 234$ | $-10.2 \pm 1.0$ | $294 \pm 41$ | $1204 \pm 615$ |
| DDPG | $0 \pm 0$ | $703 \pm 283$ | $-4.6 \pm 0.6$ | $179 \pm 150$ | $1290 \pm 348$ |
| PSVF | $84 \pm 31$ | $493 \pm 462$ | $-11.3 \pm 0.8$ | $290 \pm 70$ | $1003 \pm 572$ |
| PAVF | $92 \pm 7$ | $854 \pm 295$ | $-10.1 \pm 0.9$ | $307 \pm 34$ | $967 \pm 411$ |
| **Policy: [64,64]** | | | | | |
| ARS | $11 \pm 30$ | $976 \pm 83$ | $-9.4 \pm 0.4$ | $157 \pm 54$ | $1006 \pm 47$ |
| PSSVF | $91 \pm 16$ | $898 \pm 227$ | $-10.7 \pm 0.6$ | $224 \pm 99$ | $1412 \pm 691$ |
| DDPG | $0 \pm 0$ | $943 \pm 73$ | $-4.4 \pm 0.4$ | $196 \pm 151$ | $1437 \pm 752$ |
| PSVF | $93 \pm 1$ | $1000 \pm 0$ | $-10.6 \pm 1.0$ | $257 \pm 26$ | $1247 \pm 344$ |
| PAVF | $93 \pm 2$ | $827 \pm 267$ | $-10.6 \pm 0.4$ | $232 \pm 42$ | $1005 \pm 155$ |

| Policy: [] | Acrobot-v1 | CartPole-v1 |
|---|---|---|
| ARS | $-126 \pm 26$ | $499 \pm 2$ |
| PSSVF | $-97 \pm 6$ | $482 \pm 53$ |
| PSVF | $-100 \pm 18$ | $500 \pm 0$ |
| **Policy:[32]** | | |
| ARS | $-215 \pm 97$ | $471 \pm 110$ |
| PSSVF | $-116 \pm 33$ | $500 \pm 0$ |
| PSVF | $-244 \pm 151$ | $488 \pm 36$ |
| **Policy: [64,64]** | | |
| ARS | $-182 \pm 45$ | $492 \pm 18$ |
| PSSVF | $-233 \pm 139$ | $500 \pm 0$ |
| PSVF | $-406 \pm 51$ | $499 \pm 2$ |

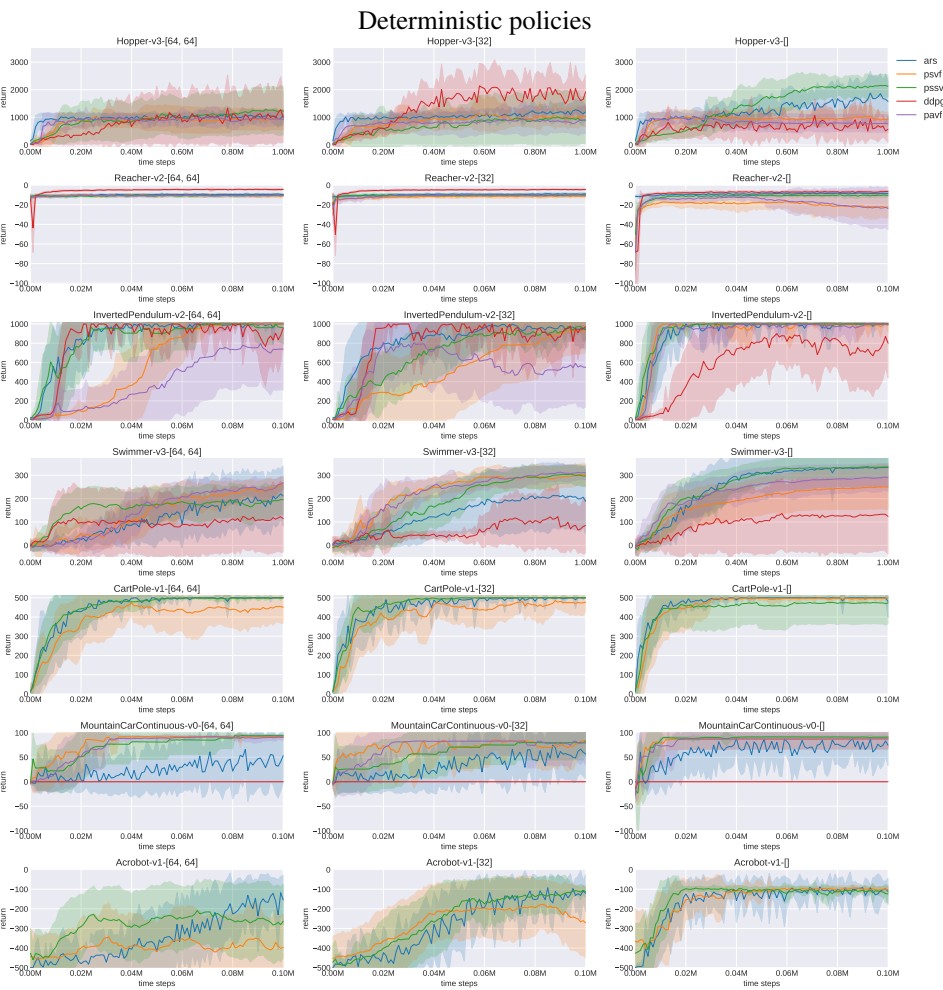

Figure 13: Learning curves representing the average return as a function of the number of time steps in the environment (across 20 runs) with different environments and deterministic policy architectures. We use the **best hyperparameters found while maximizing the average reward** for each task. For each subplot, the square brackets represent the number of neurons per policy layer. [] represents a linear policy.

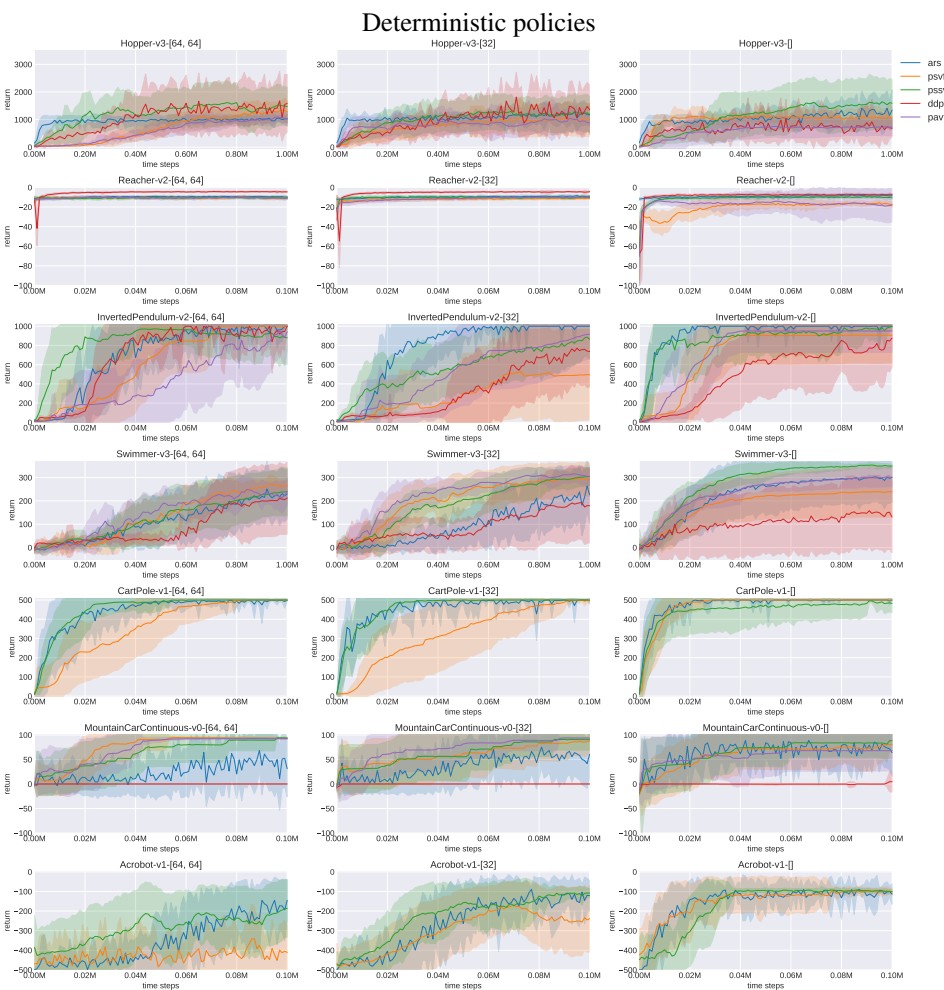

Figure 14: Learning curves representing the average return as a function of the number of time steps in the environment (across 20 runs) with different environments and deterministic policy architectures. We use the **best hyperparameters found while maximizing the final reward for each task**. For each subplot, the square brackets represent the number of neurons per policy layer. [] represents a linear policy.

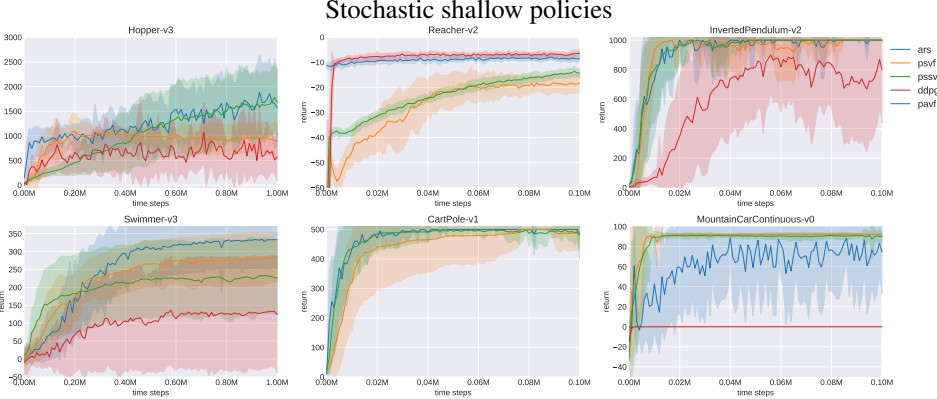

Figure 15: Learning curves representing the average return as a function of the number of time steps in the environment (across 20 runs) with different environments using stochastic shallow policies. We use the **best hyperparameters found while maximizing the average reward for each task.**

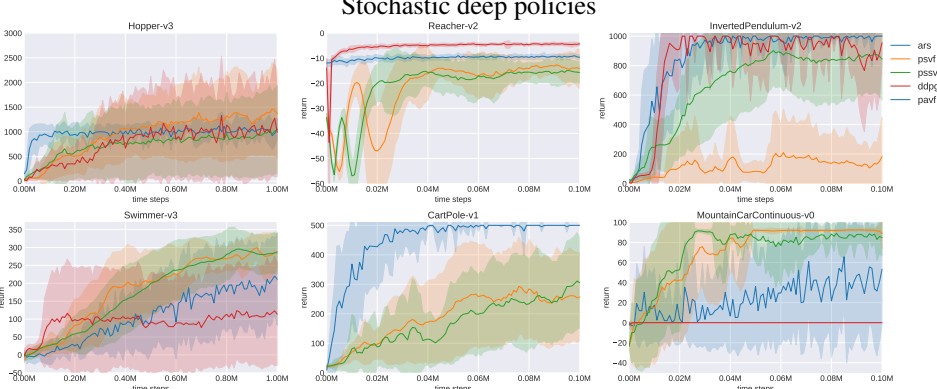

Figure 16: Learning curves representing the average return as a function of the number of time steps in the environment (across 20 runs) with different environments using stochastic deep policies ([64,64]). We use the **best hyperparameters found while maximizing the average reward for each task.**

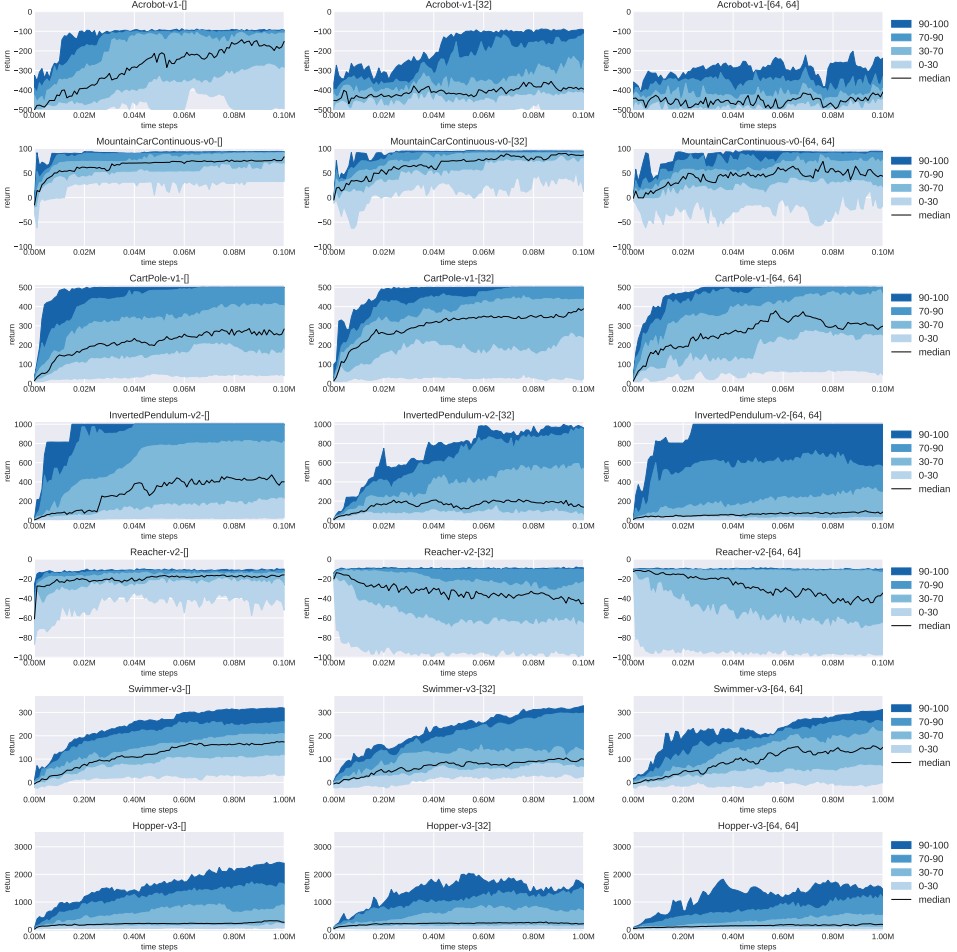

Figure 17: **Sensitivity of PSSVFs using deterministic policies** to the choice of the hyperparameter. Performance is shown by percentile using all the learning curves obtained during hyperparameter tuning. The median performance is depicted as a dark line. For each subplot, the numbers in the square brackets represent the number of neurons per layer of the policy. [] represents a linear policy.

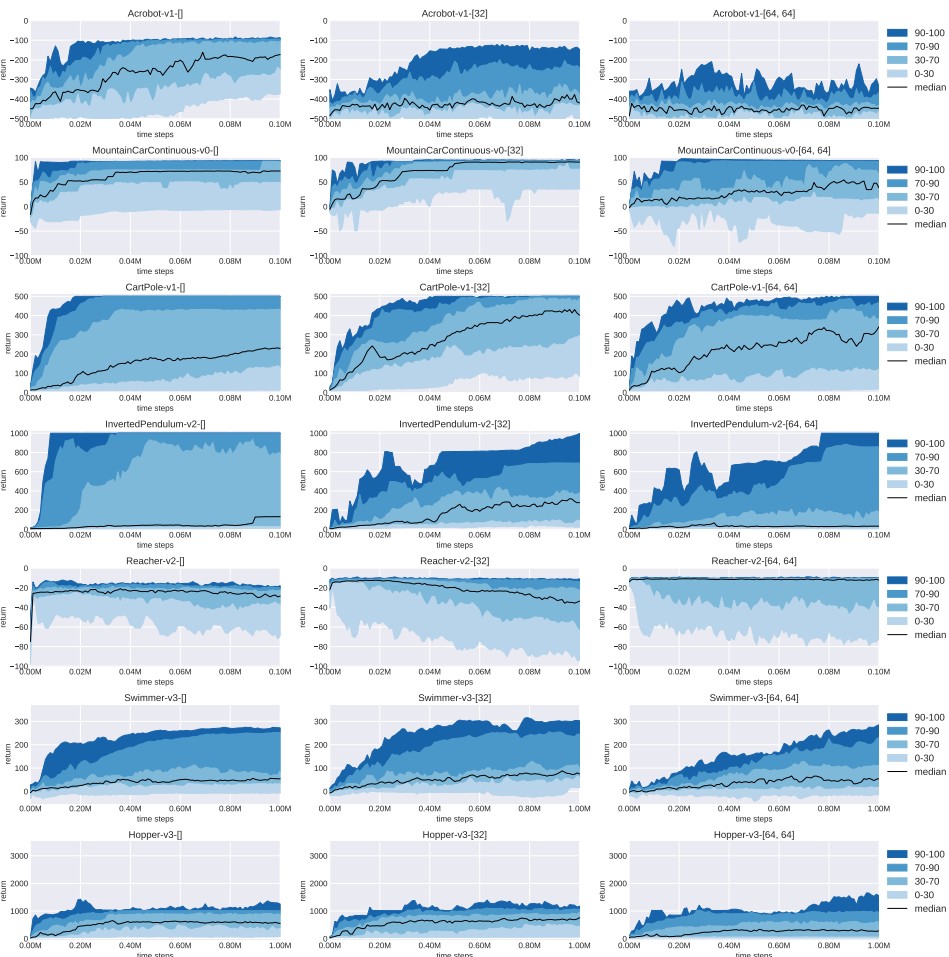

Figure 18: **Sensitivity of PSVFs using deterministic policies** to the choice of the hyperparameter. Performance is shown by percentile using all the learning curves obtained during hyperparameter tuning. The median performance is depicted as a dark line. For each subplot, the numbers in the square brackets represent the number of neurons per layer of the policy. [] represents a linear policy.

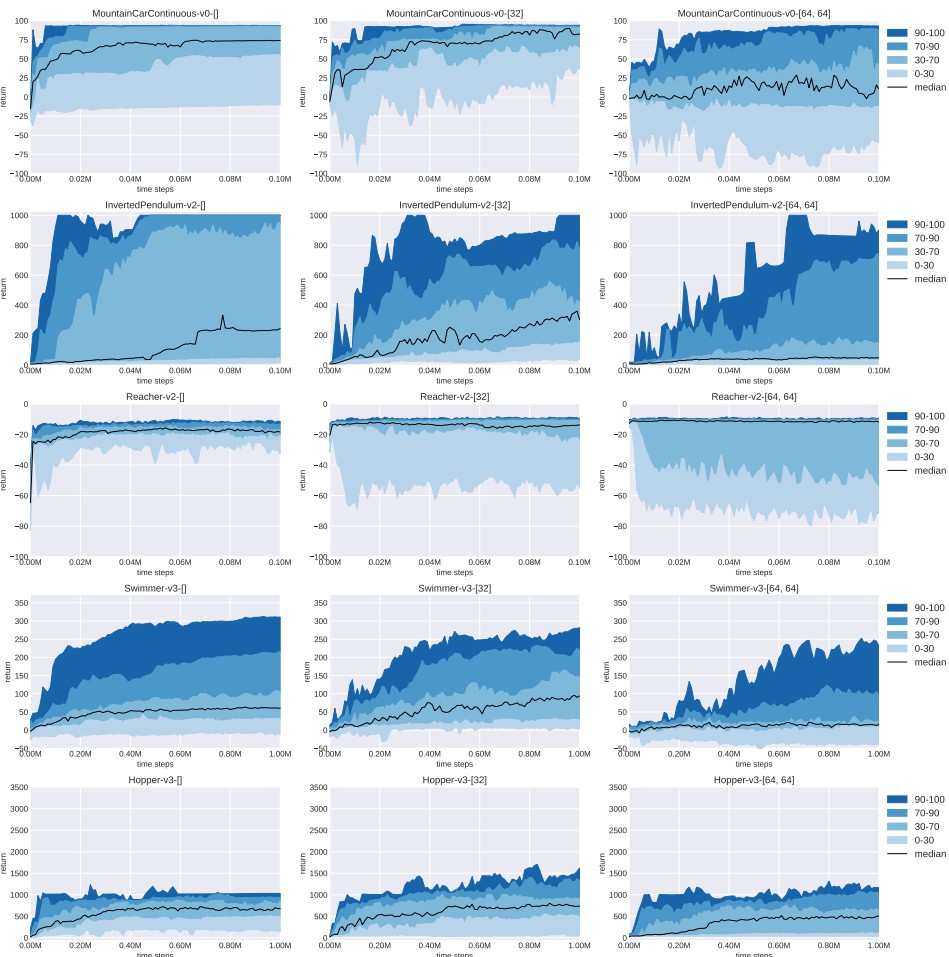

Figure 19: **Sensitivity of PAVFs using deterministic policies** to the choice of the hyperparameter. Performance is shown by percentile using all the learning curves obtained during hyperparameter tuning. The median performance is depicted as a dark line. For each subplot, the numbers in the square brackets represent the number of neurons per layer of the policy.

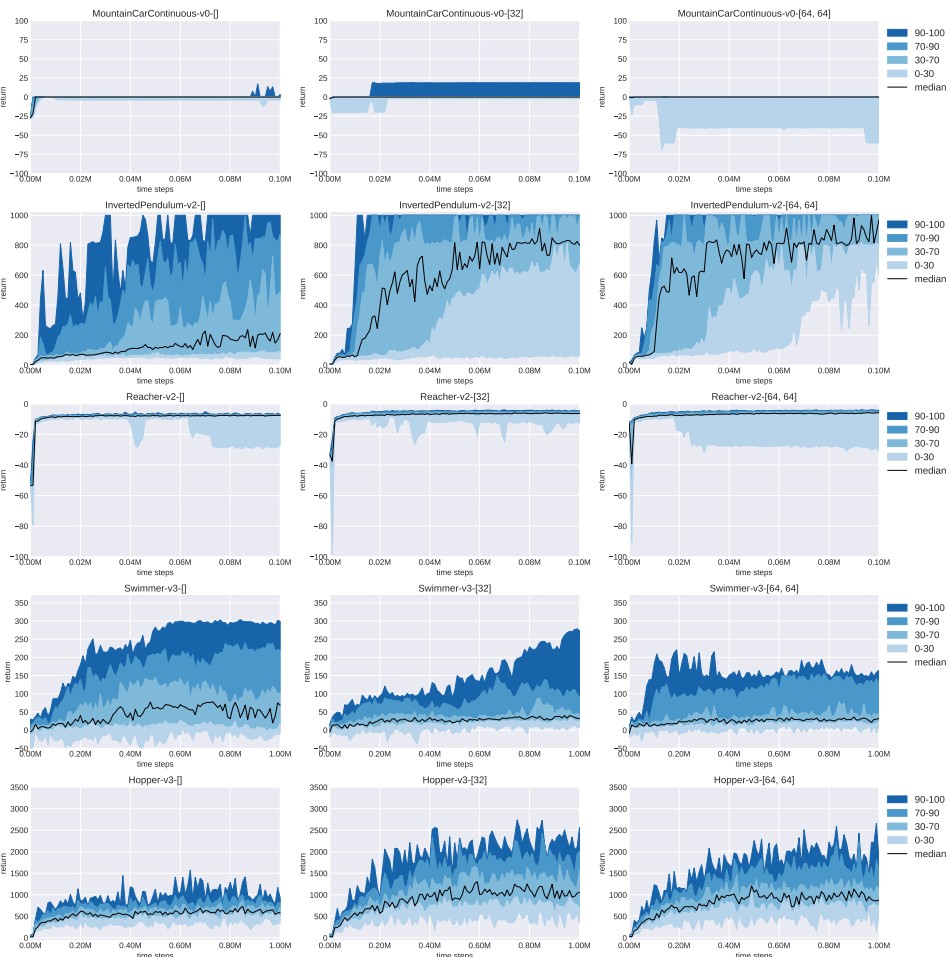

Figure 20: **Sensitivity of DDPG** to the choice of the hyperparameter. Performance is shown by percentile using all the learning curves obtained during hyperparameter tuning. The median performance is depicted as a dark line. For each subplot, the numbers in the square brackets represent the number of neurons per layer of the policy.

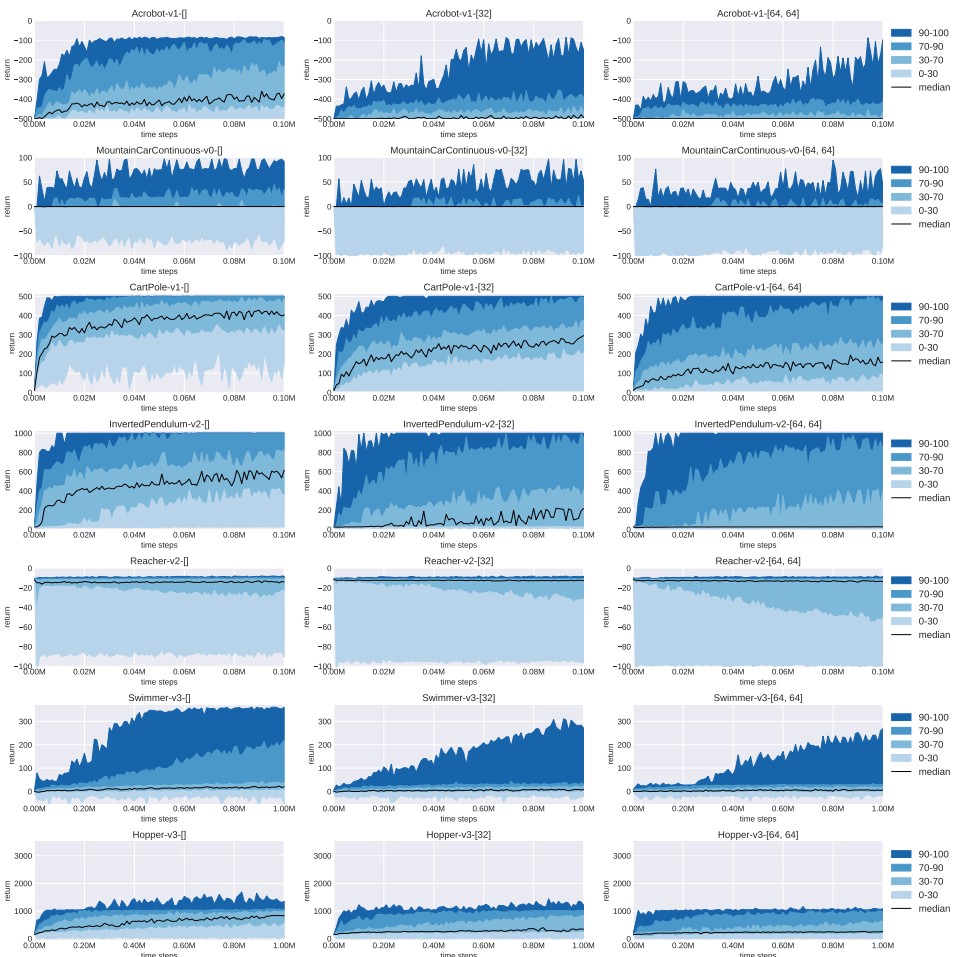

Figure 21: **Sensitivity of ARS** to the choice of the hyperparameter. Performance is shown by percentile using all the learning curves obtained during hyperparameter tuning. The median performance is depicted as a dark line. For each subplot, the numbers in the square brackets represent the number of neurons per layer of the policy.

Table 4: Table of best hyperparameters for PSSVFs using deterministic policies

| Learning rate policy | Policy: | [] | | [32] | | [64,64] | |
|---|---|---|---|---|---|---|---|
| | Metric: | avg | last | avg | last | avg | last |
| Acrobot-v1 | | 1e-2 | 1e-3 | 1e-4 | 1e-4 | 1e-4 | 1e-4 |
| MountainCarContinuous-v0 | | 1e-2 | 1e-3 | 1e-4 | 1e-4 | 1e-4 | 1e-4 |
| CartPole-v1 | | 1e-3 | 1e-3 | 1e-3 | 1e-3 | 1e-4 | 1e-4 |
| Swimmer-v3 | | 1e-3 | 1e-3 | 1e-3 | 1e-3 | 1e-2 | 1e-4 |
| InvertedPendulum-v2 | | 1e-3 | 1e-3 | 1e-3 | 1e-3 | 1e-4 | 1e-4 |
| Reacher-v2 | | 1e-4 | 1e-4 | 1e-4 | 1e-4 | 1e-4 | 1e-4 |
| Hopper-v3 | | 1e-4 | 1e-4 | 1e-4 | 1e-3 | 1e-4 | 1e-4 |
| **Learning rate critic** | | | | | | | |
| Acrobot-v1 | | 1e-2 | 1e-3 | 1e-2 | 1e-2 | 1e-2 | 1e-2 |
| MountainCarContinuous-v0 | | 1e-3 | 1e-2 | 1e-3 | 1e-2 | 1e-2 | 1e-2 |
| CartPole-v1 | | 1e-2 | 1e-2 | 1e-3 | 1e-3 | 1e-2 | 1e-2 |
| Swimmer-v3 | | 1e-3 | 1e-3 | 1e-2 | 1e-2 | 1e-3 | 1e-2 |
| InvertedPendulum-v2 | | 1e-2 | 1e-2 | 1e-3 | 1e-2 | 1e-3 | 1e-3 |
| Reacher-v2 | | 1e-3 | 1e-3 | 1e-3 | 1e-3 | 1e-4 | 1e-4 |
| Hopper-v3 | | 1e-3 | 1e-3 | 1e-2 | 1e-2 | 1e-2 | 1e-2 |
| **Noise for exploration** | | | | | | | |
| Acrobot-v1 | | 1.0 | 1.0 | 1e-1 | 1e-1 | 1e-1 | 1e-1 |
| MountainCarContinuous-v0 | | 1.0 | 1.0 | 1e-1 | 1e-1 | 1e-1 | 1e-1 |
| CartPole-v1 | | 1.0 | 1.0 | 1.0 | 1.0 | 1e-1 | 1e-1 |
| Swimmer-v3 | | 1.0 | 1.0 | 1.0 | 1.0 | 1.0 | 1e-1 |
| InvertedPendulum-v2 | | 1.0 | 1.0 | 1.0 | 1.0 | 1e-1 | 1e-1 |
| Reacher-v2 | | 1e-1 | 1e-1 | 1e-1 | 1e-1 | 1e-1 | 1e-1 |
| Hopper-v3 | | 1.0 | 1.0 | 1e-1 | 1.0 | 1e-1 | 1e-1 |

Table 5: Table of best hyperparameters for ARS

| Learning rate policy | Policy: | [] | | [32] | | [64,64] | |
|---|---|---|---|---|---|---|---|
| | Metric: | avg | last | avg | last | avg | last |
| Acrobot-v1 | | 1e-2 | 1e-3 | 1e-2 | 1e-2 | 1e-2 | 1e-2 |
| MountainCarContinuous-v0 | | 1e-2 | 1e-2 | 1e-2 | 1e-2 | 1e-2 | 1e-2 |
| CartPole-v1 | | 1e-2 | 1e-2 | 1e-2 | 1e-2 | 1e-2 | 1e-2 |
| Swimmer-v3 | | 1e-2 | 1e-2 | 1e-2 | 1e-2 | 1e-2 | 1e-2 |
| InvertedPendulum-v2 | | 1e-2 | 1e-2 | 1e-2 | 1e-2 | 1e-2 | 1e-2 |
| Reacher-v2 | | 1e-2 | 1e-2 | 1e-3 | 1e-2 | 1e-3 | 1e-3 |
| Hopper-v3 | | 1e-2 | 1e-2 | 1e-2 | 1e-2 | 1e-2 | 1e-2 |
| **Number of directions and elite directions** | | | | | | | |
| Acrobot-v1 | | (4,4) | (4,4) | (1,1) | (1,1) | (1,1) | (1,1) |
| MountainCarContinuous-v0 | | (1,1) | (1,1) | (1,1) | (16,4) | (1,1) | (1,1) |
| CartPole-v1 | | (4,4) | (4,4) | (1,1) | (1,1) | (4,1) | (4,1) |
| Swimmer-v3 | | (1,1) | (1,1) | (1,1) | (4,1) | (1,1) | (1,1) |
| InvertedPendulum-v2 | | (4,4) | (4,4) | (1,1) | (4,4) | (4,1) | (16,1) |
| Reacher-v2 | | (16,16) | (16,16) | (1,1) | (16,4) | (1,1) | (1,1) |
| Hopper-v3 | | (4,1) | (4,1) | (1,1) | (1,1) | (1,1) | (1,1) |
| **Noise for exploration** | | | | | | | |
| Acrobot-v1 | | 1e-2 | 1e-3 | 1e-1 | 1e-1 | 1e-1 | 1e-1 |
| MountainCarContinuous-v0 | | 1e-1 | 1e-1 | 1e-1 | 1e-1 | 1e-1 | 1e-1 |
| CartPole-v1 | | 1e-2 | 1e-2 | 1e-1 | 1e-1 | 1e-2 | 1e-2 |
| Swimmer-v3 | | 1e-1 | 1e-1 | 1e-2 | 1e-1 | 1e-1 | 1e-1 |
| InvertedPendulum-v2 | | 1e-2 | 1e-2 | 1e-2 | 1e-2 | 1e-2 | 1e-2 |
| Reacher-v2 | | 1e-2 | 1e-2 | 1e-2 | 1e-2 | 1e-2 | 1e-2 |
| Hopper-v3 | | 1e-1 | 1e-1 | 1e-1 | 1e-1 | 1e-1 | 1e-1 |

Table 6: Table of best hyperparameters for PSVFs using deterministic policies

| Learning rate policy | Policy: | [] | | [32] | | [64,64] | |
|---|---|---|---|---|---|---|---|
| | Metric: | avg | last | avg | last | avg | last |
| Acrobot-v1 | | 1e-2 | 1e-2 | 1e-4 | 1e-4 | 1e-4 | 1e-2 |
| MountainCarContinuous-v0 | | 1e-2 | 1e-3 | 1e-2 | 1e-4 | 1e-3 | 1e-4 |
| CartPole-v1 | | 1e-2 | 1e-2 | 1e-2 | 1e-4 | 1e-3 | 1e-4 |
| Swimmer-v3 | | 1e-3 | 1e-3 | 1e-3 | 1e-3 | 1e-3 | 1e-3 |
| InvertedPendulum-v2 | | 1e-2 | 1e-3 | 1e-4 | 1e-4 | 1e-4 | 1e-4 |
| Reacher-v2 | | 1e-3 | 1e-2 | 1e-4 | 1e-4 | 1e-4 | 1e-4 |
| Hopper-v3 | | 1e-3 | 1e-3 | 1e-4 | 1e-4 | 1e-4 | 1e-3 |
| **Learning rate critic** | | | | | | | |
| Acrobot-v1 | | 1e-3 | 1e-4 | 1e-2 | 1e-2 | 1e-3 | 1e-2 |
| MountainCarContinuous-v0 | | 1e-4 | 1e-3 | 1e-2 | 1e-4 | 1e-3 | 1e-3 |
| CartPole-v1 | | 1e-2 | 1e-2 | 1e-2 | 1e-3 | 1e-2 | 1e-4 |
| Swimmer-v3 | | 1e-4 | 1e-4 | 1e-4 | 1e-4 | 1e-4 | 1e-4 |
| InvertedPendulum-v2 | | 1e-3 | 1e-2 | 1e-3 | 1e-4 | 1e-4 | 1e-3 |
| Reacher-v2 | | 1e-2 | 1e-2 | 1e-3 | 1e-3 | 1e-4 | 1e-4 |
| Hopper-v3 | | 1e-2 | 1e-2 | 1e-4 | 1e-4 | 1e-2 | 1e-4 |
| **Noise for exploration** | | | | | | | |
| Acrobot-v1 | | 1.0 | 1.0 | 1e-1 | 1e-1 | 1e-1 | 1e-1 |
| MountainCarContinuous-v0 | | 1.0 | 1e-1 | 1e-1 | 1.0 | 1e-1 | 1e-1 |
| CartPole-v1 | | 1.0 | 1.0 | 1.0 | 1e-1 | 1e-1 | 1e-1 |
| Swimmer-v3 | | 1.0 | 1.0 | 1.0 | 1.0 | 1.0 | 1.0 |
| InvertedPendulum-v2 | | 1.0 | 1.0 | 1e-1 | 1e-1 | 1e-1 | 1e-1 |
| Reacher-v2 | | 1.0 | 1.0 | 1.0 | 1.0 | 1e-1 | 1e-1 |
| Hopper-v3 | | 1.0 | 1.0 | 1e-1 | 1e-1 | 1e-1 | 1.0 |

Table 7: Table of best hyperparameters for PSSVFs and PSVFs using stochastic policies

| | Algo: | PSSVF | | PSVF | |
|---|---|---|---|---|---|
| **Learning rate policy** | Policy: | [] | [64,64] | [] | [64,64] |
| | Metric: | avg | avg | avg | avg |
| Acrobot-v1 | | 1e-2 | 1e-2 | 1e-2 | 1e-3 |
| MountainCarContinuous-v0 | | 1e-2 | 1e-3 | 1e-2 | 1e-3 |
| CartPole-v1 | | 1e-3 | 1e-4 | 1e-2 | 1e-3 |
| Swimmer-v3 | | 1e-2 | 1e-4 | 1e-3 | 1e-4 |
| InvertedPendulum-v2 | | 1e-3 | 1e-4 | 1e-2 | 1e-3 |
| Reacher-v2 | | 1e-4 | 1e-3 | 1e-2 | 1e-2 |
| Hopper-v3 | | 1e-4 | 1e-4 | 1e-3 | 1e-4 |
| **Learning rate critic** | | | | | |
| Acrobot-v1 | | 1e-2 | 1e-4 | 1e-4 | 1e-2 |
| MountainCarContinuous-v0 | | 1e-2 | 1e-2 | 1e-3 | 1e-3 |
| CartPole-v1 | | 1e-2 | 1e-3 | 1e-2 | 1e-2 |
| Swimmer-v3 | | 1e-2 | 1e-3 | 1e-3 | 1e-4 |
| InvertedPendulum-v2 | | 1e-3 | 1e-3 | 1e-3 | 1e-2 |
| Reacher-v2 | | 1e-3 | 1e-3 | 1e-3 | 1e-3 |
| Hopper-v3 | | 1e-3 | 1e-2 | 1e-2 | 1e-4 |
| **Noise for exploration** | | | | | |
| Acrobot-v1 | | 1.0 | 1.0 | 1.0 | 1.0 |
| MountainCarContinuous-v0 | | 1.0 | 1e-1 | 1.0 | 1e-1 |
| CartPole-v1 | | 1.0 | 1.0 | 1.0 | 1e-1 |
| Swimmer-v3 | | 1.0 | 1e-1 | 1.0 | 1e-1 |
| InvertedPendulum-v2 | | 1.0 | 1.0 | 1.0 | 1e-1 |
| Reacher-v2 | | 1e-1 | 0.0 | 1.0 | 0.0 |
| Hopper-v3 | | 1.0 | 1e-1 | 1.0 | 1e-1 |

Table 8: Table of best hyperparameters for PAVFs using deterministic policies

| **Learning rate policy** | Policy: | [] | | [32] | | [64,64] | |
|---|---|---|---|---|---|---|---|
| | Metric: | avg | last | avg | last | avg | last |
| MountainCarContinuous-v0 | | 1e-2 | 1e-3 | 1e-3 | 1e-4 | 1e-4 | 1e-4 |
| Swimmer-v3 | | 1e-3 | 1e-3 | 1e-3 | 1e-3 | 1e-3 | 1e-3 |
| InvertedPendulum-v2 | | 1e-2 | 1e-3 | 1e-3 | 1e-4 | 1e-4 | 1e-4 |
| Reacher-v2 | | 1e-3 | 1e-3 | 1e-4 | 1e-4 | 1e-4 | 1e-4 |
| Hopper-v3 | | 1e-3 | 1e-4 | 1e-4 | 1e-4 | 1e-4 | 1e-3 |
| **Learning rate critic** | | | | | | | |
| MountainCarContinuous-v0 | | 1e-4 | 1e-4 | 1e-4 | 1e-3 | 1e-4 | 1e-3 |
| Swimmer-v3 | | 1e-4 | 1e-4 | 1e-4 | 1e-4 | 1e-4 | 1e-4 |
| InvertedPendulum-v2 | | 1e-3 | 1e-2 | 1e-2 | 1e-4 | 1e-2 | 1e-3 |
| Reacher-v2 | | 1e-3 | 1e-3 | 1e-3 | 1e-2 | 1e-3 | 1e-3 |
| Hopper-v3 | | 1e-4 | 1e-3 | 1e-3 | 1e-2 | 1e-4 | 1e-3 |
| **Noise for exploration** | | | | | | | |
| MountainCarContinuous-v0 | | 1.0 | 1e-1 | 1e-1 | 1e-1 | 1e-1 | 1e-1 |
| Swimmer-v3 | | 1.0 | 1.0 | 1.0 | 1.0 | 1.0 | 1.0 |
| InvertedPendulum-v2 | | 1.0 | 1.0 | 1e-1 | 1e-1 | 1e-1 | 1e-1 |
| Reacher-v2 | | 1e-1 | 1e-1 | 1e-1 | 1.0 | 1.0 | 1.0 |
| Hopper-v3 | | 1.0 | 1.0 | 1e-1 | 1e-1 | 1e-1 | 1.0 |

Table 9: Table of best hyperparameters for DDPG

| **Learning rate policy** | Policy: | [] | | [32] | | [64,64] | |
|---|---|---|---|---|---|---|---|
| | Metric: | avg | last | avg | last | avg | last |
| MountainCarContinuous-v0 | | 1e-2 | 1e-2 | 1e-2 | 1e-4 | 1e-3 | 1e-3 |
| Swimmer-v3 | | 1e-3 | 1e-3 | 1e-2 | 1e-2 | 1e-2 | 1e-2 |
| InvertedPendulum-v2 | | 1e-4 | 1e-4 | 1e-3 | 1e-3 | 1e-3 | 1e-4 |
| Reacher-v2 | | 1e-4 | 1e-3 | 1e-2 | 1e-2 | 1e-3 | 1e-3 |
| Hopper-v3 | | 1e-2 | 1e-2 | 1e-2 | 1e-4 | 1e-2 | 1e-2 |
| **Learning rate critic** | | | | | | | |
| MountainCarContinuous-v0 | | 1e-4 | 1e-4 | 1e-4 | 1e-3 | 1e-3 | 1e-3 |
| Swimmer-v3 | | 1e-3 | 1e-3 | 1e-3 | 1e-3 | 1e-2 | 1e-3 |
| InvertedPendulum-v2 | | 1e-3 | 1e-3 | 1e-3 | 1e-4 | 1e-3 | 1e-3 |
| Reacher-v2 | | 1e-3 | 1e-3 | 1e-3 | 1e-3 | 1e-3 | 1e-3 |
| Hopper-v3 | | 1e-3 | 1e-3 | 1e-4 | 1e-4 | 1e-4 | 1e-4 |
| **Noise for exploration** | | | | | | | |
| MountainCarContinuous-v0 | | 1e-2 | 1e-2 | 1e-2 | 1e-1 | 1e-1 | 1e-1 |
| Swimmer-v3 | | 1e-1 | 1e-1 | 1e-2 | 1e-2 | 1e-2 | 1e-1 |
| InvertedPendulum-v2 | | 1e-1 | 1e-1 | 1e-2 | 1e-2 | 1e-2 | 1e-2 |
| Reacher-v2 | | 1e-1 | 1e-2 | 1e-1 | 1e-1 | 1e-1 | 1e-1 |
| Hopper-v3 | | 1e-1 | 1e-1 | 1e-1 | 1e-2 | 1e-1 | 1e-2 |

