# OpenReview forum: "Parameter-Based Value Functions"
_ICLR.cc/2021/Conference — ICLR 2021 Poster_

### Official Review · AnonReviewer4 · 2020-10-25
**Reject for theoretical reasons. (Update: the theoretical issues were cleared up)**

**Rating:** 7
**Confidence:** 4

**Review:**

**Update**

I have updated my score to 7.
One of the points that was not explained in the original paper was that (ignoring function approximation effects) an optimal solution for $J_b$ (the OffPAC objective) will be optimal also for the original off-policy RL objective $J$ (i.e. estimating the on-policy objective in an unbiased manner from off-policy data). From this point of view, I agree that optimizing $J_b$ directly is an interesting question, despite the fact that the exact gradient for $J_b$ may be less similar to the gradient of $J$ compared to the usually used approximate gradient of $J_b$ that drops the $\nabla_\theta Q$ term. It still remains unclear which of the two methods has a theoretical advantage over the other in the function approximation setting (in terms of optimizing for $J$); however, because it is unclear, it is interesting to evaluate the method proposed here and to perform experiments as done in the paper to try to find out which method performs better.

The results were mixed; however, the evaluation is fairly thorough and some potential advantages of the new methods such as generalization in the $\theta$ space and zero-shot learning were explained.

The discussion in the paper is much improved compared to the original version. Also, additional ablation studies such as testing what happens when the $\nabla_\theta Q$ term is dropped were added (when $Q$ includes $\theta$ as an input). Moreover, LQR experiments for $Q(s,a,\theta)$ and $V(s,\theta)$ were added in the appendix (the results here do not give as good a match as the $V(\theta)$ formulation gave, but they are reasonable).

______________________________________________________________
1. Summarize what the paper claims to contribute. Be positive and
generous.

They propose to include the policy parameters as an input
to the value function, so that the value function could generalize
across different policies (there are 2 other concurrent works with a similar
idea, one they have cited and discussed "Policy evaluation networks"
https://arxiv.org/abs/2002.11833, another is submitted to ICLR2021 on
openreview https://openreview.net/forum?id=V4AVDoFtVM).
They put the policy parameters theta as an input to the value function
in 3 cases $V(\theta)$ (PSSVF), $V(s, \theta)$ (PSVF), and $Q(s,a,\theta)$ (PAVF).
They propose new policy gradient theorems for the $V(s,\theta)$ and
$Q(s,a,\theta)$ cases (but I believe these to be theoretically flawed).

They perform experiments testing $V(\theta)$ in 2 cases: 4.1) (sanity check
experiment) visualizing and testing for correctness on an LQR task,
4.3) zero-shot learning: after training a policy pi using the $V(\theta)$ method,
a new policy is reinitialized pi_new and trained from scratch using only the
trained $V(\theta)$ without interacting with the environment. The interesting
bit was that $\pi_{new}$ managed to outperform the learned policy during
data collection $\pi$ (this implies that the $V(\theta)$ function managed to
generalize. It would have been nice to, in addition to $\pi_{new}$, also
see whether $\pi$ could have been improved by just continuing to optimize
it without interacting with the environment, but this was not done).

They tested $V(\theta)$, $V(s,\theta)$ and $Q(s,a,\theta)$ on MuJoCo tasks
compared to augmented random search (this is similar to evolution
strategies) and to deep deterministic policy gradients (DDPG). And
the performance did not change much, and sometimes all the new methods
failed when DDPG worked (on the reacher task).

The final experiment 4.4 was for offline learning with fragmented
behaviors, i.e. they do not observe full episode data for a fixed
theta, which makes it impossible to learn $V(\theta)$ directly, but
$V(s,\theta)$ can be learned by TD methods (also note that the data is
collected from a different behavior policy). Then they test a similar
zero-shot learning procedure as they did for $V(\theta)$ at different
stages of the learning (but as far as I understood, for $V(s,\theta)$ they sampled
data from the replay buffer when training the policy (thus not fully without
interacting with the data). Perhaps the authors can clarify this), and
show that the newly learned policy can outperform the behavior policy,
thus demonstrating the generalizability of the method.


2. List strong and weak points of the paper. Be as comprehensive as possible.

\+ The experiment on zero-shot learning is nice to show that the $V(\theta)$
function can generalize.
\+ The paper is clearly written.
\+ They discuss a lot of related work.
\+ The experimental methodology seemed mostly good and honest, and
was explained in detail in the appendix (some nice points: They include
a sensitivity analysis showing quantiles of the performance.
Also the final best chosen hyperparameters were evaluated with
20 new seeds, separate from the 5 seeds used during hyperparameter tuning).

\- The new policy gradient theorems seemed flawed. Also some discussion
around off-policy learning seemed incomplete.
\- The methods were not shown to experimentally lead to major gains.
\- One of the difficulties with searching in parameter space is how
to deal with large parameter spaces. The two concurrent works considering
$V(\theta)$ proposed solutions to this issue by embedding the policy into
a smaller space. In the current work no solution is proposed. The
experiments on zero-shot learning using $V(\theta)$ were only good with
low-dimensional linear policies.
\- A sanity check experiment on LQR was performed for only $V(\theta)$ (which was
the only one for which the gradient was theoretically sound); it would
have been good to do similar experiments for the other ones.
\- I would expect $V(s,\theta)$ to outperform $V(\theta)$ due to using the
state information, but this did not appear to be the case.


3. Clearly state your recommendation (accept or reject) with one or two
key reasons for this choice.

I recommend rejecting the paper due to the theoretical flaws in the newly
proposed policy gradient theorems using $V(s,\theta)$ and $Q(s,a,\theta)$. Also,
the practical advantages of using $V(s,\theta)$ and $Q(s,a,\theta)$ were not shown.


4. Provide supporting arguments for your recommendation.

The theoretical issues in this paper start in equation 1. They write:
"... we can express the maximization of the expected cumulative reward
in terms of the state-value function:"

$J(\pi_\theta) = \int d^{\pi}(s) V(s) ds,$  (in the paper)

where $d(s)$ is the discounted state visitation distribution. However, this
is not the RL objective. The RL objective would be

$J(\pi_\theta) = \int d^{\pi}(s) R(s) ds.$    (what it should actually be)

The authors probably took their objective from the work by
Degris et al (2012, https://arxiv.org/pdf/1205.4839.pdf);
however, in Degris'12, $d(s)$ is _not_ the discounted
state visitation distribution. It is the limiting distribution as
$t \to \infty$, which is a stationary distribution. When $d^{\pi}(s)$ is stationary,
then the two objectives become equivalent: $d(s)$ does not change
from one time step to the next, so the difference between the objectives
will be just a $1/(1-\gamma)$ constant factor. Putting aside this issue,
probably the limiting distribution formulation is not realistic as most
RL researchers consider the episodic setting, so using a discounted
state visitation distribution is probably better. However, the newly
proposed policy gradient theorems do not appear sound for the true RL
objective using $R(s)$.

Next, they replace the distribution $d^{\pi}(s)$ with a
distribution $d^{\pi_b}(s)$ gathered using a behavioral policy (so they are
working off-policy). However, they do not apply an importance weighting
correction for the distribution shift, and just ignore the importance
weights (Note that this is also done by Silver et al (2014) in deterministic
policy gradients, and by Lillicrap et al (2015) in DDPG, so it is not that
strange per se, as long as it gives better practical performance. However,
it should at least be acknowledged that the importance weights are being
ignored). Note that they still apply an importance weight on the actions
($\pi(a|s)/\pi_b(a|s)$) once the state is sampled from the data buffer, however,
this does not correct for the distribution shift from $d^{\pi}$ to $d^{\pi_b}$,
so the policy gradient computed using such a method will necessarily be biased.
For example, see the following works for examples that try to deal with the
distribution shift problem:
Munos et al (2016, https://arxiv.org/abs/1606.02647),
Wang et al (2016, https://arxiv.org/abs/1611.01224),
Gruslys et al (2017, https://arxiv.org/abs/1704.04651)

Putting aside the issue of whether ignoring the distribution shift is OK,
the main issues are the new policy gradient theorems derived from this
formulation. Both the $V(s,\theta)$ as well as $Q(a,s,\theta)$ formulations appear
flawed:
In the $V(s,\theta)$ case they propose the policy gradient:

$\nabla_\theta J(\theta) = \int d^{\pi_b}(s) dV(s,\theta)/d\theta ~~ds$ in equation 8.

However, the true policy gradient is:
$\nabla_\theta J(\theta) = \int \mu(s) dV(s, \theta)/d\theta ~~ds,$
where $\mu(s)$ is the start-state distribution. Actually they wrote
this also in equation 7, when they considered the $V(\theta)$ formulation,
but for some reason sampled from $d(s)$ instead for $V(s, \theta)$ when computing
the policy gradient in the $V(s,\theta)$ formulation.

In the $Q(a,s,\theta)$ formulation, they add an extra $dQ/d\theta$ term to
the policy gradient. Their motivation is the following:

$\nabla_\theta J(\theta) = \int d^{\pi_b} (dQ(a=\pi(s,\theta),s,\theta)/d\theta) dads$
  $   	= \int d^{\pi_b} dQ(a,s,\theta)/da*da/d\theta + dQ(a,s,\theta)/d\theta~~ dads$

However, this derivation stems from the flawed definition of J that is
not maximizing the sum of rewards over the trajectory distribution, but
maximizing some other objective that sums the value functions at all states
in the trajectory distribution. My strongest argument for why the original
off-policy derivations by Degris et al and Silver et al are less flawed is
the following:
If we are on-policy, i.e. $\pi_b = \pi$ and $d^{\pi_b} = d^{\pi}$ we would want
the off-policy policy gradient theorem to be unbiased, hence it should
revert to the standard policy gradient theorem. In the formulations
of Degris and Silver, this is indeed the case, and these theorems would
be unbiased in the on-policy setting. The new theorem in the current
paper, on the other hand, would have an extra $dQ/d\theta$ term, which would
bias the gradient. Therefore, I do not see any good theoretical reason to
add this term. Moreover, the practical performance did not improve, so
there is little evidence to suggest it as a heuristic either.

If someone would say that the original policy gradient
theorem requires the $dQ/d\theta$ term, I would urge them to look at the original
proofs---there is no approximation, these theorems are exact for the true
RL objective based on maximizing the rewards over the discounted trajectory
distribution. The intuition is that the remaining $dQ/d\theta$ term for the
remainder of the trajectory from a time-step t is estimated by summing
the $dQ/da\*da/d\theta$ or $Q\*dlog/d\theta$ terms for all the future time-steps.

Another more minor theoretical issue in the paper is that while the
theory considered the discounted state visitation distribution, the
discount factors are not added into the policy gradient in the algorithmic
sections. This omission is common, and tends to work well as a heuristic
(but it should at least be mentioned that such an approximation is made).
See the following papers for more discussion on this:
Nota and Thomas (2020, https://arxiv.org/abs/1906.07073)
Thomas (2014, http://proceedings.mlr.press/v32/thomas14.html)


5. Ask questions you would like answered by the authors to help you clarify your understanding of the paper and provide the additional evidence you need to be confident in your assessment.

How did the computational times compare? Was there much of an overhead to
using the more complicated critics including theta as an input?


6. Provide additional feedback with the aim to improve the paper. Make it clear that these points are here to help, and not necessarily part of your decision assessment.

For me to change my assessment, first the theoretical issues should be
fixed or cleared up.

Next, I have some possible suggestions:
1) Test also $V(s,\theta)$ on LQR as well as on zero-shot learning while sampling
s from the initial state distribution $\mu(s)$. This does not require interacting
with the environment (because you never apply any action), and I would consider
it fair in terms of comparing to $V(\theta)$. If the learning from the TD error
is working well, I would expect it to outperform the $V(\theta)$ formulation
in the zero-shot task.
2) Test the parameter value functions using the standard policy gradients
without adding the $dQ/d\theta$ term. Because you are using $Q(a,s,\theta)$, there
may be some learning to generalize across different policies due to the
theta input, so it may outperform the original policy gradients without
changing the policy gradient theorem. Actually, it would have been better to
perform such experiments as an ablation study from the beginning anyhow.
3) Test $Q(a,s,\theta)$ also on the LQR task to show it's correctness
(for example by sampling s from the initial state distribution and computing
the action). It may also be nice to test it in the zero-shot task as well.
4) Perhaps test combinations of the various gradients, for example taking
the average of the $V(\theta)$ gradient with the policy gradient using $Q$
(i.e. taking the average of two equivalent policy gradients).

If the above points are convincingly done, I may increase to marginal
accept. The current contributions are not enough for me to go higher than
that: taking away the proposed new policy gradients, the main contribution
is to add $\theta$ as an input to $V$ and $Q$, which I think is not enough.
Moreover, the advantage of adding $\theta$ as an input was not shown convincingly
using compelling evidence. Currently the most compelling evidence is the
zero-shot task, which shows that there is some generalization happening in
the $\theta$ space; however, what is missing to me, is a demonstration of how
this additional generalization helps in solving the original task in a more
data-efficient manner. Perhaps interleaving the policy search with longer
sessions of off-line learning (without any interaction) using $dV/d\theta$
to take advantage of the generalization may improve the data-efficiency
and show the advantage of the new method (exaplaining good practices on how
to do this may be a useful contribution). I think it would also be important
to show compelling evidence that including the s input helps in learning
better $V$ and $Q$ functions. Perhaps there are also other ways to better
show the advantage of the method.

Another option may be to change the problem setup, so that the new policy gradient theorems would be more sound. For example, using the original formulation of Degris'12 where $d^{\pi_b}(s)$ is the limiting distribution as $t \to \infty$ would make the new policy gradients correct; however, the standard setup would not correspond to this. One setup that would correspond to this objective is the following: an infinite horizon continuing setting, where the agent is never reset into the initial distribution, but has to continually change the policy to improve. The learning would iterate between running one behavioral policy until it converges to its stationary distribution, then optimizing a new policy while in the off-policy setting, then switching the behavioral policy to this new policy, and repeating the process. In this situation, $d^{\pi_b}(s)$ can be seen as the initial distribution for the new policy, and in this case the new policy gradient theorems would make sense. My previous argument about wanting the policy gradient theorem to be unbiased in the on-policy case would also be satisfied, because if $d(s)$ is stationary then
the $dQ/da\*da/d\theta$ and $dQ/d\theta$ gradients would differ by only a constant factor.

---

> ### Author Response · Authors · 2020-11-21
> **Response to reviewer #4 with comments and improvements [4/4]**
>
> >Currently the most compelling evidence is the zero-shot task, which shows that there is some generalization happening in the  $\theta$ space; however, what is missing to me, is a demonstration of how this additional generalization helps in solving the original task in a more data-efficient manner. Perhaps interleaving the policy search with longer sessions of off-line learning (without any interaction) using $dV/d\theta$ to take advantage of the generalization may improve the data-efficiency and show the advantage of the new method (exaplaining good practices on how to do this may be a useful contribution).
>
> We would like to emphasize that the generalization observed in the zero-shot learning experiments is already affecting the main results. In particular, with PSSVFs we are alternating online interactions with the environment where the policy is collecting more data and offline learning, where first the PSSVF is trained for 10 gradient steps and then the policy is trained completely offline for another 10 gradient steps. We found that, across many environments, 10 offline gradient steps were a good tradeoff between exploiting the generalization of V and remaining in the part of the parameter space where V is accurate. Measuring the generalization in the zero-shot learning tasks can be useful for determining the number of offline gradient steps to perform. Our algorithms using PSVFs and PAVFs also perform multiple offline gradient steps, since the behavioral policy is changing every episode, while the policy is updated every 50 timesteps.
>
> >I would expect $V(s,\theta)$ to outperform $V(\theta)$ due to using the state information, but this did not appear to be the case.
>
> >I think it would also be important to show compelling evidence that including the s input helps in learning better $V$ and $Q$  functions. Perhaps there are also other ways to better show the advantage of the method.
>
> We believe that in most of the cases it is hard to see an improvement of PSVF and PAVF over the simple PSSVF, because our algorithms based on TD learning, despite having the information on the state, have a much more complicated function to learn. Similarly, one could argue that ARS is outperforming DDPG in most of the tasks. Here the most interesting comparison is the one between $V(s,\theta)$, $Q(s,a,\theta)$ and DDPG and the one between $V(\theta)$ and ARS. Apart from Reacher, the PAVF obtained better results than DDPG in Swimmer, MountainCarContinuous and sometimes in Hopper.
>
> >How did the computational times compare? Was there much of an overhead to using the more complicated critics including theta as an input?
>
> Regarding the computational time, if we were performing the exact amount of gradient steps in PAVF as in DDPG, we would have our algorithm to be 4 times slower when using a 2-layers (64,64) MLP as policy. However, since our PAVF does not need to constantly track a single policy, we need much fewer policy and value functions updates. In the experiments we performed, PAVF with less updates was 30\% faster than DDPG when using a deep policy.

---

> ### Author Response · Authors · 2020-11-21
> **Response to reviewer #4 with comments and improvements [3/4]**
>
> >some discussion around off-policy learning seemed incomplete.
>
> We expanded our related work section including other algorithms that try to improve the off-policy policy theorem from Off-PAC.
>
> >Then they test a similar zero-shot learning procedure as they did for $V(\theta)$  at different stages of the learning (but as far as I understood, for $V(s,\theta)$ they sampled data from the replay buffer when training the policy (thus not fully without interacting with the data). Perhaps the authors can clarify this), and show that the newly learned policy can outperform the behavior policy, thus demonstrating the generalizability of the method.
>
> We would like to clarify our terminology of the 'offline setting': The requirement for an RL task to be offline is that there is no additional interaction with the environment when optimization is started. In our offline RL experiment, the critic needs to interact with the offline dataset in order to be trained. In an offline setting it is OK to use the data in order to perform policy gradient updates. The only important aspect is that no additional data is coming from the environment after we start learning.
>
> >Test the parameter value functions using the standard policy gradients without adding the $dQ/d\theta$ term. Because you are using $Q(s,a,\theta)$, there may be some learning to generalize across different policies due to the theta input, so it may outperform the original policy gradients without changing the policy gradient theorem. Actually, it would have been better to perform such experiments as an ablation study from the beginning anyhow.
>
> We included in  Appendix A.3.1 an ablation when training PAVF without the last part of the gradient $\nabla_\theta Q(s,a,\theta)$ in Swimmer and Hopper and with shallow and deep policies. We used the same procedure as for the original PAVF in order to tune the hyperparameters and evaluate the final performance on 20 different seeds. From the results, we observed that using the biased gradient (the one without $\nabla_\theta Q(s,a,\theta)$) the performance dropped significantly in Swimmer. In Hopper we observed a much smaller drop, possibly because both algorithms are converging to a sub-optimal behavior.
>
> >Test also $V(s,\theta)$ on LQR as well as on zero-shot learning while sampling s from the initial state distribution $\mu(s)$. This does not require interacting with the environment (because you never apply any action), and I would consider it fair in terms of comparing to $V(\theta)$. If the learning from the TD error is working well, I would expect it to outperform the $V(\theta)$ formulation in the zero-shot task.
>
> >Test  also $Q(s,a,\theta)$ on the LQR task to show it's correctness (for example by sampling s from the initial state distribution and computing the action). It may also be nice to test it in the zero-shot task as well.
>
> We are currently running additional experiments on zero-shot learning using $V(s,\theta)$ and $Q(s,a,\theta)$ in different environments and we will include them before the end of the rebuttal. However, it is difficult to have a fair comparison between them since, in this task, data are coming from the policy we are learning and thus strongly depend on the algorithm being used. In other words, in order to disentangle generalization and learnability, we would like to learn zero-shot having access to the same data. Perhaps a fair comparison would include an offline scenario like in our last experiment in which first we collect full trajectories using some policies $\pi_b$, then train offline PSSVF, PSVF and PAVF and finally train their policies.
>
> The purpose of our LQR experiment was to show how $V_w(\theta)$ is able to approximate $J(\pi_{\theta})$ over the policy parameter space. We did not compare this with $V(s,\theta)$ and $Q(s,a,\theta)$ because in order to represent these value functions in a 2D plot we would have to remove the bias from the policy, hence the comparison would not be fair. Before the end of the rebuttal, we will provide some visualization plots of $V(s,\theta)$ and $Q(s, \pi(s), \theta)$ as a function of $s$ and $\theta$, where a deterministic policy with one weight and no bias is used for learning. Note that for the LQR experiment we minimally tuned the hyperparameters because our goal was simply to visualize the algorithm operation and not to achieve the best performance.

---

> > ### Comment · AnonReviewer4 · 2020-11-23
> > **LQR task**
> >
> > "The purpose of our LQR experiment was to show how
> > is able to approximate over the policy parameter space. We did not compare this with and because in order to represent these value functions in a 2D plot we would have to remove the bias from the policy, hence the comparison would not be fair. "
> >
> > You can evaluate $V(s, \theta)$ and $Q(s,a,\theta)$ on the LQR task as a function of only $\theta$ by taking the expectation over $\mu(s)$. E.g. sample many points from $\mu$ take the average of $V(s,\theta)$ for a particular $\theta$ and show that this average has a sensible value (by plotting it, and comparing).

---

> ### Author Response · Authors · 2020-11-21
> **Response to reviewer #4 with comments and improvements [2/4]**
>
>
> >My strongest argument for why the original off-policy derivations by Degris et al and Silver et al are less flawed is the following: If we are on-policy, i.e. $\pi_b = \pi$ and  $d^{\pi_b} = d^{\pi}$we would want the off-policy policy gradient theorem to be unbiased, hence it should revert to the standard policy gradient theorem. In the formulations of Degris and Silver, this is indeed the case, and these theorems would be unbiased in the on-policy setting. The new theorem in the current paper, on the other hand, would have an extra $dQ/d\theta$ term, which would bias the gradient. Therefore, I do not see any good theoretical reason to add this term. Moreover, the practical performance did not improve, so there is little evidence to suggest it as a heuristic either.
>
> If we were optimizing $J_b$ on-policy, the off-policy policy gradients in Degris et al. and Silver et al. would be equivalent to the on-policy policy gradient only because they make the aforementioned approximation of dropping $\nabla_{\theta}Q^{\pi_{\theta}}(s,a)$. If they were able to estimate $\nabla_{\theta}Q^{\pi_{\theta}}(s,a)$, they would also have an additional term. The fact that if we take $d^{\pi_b} = d^{\pi_{\theta}}$ and $\pi_b = \pi_{\theta}$ we do not have the on-policy policy gradient theorem is a problem of the objective function, which does not consider the distribution shift and NOT a problem of our method, which is an improvement upon Degris et al. and Silver et al. If we accept $\int_{\mathcal{S}} d^{\pi_b}(s) V^{\pi_{\theta}}(s) \mathrm{d} s$ as off-policy objective, which is the objective that many researchers are using, our off-policy policy gradients are exact.
>
>
> >they replace the distribution $d^{\pi}(s)$ with a distribution $d^{\pi_b}(s)$
>  gathered using a behavioral policy (so they are working off-policy). However, they do not apply an importance weighting correction for the distribution shift, and just ignore the importance weights (Note that this is also done by Silver et al (2014) in deterministic policy gradients, and by Lillicrap et al (2015) in DDPG, so it is not that strange per se, as long as it gives better practical performance. However, it should at least be acknowledged that the importance weights are being ignored). Note that they still apply an importance weight on the actions ($\pi(a|s)/\pi_b(a|s)$) once the state is sampled from the data buffer, however, this does not correct for the distribution shift from $d^{\pi}$ to $d^{\pi_b}$, so the policy gradient computed using such a method will necessarily be biased. For example, see the following works for examples that try to deal with the distribution shift problem: Munos et al (2016, https://arxiv.org/abs/1606.02647), Wang et al (2016, https://arxiv.org/abs/1611.01224), Gruslys et al (2017, https://arxiv.org/abs/1704.04651)
>
> We agree with the reviewer that we should acknowledge that in our off-policy formulation we are ignoring the distribution shift from $d^{\pi_b}(s)$ to $d^{\pi_{\theta}}(s)$. We related this to recent methods trying to solve the distribution shift problem [5]. However, it seems to us that the papers suggested by the reviewer do not deal with the distribution shift problem from $d^{\pi_b}(s)$ to $d^{\pi_{\theta}}(s)$. They deal instead with the variance introduced by the importance weights (IWs) $\frac{\pi_{\theta}(a|s)}{\pi_{b}(a|s)}$ and the bias introduced when IWs are clipped. In particular, ACER(Wang et al), is mentioned even in a paper on the distribution shift problem[5] as one of the methods that, like ours, completely ignore the distribution shift. It is worth mentioning that ACER(Wang et al) is using an off-policy formulation similar to ours. They start from the formulation of Degris et. al. with $d^{\pi_b}(s)$ being the stationary distribution of the behavioral policy and they then approximate it using trajectories from the behavioral policy. We instead, similar to Silver et. al., define directly the off-policy objective with respect to the data obtained from the trajectories.
>
> >Another more minor theoretical issue in the paper is that while the theory considered the discounted state visitation distribution, the discount factors are not added into the policy gradient in the algorithmic sections. This omission is common, and tends to work well as a heuristic (but it should at least be mentioned that such an approximation is made). See the following papers for more discussion on this: Nota and Thomas (2020, https://arxiv.org/abs/1906.07073) Thomas (2014, http://proceedings.mlr.press/v32/thomas14.html)
>
> We mentioned the omission of the discount factor from $d^{\pi_b}(s)$ when deriving algorithms for $V(s,\theta)$ and $Q(s,a,\theta)$. We thank the reviewer for pointing out this issue.
>
>
> [5]Yao Liu, Adith Swaminathan, Alekh Agarwal, and Emma Brunskill. Off-policy policy gradient with state distribution correction.arXiv preprintarXiv:1904.08473, 2019.

---

> ### Author Response · Authors · 2020-11-21
> **Response to reviewer #4 with comments and improvements [1/4]**
>
>
> We thank the reviewer for their valuable and detailed feedback. The insightful comments provided by the reviewer have helped us to significantly improve our submission.
> Below are our specific responses to the concerns raised by the reviewer:
>
> >The theoretical issues in this paper start in equation 1. They write: "... we can express the maximization of the expected cumulative reward in terms of the state-value function:"
> $$J(\pi_{\theta}) = \int_{\mathcal{S}} d^{\pi_{\theta}}(s) V^{\pi_{\theta}}(s) \mathrm{d} s$$
>  (in the paper)
> where $d(s)$  is the discounted state visitation distribution. However, this is not the RL objective. The RL objective would be
> $$J(\pi_{\theta}) = \int_{\mathcal{S}} d^{\pi_{\theta}}(s) R(s) \mathrm{d} s$$
>  (what it should actually be)
>
> Let us start from the on-policy setting. There was a major point of confusion when we claimed that the on-policy objective is $J(\pi_{\theta}) = \int_{\mathcal{S}} d^{\pi_{\theta}}(s) V^{\pi_{\theta}}(s) \mathrm{d} s$ and we agree that the RL objective in the on-policy case is $J(\pi_{\theta}) = \int_{\mathcal{S}} \mu_0(s) V^{\pi_{\theta}}(s) \mathrm{d} s$, which becomes $J(\pi_{\theta}) = \int_{\mathcal{S}} \mu_0(s) V(s,\theta)\mathrm{d} s$ using PVFs.
>
> In the on-policy formulation, the gradient of the action-value function with respect to the policy parameters is not present in the original on-policy policy gradient theorems (Th 1 in Sutton, 1999 [1] and Th1 in Silver, 2014 [2]). This term is also not present in the on-policy policy gradient theorem with PVFs. In the on-policy case we can expand it using Bellman equation and following the exact same procedure as in Sutton, 1999 [1] and Silver, 2014 [2] we obtain an expression that depends on $d^{\pi_{\theta}}(s)$ (see Th 3.1 and Th 3.2 in revised pdf).
>
>
> In the off-policy case, however, this is different. A widely used objective for off-policy RL is $J_b(\pi_{\theta}) = \int_{\mathcal{S}} d^{\pi_b}(s) V^{\pi_{\theta}}(s) \mathrm{d} s$, where $d^{\pi_b}$ is the limiting distribution of the states under $\pi_b$ for continuing tasks[3,4] or the discounted weighting of states encountered starting at $s_0 \sim \mu_0(s)$ and following the policy $\pi_{b}$[2] when we have access to trajectories. In our work we use the latter, although our results can be easily extended to the continuing setting.
>
>  When taking the gradient of the off-policy objective, either in standard RL or with PVF, the gradient of the action-value function with respect to the policy parameters must be estimated and can no longer be condensed into $d^{\pi_{\theta}}$, since we have only access to $d^{\pi_b}$. In DPG[2], this term is completely ignored in eq. 15 as "Analogous to the stochastic case (see Equation 4), we have dropped a term that depends on $\nabla_{\theta}Q^{\pi_{\theta}}(s,a)$; justification similar to Degris et al. (2012b)[3] can be made in support of this approximation". On the other hand, Degris et al.[3] claim "The final term in this equation, $\nabla_{\theta}Q^{\pi_{\theta}}(s,a)$, is difficult to estimate in an incremental off-policy setting. The first approximation involved in the theory of OffPAC is to omit this term". They provide a justification on this approximation proving that the set of stationary points of the approximated gradient is included in the set of stationary points for the true gradient. However, in the off-policy setting, this is true only when the policy is tabular (see errata in section B of Degris et al.[3]). PAVFs can directly estimate this term, providing a more theoretically sound off-policy policy gradient for $J_b$.
>
> [1]Richard S. Sutton, David McAllester, Satinder Singh, and Yishay Mansour.Policy gradient methods for reinforcement learning with function approximation.   In Proceedings of the 12th International Conference on Neural Information Processing Systems,  NIPS’99,  pages  1057–1063,  Cambridge,MA, USA, 1999. MIT Press.
> [2]David Silver, Guy Lever, Nicolas Heess, Thomas Degris, Daan Wierstra, and Martin Riedmiller. Deterministic policy gradient algorithms. In Proceedings of the 31st International Conference on International Conference on Machine Learning - Volume 32, ICML’14, pages I–387–I–395.JMLR.org, 2014.
> [3]Thomas Degris, Martha White, and Richard S. Sutton. Off-policy actor-critic. In Proceedings of the 29th International Conference on International Conference on Machine Learning, ICML’12, pages 179–186, USA, 2012.Omnipress.
> [4] Ehsan Imani, Eric Graves, and Martha White. An off-policy policy gradient theorem using emphatic weightings. In Advances in Neural Information Processing Systems, pages 96–106, 2018.

---

> > ### Comment · AnonReviewer4 · 2020-11-22
> > **The goal of off-policy RL is to solve the original RL task using off-policy data**
> >
> > Thank you for the response and the clarifications.
> >
> > The goal of off-policy RL is to solve the original RL task using
> > off-policy data. I don't think this is controversial. Any newly
> > defined objective, such as $\int d^b V(s) ds$ is useful only in so far
> > as it helps in solving this original RL task. In fact, all researchers
> > who claim to use this off-policy objective, never actually evaluate the
> > performance of their algorithms based on this objective. To evaluate
> > the performance, everyone uses the episodic returns corresponding to
> > the original RL objective that sums the rewards over the trajectory
> > distribution, not the value functions, because that is the true
> > objective they are interested in (also the work by the authors here
> > does the same).
> >
> > Reiterating my main argument, a simple sanity check for any off-policy
> > policy gradient theorem is whether the theorem gives an unbiased gradient
> > estimator in the on-policy case. This sanity checks works for the previous
> > gradient theorems by Degris et al and Silver et al, which are unbiased in
> > the on-policy case, but not for the method proposed in this paper; the
> > gradient would be biased in the on-policy case due to adding an
> > unnecessary $\nabla_\theta Q$ term. Due to this, I cannot see how the
> > newly proposed policy gradients could be conceived as being somehow
> > theoretically more sound than the original works by Degris or Silver.
> > While these previous works are biased due to ignoring the distribution shift
> > from $d^\pi$ to $d^b$, the current work is biased due to both ignoring
> > the distribution shift, and due to adding the extra gradient term.
> >
> > Indeed Degris and Silver mentioned that they believe they should have
> > this extra $\nabla_\theta Q$ term. However, Degris mentioned this
> > because they considered a different setting where $d$ is not the
> > discounted state visitation distribution, but the stationary
> > distribution as $t\to \infty$, while Silver just copied Degris's
> > reasoning, but failed to mention that they had swapped $d$ to be the
> > discounted state visitation distribution without providing any
> > justification for why they swapped it. Moreover, the algorithms they
> > ended up with in the end matter more than what their intentions were
> > when creating their algorithm (I believe their intentions are
> > irrelevant to deciding which policy gradient theorems are more sound).
> >
> > I'm afraid I am voting to reject the paper unless the newly proposed
> > theorems are removed, or the problem setting is changed so that the
> > new theorems would correspond to the problem setting (as described in my
> > original review). Perhaps another way I may be able to accept is if the
> > paper were to admit that probably it is incorrect to add the extra term,
> > but they choose to add it anyway to explore the idea (it is true that
> > there is some confusion in the literature about this). However, currently
> > the paper claims that the new theorems are more correct than the
> > previous ones, and I don't find that appropriate.
> >
> >
> > About some other details in the response:
> >
> > Regarding the references I provided for correcting the distribution shift,
> > indeed these were not fully appropriate; I took them from another publication,
> > but should have checked them more carefully, and I am sorry. However, the main point
> > of these works was that off-policy corrections can be performed by chaining
> > importance weights together over the whole trajectory. To perform the
> > off-policy correction from $d^b$ to $d^\pi$, it is necessary to know
> > the importance weight $d^\pi_t/d^b_t$ for each time step $t$. These
> > distributions can be expressed as
> > $d_t(s) = \int \mu_0(s_0)\pi(a_0|s_0)p(s_1|s_0,a_0)\pi(a_1|s_1)...p(s_t|s_{t-1}, a_{t-1}) ds_0da_0ds_1...ds_{t-1}da_{t-1}$, where the $p$ is the transition dynamics.
> > Then, by writing
> > $\int d_t(s_t)\pi(a_t|s_t) \nabla_\theta\log \pi(a_t|s_t)Q(s,a) ds_tda_t = \int \pi(a_t|s_t)\nabla_\theta\log \pi(a_t|s_t)Q(s,a)da_t\mu_0(s_0)\pi(a_0|s_0)p(s_1|s_0,a_0)\pi(a_1|s_1)...p(s_t|s_{t-1}, a_{t-1}) ds_0da_0ds_1...ds_{t-1}da_{t-1}$,
> > it becomes possible to compute an unbiased policy gradient by applying importance weights along a sampled trajectory.
> > By taking the ratio, the transition dynamics disappears, because it is
> > the same for both policies, and one is left with the importance weight
> > $\Pi_{t=0}^{t-1}\frac{\pi_\theta(a_t|s_t)}{\pi_b(a_t|s_t)}$ that depends only
> > the actions chosen along the trajectory. This can allow to correct for the
> > distribution shift, and the methods such as Retrace, etc. could be useful
> > in trading off bias and variance in this importance sampling correction.

---

> > > ### Comment · AnonReviewer4 · 2020-11-23
> > > **Changed my mind a bit**
> > >
> > > I read a bit more literature, and one argument I came across for the $J_b$ objective is the following:
> > >
> > > If $d^{\pi_b}$ places probability mass everywhere where $\mu_0$ places probability mass, then the optimal solution to
> > > $\int d^{\pi_b}(s) V_\theta(s)ds$ will also be optimal for the original objective $\int \mu_0(s) V_\theta(s)ds$, because the optimal policy is optimal independent of the start state distribution. However, this argument will break down when function approximation is used. From this point of view, I agree that it is also reasonable to add the $\nabla_\theta Q$ term; however, it is still not clear whether it is theoretically better than the theorems in Degris and Silver, because of the other arguments I made, and for several other reasons that I will explain below. But I agree that experimentally evaluating the method with $\nabla_\theta Q$ is a valid research question, and if the discussion around the different theorems is good, I may like the paper.
> > >
> > > I list the main reasons for why it is not clear that the new theorems are better (not in the function approximation setting, nor in the setting with no function approximation):
> > > - As I explained,the policy gradient from Degris and Silver appears theoretically closer to the gradient for the true on-policy objective, because it only omits the importance weights for the distributions, whereas the new method differs by both the importance weights, and the gradient (so, in the function approximation setting, it is not clear, which method would be more appropriate for finding a solution that works well on the original RL objective).
> > > - In the no function approximation setting, Degris' work actually proved that even if you omit the $\nabla_\theta Q$ term, the policy gradient algorithm will converge to the optimal solution of $J_b$ (this also requires that $d^b$ will visit all states in $d^\pi$), so adding in this extra term is not necessary to solve the objective, and it's not clear that adding it in will improve the performance (see Theorems 1 and 2 in Degris).
> > >
> > > Another work on solving the distribution shift problem is AlgaeDice (https://arxiv.org/pdf/1912.02074.pdf), and they have some good discussion on these topics.
> > >
> > >
> > > In conclusion, what I think should be explained in the paper is: 1) The solution to $J_b$ would also be optimal for the original objective as long as $d^b$ visits all states in $\mu_0$. 2) Explain the pros and cons of adding or omitting the extra $\nabla_\theta Q$ term, and perform experimental analysis to try to solve the debate. It may be nice to find some simple toy example where adding in the extra $\nabla_\theta Q$ term will lead to a better solution (off the top of my head, your theorem would require that $d^b$ visits all states in $\mu_0$ whereas Degris or Silver additionally require that $d^b$ visits all states in $d^\pi$, so potentially you could show an example where your method beats theirs when the second condition is not met. However, I guess this will also require accurately learning Q in the states not visited by $d^b$, so I am not sure whether this is feasible).

---

> > > > ### Author Response · Authors · 2020-11-23
> > > > **A more detailed comparison with Off-PAC**
> > > >
> > > >
> > > > We provided clarifications in the answer above regarding a comparison of the true and approximate gradient of $J_b$. Below we extend them based on the more recent comments by the reviewer.
> > > >
> > > > We agree with the reviewer that the maximum of $J$ is the maximum of $J_b$, assuming we are in a markovian setting and the support of $\mu_0$ is included in the support of $d_{\infty}^{\pi_b}$. We stated this explicitly in the paper.
> > > >
> > > > >In the no function approximation setting, Degris' work actually proved that even if you omit the $\nabla_{\theta}Q$
> > > >  term, the policy gradient algorithm will converge to the optimal solution of $J_b$
> > > >  (this also requires that $d^b$
> > > >  will visit all states in $d^{\pi}$
> > > > ), so adding in this extra term is not necessary to solve the objective, and it's not clear that adding it in will improve the performance (see Theorems 1 and 2 in Degris).
> > > >
> > > > Unfortunately, the proofs by Degris et al. (https://arxiv.org/pdf/1205.4839.pdf) work only if the policy is tabular (i.e. with different weights for each state). The errata in Appendix B in Degris et al. states "The current theoretical results only apply to tabular representations for the policy $\pi$ and not necessarily to function approximation for the policy". Our results, instead, are valid for general policy parametrization.
> > > >
> > > > It seems to us that our policy gradients are more sound than those in Degris et al., Silver et al., due to these considerations: following the exact gradient of $J_b$, if we can obtain the maximum of $J_b$, under covering assumptions this will match the maximum of $J$. On the other hand, Degris et al. (probably the same argument holds in Silver et al.) for non-tabular policies claim:  "Because the approximate gradient is not
> > > > the gradient of any objective function, it is not clear if any stable minima exist" (errata in Appendix B in Degris et al). Moreover, the policy improvement theorem for $J_b$ holds only in the tabular setting (or in the on-policy case), so it is not guaranteed that $J_b$ is maximized when following the approximate gradient. Nevertheless, as pointed out in the previous response, it might happen in some cases that the approximate gradient is closer to the gradient of $J$ than the true gradient of $J_b$ (at least in the near-on-policy case).

---

> > > > > ### Comment · AnonReviewer4 · 2020-11-23
> > > > > **In the non-tabular function approximation case**
> > > > >
> > > > > In the function approximation case, with a space of realizable policies $C$, you can only find the optimal policy for $J_b$ contained in $C$. This optimal policy will not necessarily coincide with the optimal policy for $J$ nor with the optimal policy for $J$ contained within $C$. Therefore, I am not convinced that your approach has a theoretical advantage, as both your approach and Degris' approach converges to the optimal policy for $J$ in the tabular case, and neither of the approaches is guaranteed to converge to the optimal policy in the non-tabular function approximation case.
> > > > >
> > > > > However, I am no longer concerned about the newly proposed policy gradient theorems, which lifts my main concern.

---

> > > > > > ### Author Response · Authors · 2020-11-23
> > > > > > **Possible theoretical advantage**
> > > > > >
> > > > > > It is true that in the approximate case the optimum of $J_b$ will not necessarily correspond to the optimum of $J$. In general, it is also not guaranteed to find the global maximum of $J_b$ at all, as the objective might be highly non-convex. What one can hope is to prove that an algorithm converges to a point which corresponds to one of the local maxima of $J_b$. The short answer and main argument to why we believe PVFs offer a theoretical advantage is that they optimize for $J_b$, while we do not know what Off-PAC (Degris et al.) or DPG (Silver et al.) are optimizing for, since their gradient is not the gradient of any objective. Therefore, if we are able to prove that in the actor-critic algorithm with PAVF the actor converges, it necessarily converges to a local optimum of $J_b$.
> > > > > >
> > > > > > Degris et al. tried to prove that the actor converges to a local optimum of $ J_b$ in their Th 3, using stochastic approximation arguments. The theorem shows that, under some assumptions, the policy parameters converge to a point such that the approximate gradient of $J_b$ is zero. Unfortunately, in the non-tabular setting it is not guaranteed that this will be also a local maximum for $J_b$. With PAVF some of the conditions in the stochastic approximations (Borkar, 2009) must be modified: Q would be linear in some feature of the states and in the policy parameters; since Q receives $\theta$ as input, we would work in a continuous setting; $GTD(\lambda)$ should be extended to PAVF. Under these modified conditions plus the standard stochastic approximation conditions, we believe it would be possible to follow a similar approach to Th 3 in Degris et al. in order to satisfy the necessary conditions that ensure the actor to converge to a local optimum of the gradient it is optimizing for. For PAVF this would be a local optimum of $J_b$. Note that here there would be no distinction between tabular or non-tabular policies: since we are optimizing $J_b$ while following the true gradient, the only requirement on the policy is that it satisfies some assumptions on stochastic approximation and we do not need further assumptions (e.g. tabular policy) for the gradients to match.
> > > > > >
> > > > > > In this work we did not try to prove convergence of PVFs (not even for tabular policies), as many of the assumptions needed are far from what we use in the experiments (linear value function, direct samples from $d_{\infty}^{\pi_b}$). However, we believe that, since our PAVF has access to the true gradient of $J_b$, it will converge to a local optimum of $J_b$. In future work, we plan to formally prove it and to compare the performance of PAVF, Off-PAC, DPG and other off-policy actor-critic methods under linear value-function approximation.
> > > > > >
> > > > > > Vivek S Borkar.Stochastic approximation:  a dynamical systems viewpoint, volume 48.  Springer,2009.

---

> > > > > > > ### Comment · AnonReviewer4 · 2020-11-24
> > > > > > > **Theoretical advantage is still not certain, but I agree that the idea has merit**
> > > > > > >
> > > > > > > I still think that the convergence to the optimal solution in the tabular case for the standard OffPAC provides a good theoretical grounding for the algorithm. And in the non-tabular case what matters is which algorithm will lead to better performance on the original objective $J$, and from the theory it is not that clear which one will be better (indeed the standard OffPAC is widely used, and seems to give decent performance). I think it should at least be mentioned in the paper that the original OffPAC proved convergence in the tabular case even when ignoring $\nabla_\theta Q$. But basically, I agree that it is an interesting question to see what effect the $\nabla_\theta Q$ term has, so I will update my score accordingly.
> > > > > > >
> > > > > > > Some more notes about the current draft:
> > > > > > > "while traditional off-policy actor-critic methods introduce off-policy policy gradients that are only biased approximations to the true gradient since they do not estimate or compute the gradient of the action-value function with respect to the policy parameters∇θQπθ(s,a)(Degris et al., 2012; Silver et al., 2014)"
> > > > > > >
> > > > > > > This should probably say that it is an approximation to the true gradient of the OffPAC objective, otherwise it may be misleading.
> > > > > > >
> > > > > > > Equations 1 and 2 are repeated in 8 and 9.
> > > > > > >
> > > > > > > Probably it should be mentioned that dropping the $\nabla_\theta Q$ term was justified in previous work by showing that the algorithm will still converge in the tabular setting (but you can argue why you think your approach is better).

---

> > > > > > > > ### Author Response · Authors · 2020-11-25
> > > > > > > > **We agree that OffPAC is a principled method for tabular policies**
> > > > > > > >
> > > > > > > > We agree that OffPAC is an important work and that it is theoretically sound when using tabular policies. We mentioned in the paper that it converges for tabular policies and we stated more clearly that their gradient is biased for the objective they introduce.
> > > > > > > >
> > > > > > > > >Equations 1 and 2 are repeated in 8 and 9.
> > > > > > > >
> > > > > > > > In Eq. 8 and 9 we provided the on-policy policy gradient for PVFs and we proved it in Appendix 2. The proof is trivial but we included it with the on-policy formulation for completeness. We simply showed that $\nabla_{\theta}Q(s,a,\theta)$ can be written in terms on $d^{\pi_{\theta}}$, exactly like in the original proofs.

---

> > > ### Author Response · Authors · 2020-11-23
> > > **Problem formulation and J vs J_b**
> > >
> > > We thank the reviewer for their prompt reply.
> > >
> > > We changed our problem formulation such that, like in the original formulation by Degris et al., we use the limiting distribution of the states under the behavioral policy. We called this term $d_{\infty}^{\pi_b}$ in order to distinguish it from the discounted weighting of states $d^{\pi_b}$. We updated the theoretical results accordingly. In practice, we do not have access to $d_{\infty}^{\pi_b}$, so in the derivation of the algorithms and in the experiments we approximate it by sampling trajectories generated by the behavioral policy. This is the same formulation and approximation done in ACER (Wang et al).
> > >
> > > >Reiterating my main argument, a simple sanity check for any off-policy policy gradient theorem is whether the theorem gives an unbiased gradient estimator in the on-policy case. This sanity checks works for the previous gradient theorems by Degris et al and Silver et al, which are unbiased in the on-policy case, but not for the method proposed in this paper;
> > >
> > >
> > >
> > > We agree with the reviewer that there is confusion in the literature about the off-policy objective. One main question is the following. Let $J_b$ be the practical off-policy objective that is widely used (Degris et al, Imani et al., Wang et al.) and let $J$ be the true RL objective. Let $\tilde \nabla_{\theta}J_b(\pi_{\theta})$ be the approximate gradient of $J_b$ provided by Degris et al. and let $\nabla_{\theta}J_b(\theta)$ be the true gradient of $J_b$ (ours, Imani et al.). Under which conditions is $|\nabla_{\theta}J(\pi_{\theta}) - \tilde \nabla_{\theta}J_b(\pi_{\theta})|<|\nabla_{\theta}J(\pi_{\theta}) - \nabla_{\theta}J_b(\theta) |$? In other words, when is the approximate gradient for $J_b$ a better direction of improvement for the original problem than the true gradient for $J_b$? If we are on-policy, we agree with the reviewer that clearly the approximate gradient is better, because it simply reverts to the on-policy policy gradient. However, when we are off-policy, this is an open question. In particular, for a single state the answer depends on $d_{\infty}^{\pi_b}(s)$, $d_{\infty}^{\pi_{\theta}}(s)$ and $\mu_0(s)$, while in expectation it depends also on $Q$, $\pi_{\theta}$ and $\gamma$. Finding a solution to this problem is beyond the scope of this work and we acknowledge that in the off-policy setting it is not clear if it is better to use the true gradient of $J_b$ (like ours and Imani et. al.) or following an approximate approach (Degris et al., Wang et al.).  Our ablation suggests that, in the Swimmer environment, using $\nabla_{\theta}J_b(\theta)$ instead of $\tilde \nabla_{\theta}J_b(\pi_{\theta})$ is better when evaluating $J$. Note that works trying to find the true gradient for $J_b$ like ours and Imani et al. will necessarily have an on-policy policy gradient which is biased because the objective functions $J_b$ and $J$ are not matching everywhere.
> > >
> > > >I'm afraid I am voting to reject the paper unless the newly proposed theorems are removed, or the problem setting is changed so that the new theorems would correspond to the problem setting (as described in my original review). Perhaps another way I may be able to accept is if the paper were to admit that probably it is incorrect to add the extra term, but they choose to add it anyway to explore the idea (it is true that there is some confusion in the literature about this). However, currently the paper claims that the new theorems are more correct than the previous ones, and I don't find that appropriate.
> > >
> > > We revised some claims in the paper and acknowledged that our gradients, despite being exact for $J_b$, might be worse than the approximate gradients for maximizing $J$.
> > >
> > > >Regarding the references I provided for correcting the distribution shift [...]. However, the main point of these works was that off-policy corrections can be performed by chaining importance weights together over the whole trajectory.
> > >
> > > We agree with the reviewer that in trajectory-based off-policy RL one can correct for the distribution shift of the entire episode by taking the product of importance weights over the trajectory. Like in the works suggested by the reviewer, it would be possible to use multi-step estimation of the action-value function $Q(a,s,\theta)$ and techniques like Retrace could help to reduce the variance.

---

### Official Review · AnonReviewer1 · 2020-10-25
**Interesting in terms of idea, but unclear advantage over common approaches**

**Rating:** 6
**Confidence:** 4

**Review:**

— idea:

A new class of value functions is introduced where the value function takes the parameters of the policy as input, in addition to its common inputs (state or state-action). The proposed type of value functions, PVFs, are also useful for off-policy learning and generalizing over policies while the common value functions lost their information about previous policies.

— comments:

1- Regarding the first algorithm, PSSVF, until converging, the data that is stored in the replay buffer does not correspond to a "reasonable" policy, unless having a prioritized replay buffer. I am also concerned about it being over fitted to the early policies and not being able to  overcome this. I see in the experiments that using PSSVF, policy is converged but am not convinced about it.

2- Another concern of mine wrt the proposed PVFs is about their sample efficiency. It would be interesting to see a comparison between DDPG and PVF based methods on their sample efficiency.

3- In the experiments section (4.2), expect for a few cases, ARS is either the best one or does not differ significantly from PVF based methods. Having this in mind, my question is that have you tried to find the best set of hyperparameters for ARS and DDPG as well as your proposed method? If the answer is no, I would like to see that experiments where ARS and DDPG have their best set of hyperparameters.

4- I am also interested in seeing results for deep policy zero shot learning. In section 4.3, authors just mention: "When using deep policies, we obtained similar results only for the simplest environments." which is not convincing without showing results.

5- As mentioned in the last part of the paper, the proposed method hugely suffers from the curse of dimensionality. However, as an initial step, PVFs seem interesting and could be beneficial in terms of learning generalized value functions.

-- minor issues:

In the first line of the paragraph above the experiments section (4), starting with "Algorithm 4 (Appendix) uses an ...", there is a redundant "and". One of them should be removed.



Overall, I liked the idea presented in the paper and would like to see what their next step would be. But the current version of the paper could benefit from more in depth experiments. I believe the most important weakness of the paper lies in the experiment section. It can be much richer and more insightful.



[1] Sutton, Richard S., et al. "Horde: A scalable real-time architecture for learning knowledge from unsupervised sensorimotor interaction." The 10th International Conference on Autonomous Agents and Multiagent Systems-Volume 2. 2011.

---

> ### Author Response · Authors · 2020-11-21
> **Response to reviewer #1 with comments and improvements**
>
> We thank the reviewer for their valuable feedback.
> We have improved our submission, here is a summary:
>
> >1- Regarding the first algorithm, PSSVF, until converging, the data that is stored in the replay buffer does not correspond to a "reasonable" policy, unless having a prioritized replay buffer. I am also concerned about it being over fitted to the early policies and not being able to overcome this. I see in the experiments that using PSSVF, policy is converged but am not convinced about it.
>
> We agree with the reviewer that PSSVF might be oversampling initial policies and that prioritized replay could help to sample data more uniformly.
>     We did not include different sampling techniques because we wanted to provide results for the most simple algorithms, limiting the number of tricks necessary.
>     Note that this problem affects less the PSVF and PAVF, since they receive much more data (one per transition instead of one per episode) and in some environments (Swimmer and Hopper) they have a small replay buffer corresponding to 1/10 of the available data during learning.
>
> >2- Another concern of mine wrt the proposed PVFs is about their sample efficiency. It would be interesting to see a comparison between DDPG and PVF based methods on their sample efficiency.
>
>  In our experiments we have already performed an extensive study comparing sample efficiency between DDPG, ARS and PVFs.
>   In particular, figures 2,9 and 10 analyze the sample efficiency on 7 different tasks and 3 policy architectures for all methods.
>
> >3- In the experiments section (4.2), expect for a few cases, ARS is either the best one or does not differ significantly from PVF based methods. Having this in mind, my question is that have you tried to find the best set of hyperparameters for ARS and DDPG as well as your proposed method? If the answer is no, I would like to see that experiments where ARS and DDPG have their best set of hyperparameters.
>
>  In all our experiments we performed an extensive hyperparameter search for ARS and DDPG.
>     The results in figures 2,9 and 10 correspond to the best hyperparameters found in ARS and DDPG.
>     We reported the value of the best hyperparameters for ARS and DDPG in Table 5 and 8 and a sensitivity analysis for all algorithms in figures 11,12,13,14,15.
>     We reported the procedure we used to find the best hyperparameters and to evaluate all algorithms at the beginning of Appendix A.4.3.
>     Note that, apart from Reacher task, ARS is never significantly better than PSSVF.
>
> >4- I am also interested in seeing results for deep policy zero shot learning. In section 4.3, authors just mention: "When using deep policies, we obtained similar results only for the simplest environments." which is not convincing without showing results.
>
>  Before the end of the rebuttal, we will include some results for zero-shot learning using deep policies, as well as more zero-shot learning results using PSVF and PAVF with linear policies. As mentioned in the paper, with deep policies we will have good zero-shot performance only in the most simple tasks.
>
> >5- As mentioned in the last part of the paper, the proposed method hugely suffers from the curse of dimensionality. However, as an initial step, PVFs seem interesting and could be beneficial in terms of learning generalized value functions.
>
>  We agree that the curse of dimensionality is the main limitation of our approach and we believe that our experimental results provide a strong baseline for methods which will try to reduce the dimensionality of the policy such as policy embeddings.
>
> >[1] Sutton, Richard S., et al. "Horde: A scalable real-time architecture for learning knowledge from unsupervised sensorimotor interaction." The 10th International Conference on Autonomous Agents and Multiagent Systems-Volume 2. 2011.
>
> We did not find this citation in the review. Could the reviewer elaborate on this? We are happy to expand our connections with General Value Functions.

---

### Official Review · AnonReviewer3 · 2020-10-29
**Interesting connection to DPG, few technical errors**

**Rating:** 7
**Confidence:** 5

**Review:**

On page 2, in the background section: the discounted state distribution, what you wrote is not a distribution (doesn't sum to 1). In order to define this $d^{\pi_\theta}$ properly, you can multiply everything by $1-\gamma$. The interpretation is that you "reset" in your initial distribution $\mu_0$ with probability $1 - \gamma$ at every step, or continue in the discounted stationary distribution with probability $\gamma$.

In think that theorem 3.1 is incorrect. I think that this is meant to describe an off-policy setting where we are collecting data from $\pi_b$ but want the policy gradient for $\pi_\theta$. In this case, the importance sampling weight should be $\frac{d_\theta(s,a)}{d_b(s,a)}$ not $\frac{\pi_\theta(a|s)}{\pi_b(a|s)}$ (where $d_b$ is the discounted stationary distribution, see above comment too). Equation 9 follows from the chain rule (because the Q function now depends on $\theta$ explicitly) using the off-policy formulation in Degris (2012), which is incorrect.

Notes:
- PVF: to me this acronym is strongly synonymous with Mahadevan's proto-value functions (PVFs), circa 2007. How about "PBVF" instead? Maybe I'm old

> we optimize for the undiscounted objective
this should be reflected in your notation and problem formulation

> can be used only for episodic tasks
it doesn't have to. See "regenerative method" in Monte Carlo estimation literature

---

> ### Author Response · Authors · 2020-11-21
> **Response to reviewer #3 with comments and improvements**
>
> We thank the reviewer for their interest and suggestions.
> We have improved our submission, here is a summary:
>
> As pointed out by the reviewer, $d^{\pi_{\theta}}(s)$ is the discounted weighting of states encountered starting at $s_0 \sim \mu_0(s)$ and following the policy $\pi_{\theta}$ and not a distribution.
>     We modified the background section in order to reflect this.  We agree that in our off-policy formulation we are ignoring the distribution shift from $d^{\pi_b}(s)$ to $d^{\pi_{\theta}}(s)$.
>     However, our off-policy objective is widely used with $d^{\pi_b}(s)$ being the discounted weighting of states when working with a start-state formulation[1] or the limiting distribution of states under $\pi_b$ in continuing problems[2,3]
>     There are works trying to correct for the distribution shift and deal with the challenge of estimating $\frac{d^{\pi_{\theta}}(s)}{d^{\pi_b}(s)}$[4].
>     We compared our methods to theirs and acknowledged the bias introduced by this formulation.
>     Note that in theorem 3.1 (theorem 3.3 in the updated version) the importance sampling correction $\frac{\pi_{\theta}(a|s)}{\pi_{b}(a|s)}$ is still required from the action-selection process when using stochastic policies.
>
> >PVF: to me this acronym is strongly synonymous with Mahadevan's proto-value functions (PVFs), circa 2007. How about "PBVF" instead? Maybe I'm old
>
> We will investigate the use of the acronym PBVF in the literature and use it instead of PVFs if it provides less overlapping.
>
> >we optimize for the undiscounted objective this should be reflected in your notation and problem formulation
>
> We clarified our usage of the discount factor.
>     In particular, when training $V(\theta)$ we ignore the discounting in the reward because we are in the episodic setting.
>     Using $V(s,\theta)$ and $Q(s,a,\theta)$ we want to predict the cumulative expected discounted reward, so we use $\gamma < 1$.
>     When training the actor, we ignore the discount factor in $d^{\pi_b}$.
>     This is a widely used approximation[5,6] and we clarified this in the paper.
>
>
> >can be used only for episodic tasks it doesn't have to. See "regenerative method" in Monte Carlo estimation literature
>
> We mentioned the regenerative method as a possible use of PSSVF for non-episodic tasks.
>
> [1]David Silver, Guy Lever, Nicolas Heess, Thomas Degris, Daan Wierstra, and Martin Riedmiller. Deterministic policy gradient algorithms. In Proceedings of the 31st International Conference on International Conference on Machine Learning - Volume 32, ICML’14, pages I–387–I–395.JMLR.org, 2014.
> [2]Thomas Degris, Martha White, and Richard S. Sutton. Off-policy actor-critic. In Proceedings of the 29th International Conference on International Conference on Machine Learning, ICML’12, pages 179–186, USA, 2012.Omnipress.
> [3] Ehsan Imani, Eric Graves, and Martha White.  An off-policy policy gradient theorem  using  emphatic  weightings.   In Advances in Neural Information Processing Systems, pages 96–106, 2018.
> [4]  Yao Liu, Adith Swaminathan, Alekh Agarwal, and Emma Brunskill.  Off-policy  policy  gradient  with  state  distribution correction.arXiv preprintarXiv:1904.08473, 2019.
> [5]  Philip Thomas.  Bias in natural actor-critic algorithms.  In International conference on machine learning, pages 441–448, 2014.
> [6]  Chris Nota and Philip S Thomas. Is the policy gradient a gradient? arXivpreprint arXiv:1906.07073, 2019.

---

### Official Review · AnonReviewer2 · 2020-10-29
**Interesting idea that is well-investigated**

**Rating:** 7
**Confidence:** 4

**Review:**

### Summary:
The paper proposes passing the parameters of a policy to the value function attempting to learn estimates of the return for that policy. This allows the value function to generalize across policies and estimate values for arbitrary policies. The paper derives several algorithms for various objectives and value functions, and empirically investigates the deterministic versions.

### Pros:
- Several new algorithms are proposed
- The new algorithms can generalize across policies
- The new algorithms can estimate the value of unseen policies

### Cons:
- Only the deterministic algorithms are empirically investigated
- Computation and memory cost seem quite high (the critic takes all of the actor’s parameters as arguments)
- Empirical results seem mixed

### Decision
I recommend accepting the paper for publication.

The paper investigates a simple, interesting, original idea—including the actor’s parameters as inputs to the critic—fairly thoroughly. Several actor-critic algorithms are derived using expressions for the gradient of various performance measures obtained by including the actor’s parameters as inputs to the critic.

The benefits of doing this are illustrated by some experiments, and the deterministic versions of the new methods are compared with reasonable competitors (DDPG and ARS) in other experiments. Unfortunately the results seem somewhat limited by the number of runs that can be conducted by parameterizing the policies and value functions as neural networks and experimenting on the chosen environments. Overall the empirical results seem mixed; in many environments it’s fine to just disregard the second part of the gradient that is dropped in DDPG and computed by PVFs. However, that’s not the fault of the new algorithms, and there are some environments where not dropping the second part of the gradient is helpful.

The paper is clearly written for the most part, with the exception of some parts of the related work that are overly terse (i.e., the connection with UVFAs could be expanded). Other parts of the related work seem frankly unrelated (i.e., predicting gradients of RNNs from their inputs in the 90s, and mapping weights of CNNs to their accuracy), and I would recommend removing them in favour of moving the more detailed comparison of PENs and PVFs into the main paper.

### Miscellaneous comments:
- Grammatical error in the final sentence of the abstract: “Their performance is comparable to the one of state-of-the-art methods”
- “In practice, like in standard actor-critic algorithms, we use a noisy version of the current learned policy in order to act in the environment and collect data” This should probably read standard deterministic actor-critic algorithms.
- I was disappointed to see that only the deterministic algorithms were implemented and analysed. Even if the stochastic versions of the algorithm are only demonstrated in a simple linear setting, that would be better than just not investigating them at all.
- The paper doesn’t mention related work that fixes the Off-PAC policy gradient theorem, which gives an expression for the true gradient of the off-policy objective without requiring PVFs (Imani 2018).
- Passing the actor’s parameters to the critic seems to necessarily break the requirement of compatible features for the actor to follow the true gradient of performance (Sutton 2000). It might be good to mention this.

### References:
1. Imani, E., Graves, E., & White, M. (2018). An off-policy policy gradient theorem using emphatic weightings. In Advances in Neural Information Processing Systems (pp. 96-106).
2. Sutton, R. S., McAllester, D. A., Singh, S. P., & Mansour, Y. (2000). Policy gradient methods for reinforcement learning with function approximation. In Advances in neural information processing systems (pp. 1057-1063).

---

> ### Author Response · Authors · 2020-11-21
> **Response to reviewer #2 with comments and improvements**
>
> We thank the reviewer for their valuable feedback.
> We have improved our submission, here is a summary:
> > The paper is clearly written for the most part, with the exception of some parts of the related work that are overly terse (i.e., the connection with UVFAs could be expanded). Other parts of the related work seem frankly unrelated (i.e., predicting gradients of RNNs from their inputs in the 90s, and mapping weights of CNNs to their accuracy), and I would recommend removing them in favour of moving the more detailed comparison of PENs and PVFs into the main paper.
>
> >The paper doesn’t mention related work that fixes the Off-PAC policy gradient theorem, which gives an expression for the true gradient of the off-policy objective without requiring PVFs (Imani 2018).
>
> We expanded the discussion on the connection with UVFAs, PENs, and alternative approaches for deriving an off-policy policy gradient theorem. We believe that it is important to mention a connection with synthetic gradients[1,2], because although they focus on supervised learning tasks, they include the possibility of learning maps from policies activations to gradients, losses or cumulative rewards, which is a setting similar to ours.
>
> >I was disappointed to see that only the deterministic algorithms were implemented and analysed. Even if the stochastic versions of the algorithm are only demonstrated in a simple linear setting, that would be better than just not investigating them at all.
>
> We agree with the reviewer on the importance of evaluating also stochastic policies. We are currently running more experiments and we will include some results for stochastic policies for the algorithms using $V(\theta)$ and $V(s,\theta)$.
>
> >Passing the actor’s parameters to the critic seems to necessarily break the requirement of compatible features for the actor to follow the true gradient of performance (Sutton 2000). It might be good to mention this.
>
> The reviewer suggested that PVFs might avoid the requirement of compatible function approximation.
>     Unfortunately, with PVFs there are still linear conditions to be satisfied in order for $V_{w}$ or $Q_{w}$ to follow the true gradient.
>     In particular, $V_w(\theta)$ needs to be linear in the policy parameters; $V_w(s,\theta)$ needs to be linear in the policy parameters and in some fixed feature of the state; for $Q_w(s,a,\theta)$ the conditions are identical to Off-PAC[3] and DPG[4], except for the requirement of Q to be linear in the policy parameters in the off-policy setting.
>     We did not include these results because in the experiments we are using nonlinear value functions.
>     However, they will be important when studying the convergence of the algorithms under linear value function approximation.
>     We mentioned these conditions in the updated version of the paper.
>
> [1] Jürgen  Schmidhuber.   Networks  adjusting  networks.   In Proceedings of ”Distributed Adaptive Neural Information Processing”, pages 197–208, 1990
> [2] Max  Jaderberg,   Wojciech  Marian  Czarnecki,   Simon  Osindero,   Oriol Vinyals,  Alex Graves,  David Silver,  and Koray Kavukcuoglu.  Decoupled neural interfaces using synthetic gradients.  In Proceedings of the 34th International Conference on Machine Learning-Volume 70, pages 1627–1635.JMLR. org, 2017.
> [3]Thomas Degris, Martha White, and Richard S. Sutton.  Off-policy actor-critic.  In Proceedings of the 29th International Conference on International Conference on Machine Learning,  ICML’12,  pages  179–186,  USA,  2012.Omnipress.
> [4]David Silver,  Guy Lever,  Nicolas Heess,  Thomas Degris,  Daan Wierstra, and  Martin  Riedmiller. Deterministic  policy  gradient  algorithms. In Proceedings of the 31st International Conference on International Conference on Machine Learning - Volume 32,  ICML’14,  pages I–387–I–395.JMLR.org, 2014.

---

> > ### Comment · AnonReviewer2 · 2020-11-23
> > **Clarification about related work**
> >
> > > We believe that it is important to mention a connection with synthetic gradients[1,2], because although they focus on supervised learning tasks, they include the possibility of learning maps from policies activations to gradients, losses or cumulative rewards, which is a setting similar to ours.
> >
> > The main issue is that including this barely-related work (different setting, different quantity being predicted, different motivations, etc.) has relegated an in-depth discussion of the most closely-related work (PENs) to the appendix. I strongly suggest removing the following chunk, or at the very least moving it to an appendix to create room for the discussion of PENs:
> >
> > > In 1990, adaptive critics trained by TD were used to predict the gradients of an RNN from its activations (Schmidhuber, 1990), avoiding backpropagation through time (BPTT) (Werbos, 1990). This idea was later used to update the weights of a neural network asynchronously (Jaderberg et al., 2017). In our work, the critic is predicting errors instead of gradients. If applied to POMDPs, or supervised learning tasks involving long time lags between relevant events, the PSSVF could avoid BPTT by viewing the parameters of an RNN as a static object and mapping them to their loss (negative reward).

---

### Author Response · Authors · 2020-11-25
**Summary of the updates in the revision**

Thanks again for the in-depth feedback by the reviewers during this rebuttal period.
We have submitted an improved version of our paper. Below we provide a summary of the most important changes.

* We changed the problem formulation, using the limiting distribution under the behavioral policy $d_{\infty}^{\pi_{\theta}}(s) = \lim_{t \rightarrow \infty} P(s_t = s| s_0, \pi_b)$ instead of the discounted weighting of states $d^{\pi_{\theta}}(s') = \int_{\mathcal{S}}\sum_{t=1}^{\infty} \gamma^{t-1} \mu_0(s) P(s \rightarrow s', t, \pi_{\theta}) \mathrm{d} s$ in the off-policy objective function. In practice, we do not have access to samples of $d_{\infty}^{\pi_{\theta}}(s)$, which is approximated by sampling trajectories.

 * We included an ablation to test whether removing $\nabla_{\theta}Q(s,a,\theta)$ from the PAVF's gradient affects the performance. We obtained a significant decrease in return in the Swimmer environment with shallow and deep policies. In Hopper the performance also dropped, but less significantly.

 * We expanded the connection between our methods and PENs [1] in the main paper and we compared our work with more algorithms trying to solve the problems of off-policy RL. We included many comparisons between our approach and traditional methods [2,3] in Sections 1,2,3 and in the related work section.

 * We included results with PSSVF and PSVF using stochastic shallow and deep policies in all environments. The results are sometimes inferior to those with deterministic policies, but can still outperform the baselines in some environments. Although the use of stochastic policies can help smoothing the objective function and allows the agent exploring in action space, we believe that the lower variance provided by deterministic policies can facilitate learning PVFs.

* We tested PSVF and PAVF in LQR and included visualization for $V(s_0, \pi_{\theta}(s_0))$ and $Q(s_0, \pi_{\theta}(s_0),\theta)$ over the policy space. The goal of this experiment was to assess whether PVFs can learn the underlying structure in the parameter space and from the results it seems that PSVF and PAVF are able to effectively bootstrap the values of future states.

 * We included more zero-shot learning results using PSSVF, PSVF and PAVF with shallow and deep policies in different environments. We obtained results similar to those in the main paper when using shallow policies and we discussed possible reasons why deep policies fail in this task when using complex environments.


[1]Jean Harb, Tom Schaul, Doina Precup, and Pierre-Luc Bacon.  Policy evaluation networks.arXivpreprint arXiv:2002.11833, 2020

[2]Thomas Degris, Martha White, and Richard S. Sutton. Off-policy actor-critic. In Proceedings of the 29th International Conference on International Conference on Machine Learning, ICML’12, pages 179–186, USA, 2012.Omnipress.

[3]David Silver, Guy Lever, Nicolas Heess, Thomas Degris, Daan Wierstra, and Martin Riedmiller. Deterministic policy gradient algorithms. In Proceedings of the 31st International Conference on International Conference on Machine Learning - Volume 32, ICML’14, pages I–387–I–395.JMLR.org, 2014.

---

### Decision · Program_Chairs · 2021-01-07
**Final Decision**

**Decision:**

Accept (Poster)

**Comment:**

The reviewers generally found the idea interesting and the contribution of the paper significant. I agree, I think this is quite a neat idea to investigate, and the paper is written well and is engaging to read.

I would encourage the authors to take into account all of the reviewer suggestions when preparing the camera-ready version. Of particular importance is the name: I think it's bad form to appropriate a name already used in other prior work (proto-value functions, which are very well known in the RL community), so I think it is very important for the final to change the name to something that does not conflict with an existing technique. Obviously this does not affect my evaluation of the paper, but I trust that the authors will address this feedback (I will check the camera-ready).